# Translation Heads: Disentangling meaning from language in LLM-based machine translation

**Théo Lasnier** [* 1]  **Armel Zebaze** [* 1]  **Djamé Seddah** [1]  **Rachel Bawden** [1]  **Benoît Sagot** [1]

## Abstract

Mechanistic Interpretability (MI) seeks to explain how neural networks implement their capabilities, but the scale of Large Language Models (LLMs) has limited prior MI work in Machine Translation (MT) to word-level analyses. We study sentence-level MT from a mechanistic perspective by analyzing attention heads to understand how LLMs internally encode and distribute translation functions. We decompose MT into two subtasks: producing text in the target language (i.e. target language identification) and preserving the input sentence's meaning (i.e. sentence equivalence). Across three families of open-source models and 20 translation directions, we find that distinct, sparse sets of attention heads specialize in each subtask. Based on this insight, we construct subtask-specific steering vectors and show that modifying just 1% of the relevant heads enables instruction-free MT performance comparable to instruction-based prompting, while ablating these heads selectively disrupts their corresponding translation functions. We made available the code at: https://github.com/Blyzi/mitra

## 1. Introduction

Traditionally, research on Large Language Models (LLMs) for Machine Translation (MT) has focused on deriving insights by benchmarking translation performance across multiple language pairs (Jiao et al., 2023; Zhu et al., 2024b). Previous work has looked at the effect of factors such as the level of resourcedness of the languages involved (Moslem et al., 2023; Tanzer et al., 2024; Zebaze et al., 2025b), the template used (Bawden & Yvon, 2023; Zhang et al., 2023), the selection, construction and the number of in-context

---
[*]Equal contribution  [1]Inria, Paris, France. Correspondence to: <firstname.lastname@inria.fr>.

*Proceedings of the 43rd International Conference on Machine Learning*, Seoul, South Korea. PMLR 306, 2026. Copyright 2026 by the author(s).

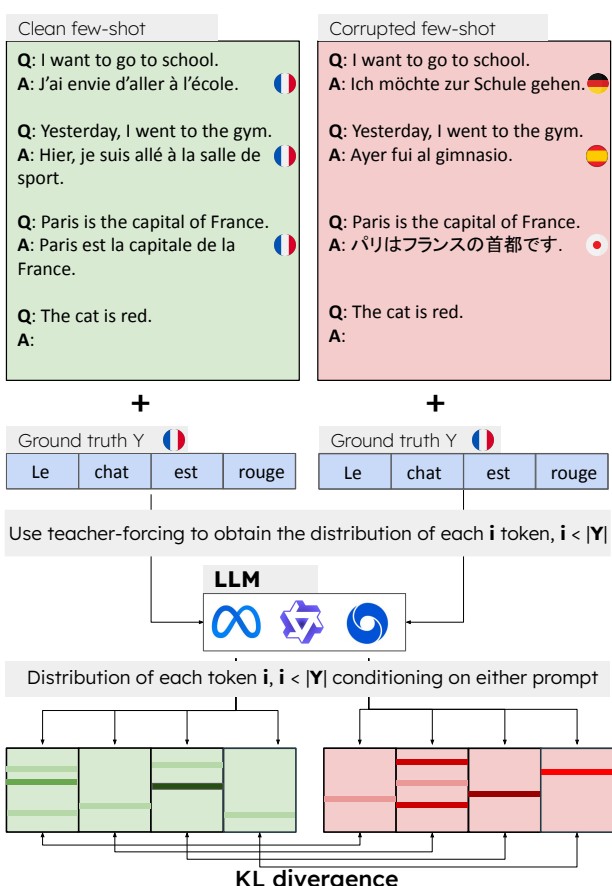

*Figure 1.* We consider two few-shot MT prompts: a clean prompt and a corrupted prompt (obtained in this case by mixing multiple target languages) and both ending with the same query. For each prompt, we apply teacher forcing and compute the token-level output distributions for the ground-truth query. At each target position, we measure the KL divergence between the distributions conditioned on the clean and corrupted prompts.

demonstrations (Vilar et al., 2023; Zebaze et al., 2025a; Li et al., 2025; Lee et al., 2025). While useful, these studies leave unanswered the question of how models encode key information when performing the MT task. This absence of mechanistic explanation limits our ability to understand how LLMs internally represent MT as a whole and which model components are most engaged. Recent advances in mechanistic interpretability (MI), a body of research aim-

ing to reverse-engineer neural network computations (Olah et al., 2020; Elhage et al., 2021), offer new tools to address this gap. Cross-lingual works have mostly focused on building representations across multiple languages (Conneau et al., 2020; Pires et al., 2019) and investigating factors that govern cross-lingual transfer success (Lauscher et al., 2020; Muller et al., 2021; Philippy et al., 2023). A few studies have examined MT through the lens of MI. However, existing work has largely focused either on word-level translation—aiming to construct task vectors (Hendel et al., 2023; Todd et al., 2024) or identify important attention heads (Zhang et al., 2025)—or on sentence-level phenomena that only indirectly affect translation performance. For example, Liu et al. (2026) identified a subset of attention heads responsible for multilingual token alignment across translation directions, but it remains unclear whether these heads outputs result in useful task representations and whether steering with them leads to strong MT performance. Similarly, Wang et al. (2024) studied heads involved in repetition errors and incorrect language prediction in MT, but steering with their MT vectors resulted in limited or inconsistent performance gains. Moreover, these works only superficially investigate whether different translation directions share similarities in the model components they primarily engage and in their task representations.

In this work, we study sentence-level MT with the aim of causally linking few-shot translation performance to specific attention heads. To this end, we use activation patching (Vig et al., 2020a), a method that quantifies the importance of model components by contrasting model behavior on clean and corrupted prompts. As illustrated in Figure 1, a corrupted prompt closely resembles a clean one but is designed in such a way to lead to an incorrect prediction according to a particular aspect (here, a translation in an incorrect target language, cf. Section 3.2). By intervening on a given component and measuring how the log-probability of the correct next token changes, we can assess that component's contribution to the task. While activation patching has previously been applied to word-level MT, we extend it to the sentence-level setting. Additionally, we propose to decompose the MT task into two subtasks each reflecting an aspect of MT: generating in the correct target language (target language identification) and producing an output equivalent in terms of meaning to the source sentence (sentence equivalence). These analytical choices are motivated by three factors: (i) attention mechanisms play a central role in in-context learning (ICL) (Olsson et al., 2022; Von Oswald et al., 2023); (ii) LLM-based few-shot MT has been extensively studied through output-based evaluation, providing a foundation for connecting empirical observations with mechanistic explanations; and (iii) incorrect target-language generation is among the most common failure modes of LLM-based MT (Wang et al., 2024).

We show across 20 language directions and multiple models families (Gemma-3 (Gemma Team et al., 2025), Qwen3 (Yang et al., 2025) and Llama-3 (Grattafiori et al., 2024)) that: (i) A small subset of attention heads ($\approx$1%) play a considerable role in MT performance across all pairs, demonstrating that translation is mechanistically localized, (ii) target language identification and sentence equivalence are encoded by distinct, separable sets of heads, and (iii) this organization holds across the 20 translation directions. Based on these findings, we construct function vectors for language and for semantic equivalence, and show that steering $\approx$1% of heads from each group enables strong MT performance when using prompts without task or language indications. Conversely, ablating the identified heads significantly impairs the model's ability to perform MT compared to ablating random heads, and in particular, ablating language heads leads to a loss of the ability to generate translations in the correct target language. Moreover, we observe that equivalence vectors are transferable from one translation direction to another with minimal MT performance loss.

## 2. Related Work

**LM Interpretability** MI provides a framework to explore the role of LM components by reverse-engineering neural network computations (Olah et al., 2020; Elhage et al., 2021). Rather than probing for linguistic properties in learned representations, MI aims to identify the specific components (attention heads, neurons, etc.) that drive model behavior. Foundational work established that transformer computations can be decomposed into interpretable subgraphs (Elhage et al., 2021), leading to the discovery of specialized structures such as induction heads for ICL (Olsson et al., 2022) and circuits for indirect object identification (Wang et al., 2023). Techniques such as activation patching (Vig et al., 2020a; Meng et al., 2022) enable causal attribution by measuring how interventions on specific components affect model outputs. A consistent finding across MI studies is that complex capabilities often emerge from surprisingly sparse subsets of model parameters (Voita et al., 2019; Conmy et al., 2023), suggesting a strong localization of high-level behaviors.

**Multilingual Representations** The ability of multilingual language models to generalize across languages has motivated substantial research into the nature of cross-lingual representations (Pires et al., 2019; Wu & Dredze, 2019; Conneau et al., 2020). Early work demonstrated that multilingual pre-training induces a shared representation space that supports zero-shot cross-lingual transfer (Artetxe et al., 2020; Karthikeyan et al., 2020; Muller et al., 2021), while subsequent studies have investigated the factors governing transfer success, including linguistic typology, lexical overlap, and pre-training data composition (Philippy et al., 2023;

Lauscher et al., 2020). A recurring finding is that models often route multilingual inputs through English-centric latent representations before task resolution (Wendler et al., 2024; Zhao et al., 2024; Schut et al., 2025), a phenomenon observed across modalities (Wu et al., 2025), which may explain why English-only instruction tuning generalizes to other languages (Li et al., 2024; Shaham et al., 2024). However, these studies primarily characterize what representations emerge rather than how the underlying computations are organized, leaving open the mechanistic question of which model components mediate cross-lingual behavior.

**MT and MI**   LLM-based MT has been studied from diverse perspectives, including the impact of prompt templates, in-context demonstrations, and the level of language resourcedness (Moslem et al., 2023; Bawden & Yvon, 2023; Zhang et al., 2023; Hendy et al., 2023; Zhu et al., 2024a; Zebaze et al., 2025b), but primarily with a focus on analyzingoutputs rather than the models' internal mechanisms. More recently, a few studies have begun to explore MT from an MI perspective. Wang et al. (2024) use activation patching to construct task vectors (Todd et al., 2024; Hendel et al., 2023) for MT and identify attention heads to intervene on, with the goal of mitigating language mismatch and repetition issues in LLM-based translation. They use activation patching on full ground-truth target sentences and identify heads by measuring how their probability changes. However, the translation vectors they build do not improve translation quality despite reducing language mismatch errors. Zhang et al. (2025) similarly aim to identify attention heads that influence translation behavior with activation patching, but they focus on word-level MT . Their perturbations alter the instruction and/or remove mentions of the target language. In contrast, we rely on ICL to eliminate the effect of task-specific instructions, aligning with our goal of identifying attention heads that are known to play a central role in ICL (Elhage et al., 2021). We further support our findings by evaluating translation quality under inference-time steering and reconciling our conclusions with established empirical observations. Finally, Liu et al. (2026) has a different focus, which is identifying the heads responsible for token alignment during MT (i.e. aligning generated tokens with their corresponding source token) (Brown et al., 1990; Vogel et al., 1996; Tillmann et al., 1997) They show the importance of the identified heads through ablation, but do not build a high-level representation of the MT task itself. Questions therefore remain about how steering the heads could impact MT performance.

## 3. Methodology

We investigate the internal mechanisms of MT in transformer-based LLMs by identifying which attention heads encode task-relevant information. We reasonably hy-

pothesize that the success of the MT task depends on two aspects: (i) *target language identification*, which consists in identifying the language in which the output is expected to be (it is crucial as generating in an incorrect language is one of the most common failure cases of LLM-based MT (Wang et al., 2024)), and (ii) *sentence equivalence*, i.e. writing a sentence equivalent to the source, which is the gist of the MT task. Our approach is illustrated in Figure 1. For this purpose, we use activation patching (Vig et al., 2020b) with contrastive prompts designed to isolate the most important heads for each aspect (subtask) of MT.

### 3.1. Activation Patching

Activation patching is a causal intervention technique that measures a component's importance by replacing its activations under a corrupted input with those from a clean input (Vig et al., 2020b; Meng et al., 2022). Given a clean prompt $c$ and a corrupted prompt $\tilde{c}$ (e.g., Section 3.2), and a model component $l$ (e.g., an attention head) whose importance we wish to assess, we denote by $a^{(l)}(c)$ and $a^{(l)}(\tilde{c})$ the activations of $l$ under $c$ and $\tilde{c}$, respectively. Let $\bar{a}^{(l)}(c)$ represent the mean activation of component $l$ when varying $c$. We run a forward pass on $\tilde{c}$ but intervene by substituting $a^{(l)}(\tilde{c})$ by $\bar{a}^{(l)}(c)$, then measure the change in model output log probabilities using the following metric:

$$\Delta_l = \log p\left(y \mid \tilde{c},\, a^{(l)}(\tilde{c}) \leftarrow \bar{a}^{(l)}(c)\right) - \log p(y \mid \tilde{c})$$

A large $\Delta_l$ indicates that component $l$ carries information about the difference between clean and corrupted inputs. In word-level MT, clean prompts typically take the form of few-shot examples such as "*Green → Vert, Red → Rouge, Yellow →*" (here for English→French), the expected output being "*Jaune*". In this setting, activation patching directly measures the influence of a component on the next-token prediction, which corresponds to the entire answer or at least a good part of it. This setup has been used in prior work on word-level MT (Todd et al., 2024; Zhang et al., 2025; Dumas et al., 2025). However, when the target output $y$ spans multiple tokens, as it does in sentence-level MT, we must identify which token position (and therefore the chunk of the answer) encodes the representation of interest in order to meaningfully assess the heads' impact on the model's predictions. We refer to this as the token position problem.To identify the position, we propose a KL-divergence-based method to select the token index where the model output distributions differ the most between the two settings. For each position $i \in \{0, \ldots, |y| - 1\}$ in the answer, we compute the KL divergence between the model's output distribution under clean and corrupted contexts, both conditioned on the same partial answer $y_{<i}$. We use teacher-forcing, meaning that the partial answer $y_{<i}$ is extracted from the ground truth $y$. We stop at $i = |y| - 1$ because we are not interested in the distribution of the token following the final token of $y$,

as it initiates a new sentence whose connection to the current task is unclear. We select the position with maximum divergence,

$$i^* = \operatorname{argmax}_i D_{\mathrm{KL}}\left(\mathbb{P}\left(\cdot|c, y_{<i}\right) \| \mathbb{P}(\cdot|\tilde{c}, y_{<i})\right),$$

as this is where the two contexts produce the most different distributions, likely indicating that this is where the representation we seek is the most influential. We then apply the usual activation patching on $c \oplus y_{<i^*}$ as the clean prompt and $\tilde{c} \oplus y_{<i^*}$ as the corrupted prompt.

### 3.2. Isolating aspects of the MT task

We assume we have access to a multiway and multilingual dataset $\mathcal{D}$ containing the same $N$ sentences written in multiple languages. Given a source sentence $x$ in language $L_{src}$ and its translation $y$ in language $L_{tgt}$, we construct a $k$-shot prompt $c_{clean}^k$, using $k$ pairs $\{(x_i, y_i)\}_{i=1}^k \subset \mathcal{D}; k < N$ as in-context demonstrations followed by the test query $x$.[1] The prompt used is as follows: Q:$x_1$\nA: $y_1$\n\n...Q: $x_k$\nA: $y_k$\n\nQ: $x$\n A:.

**Corrupted Prompts** We construct two types of corrupted prompts $c_{corrupt}^k$ based on $k$ demonstrations, each corresponding to the specific subtask that we aim to isolate.

- *Target language identification*: the prompt $c_{lang}^k$ uses the same $k$ demonstrations as $c_{clean}^k$ but replaces each target side with a correct translation in a language selected from the set $\mathcal{L} = \{$French, Spanish, Portuguese, Japanese, Chinese, Hindi, Arabic, and Russian$\}$ and different from both $L_{tgt}$ and $L_{src}$, thereby preserving the meaning of the source sentence whilst removing target language information.

- *Sentence equivalence*: the prompt $c_{MT}^k$ uses the same demonstrations as $c_{clean}^k$ but replacing example translations with random sentences in the correct output language, excluding the correct translations. All targets are written in $L_{tgt}$ but we remove the "correspondence of meaning" that defines the translation task.

**Head Identification** Given the activation patching describe above for sentence-level outputs (Section 3.1), we identify important heads for each type of corruption ($c_{lang}^k$ and $c_{MT}^k$), each corresponding to a subtask of the main MT: language heads and translation heads.

**Steering** To validate the fact that the identified heads represent the target language identity and sentence equivalence,

we test whether their activation combinations can induce translation in an anonymous zero-shot setting using the following prompt: Q: $x$\nA:. For each identified head, we compute a steering vector by averaging its activations across multiple $c_{clean}^k \oplus y_{<i^*}$ and then multiplying it by the output projection. We then build two sets of steering vectors: language vectors based on the activations of language heads and (sentence-)equivalence vectors based on the activations of translation heads. During generation, we add the steering vectors, scaled by a constant $\alpha$ (which we call amplification factor), to the residual stream at their respective layers for each generated tokens. Our aim is to investigate whether jointly steering $m\%$ of language heads and $n\%$ of translation heads enables an LLM to perform MT as effectively as when it is prompted with a task description specifying the source and target languages (using the following instructed zero-shot prompt: Translate from *source* to *target*:Q: $x$\nA:). Additionally, we investigate setting the outputs of the top-$m\%$ language heads and top-$n\%$ translation heads to zero and evaluate MT performance using the same instructed zero-shot prompt. We examine whether their removal degrades MT performance, using the ablation of $(m + n)\%$ randomly selected heads (resampled per generation) as a control to distinguish task-specific effects from general capacity reduction.

## 4. Experimental Setup

**Models and Data** We experiment with four families of base LLM: Gemma-3 (Gemma Team et al., 2025) (270M, 1B, 4B, 12B, 27B), Llama-3.2 (Grattafiori et al., 2024) (1B, 3B), Llama-2-7B (Touvron et al., 2023) and Qwen-3 (Yang et al., 2025) (0.6B, 1.7B, 4B). We use the FLORES-200 multi-way parallel dataset (Goyal et al., 2022; Costa-jussà et al., 2022), which covers over 200 languages. We use the *dev* set (997 samples) to build the prompts to identify the relevant attention heads and to build steering vectors, and the *devtest* set (1012 samples) for evaluation. We investigate 20 language directions: English from and into Arabic, Chinese, French, Hindi, Japanese, Portuguese, Russian, Spanish, Swahili and Wolof.

**Evaluation metrics** We consider two main metrics: BLEU[2] (Papineni et al., 2002) and MetricX-24-Hybrid-XXL (Juraska et al., 2024), a neural metric based on mT5 (Xue et al., 2021) and highly correlated with human judgments. BLEU scores range from 0 to 100 (higher is better), and MetricX scores range from 0 to 25 (higher scores indicate more errors). We also use FastText (Joulin et al., 2016) to compute the target language accuracy. We provide complementary results with

---

[1]The prompt $c_{clean}^k$ contains no explicit instruction or header, to force the model to infer the task solely from the input-output demonstrations.

[2]nrefs:1|case:mixed|eff:yes|tok:flores200|smooth:exp |version:2.6.0

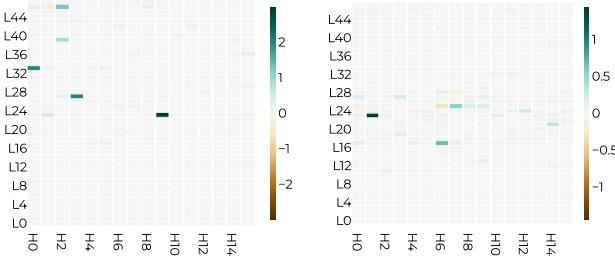

*Figure 2.* Average log probability deltas per layer and attention head in GEMMA-3-12B-PT under language (left) and translation (right) corruption (averaged across the 20 language directions)

chrF++ (Popović, 2015; Popović, 2017) and XCOMET-XXL (Guerreiro et al., 2024) in Appendix B.3.

**Implementation Details** Unless explicitly stated otherwise, we use $k = 5$ demonstrations to identify the language heads and translation heads. Across all setups, we perform greedy decoding (temperature = 0) and generate at most 100 tokens with $\alpha = 1$ as an amplification factor.

## 5. Results

**Identified heads** Figure 2 presents the log-probability deltas obtained through activation patching under translation and language corruptions, averaged across our twenty translations directions (for language-direction-specific results, see Appendix B.1, Figures 19 and 18). We observe that the attention heads that are consistently important for MT are remarkably sparse. Across all translation directions examined, only 5 to 10 heads—representing fewer than 1% of the total attention heads (approximately 5 out of 1,024 in GEMMA-3-12B-PT)—show markedly higher contributions to recovering the correct (known) next token (i.e. $y_{i*}$, see Section 3.1). This is in line with the findings of Zhang et al. (2025) in word-to-word MT where they found 5% of heads to be most important. Furthermore, these heads remain largely invariant across translation pairs, as evidenced by the strong visual correspondence between the mean activation pattern and individual language directions (Figures 19 and 18). This consistency suggests that MT specializes the same set of heads for each aspect across different directions. Additionally, comparing both tables in Figure 2 reveals that the heads encoding target language identity and those encoding language-agnostic sentence equivalence constitute mostly disjoint sets; the Jaccard index of the top 5% language heads and translation heads is 0.13. We provide all detailed activation patching results and Jaccard Index overlap for the Gemma-3, Qwen-3 and Llama-3.2 model families for all studied translation directions in Appendices B.1 and B.2.

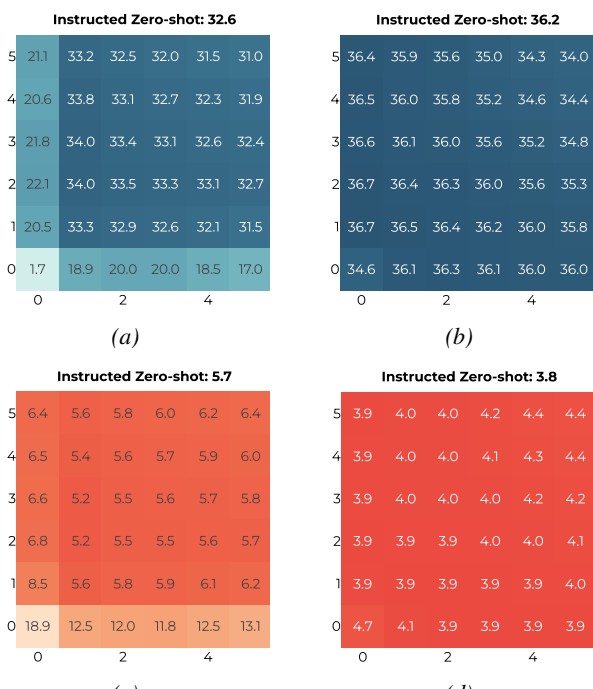

*Figure 3.* MT performance of GEMMA-3-12B-PT in an instruction-free zero-shot prompt when steering $n\%$ translation heads (x-axis) and $m\%$ language heads (y-axis) with the average outputs of these heads under few-shot examples. (a) and (b) report average BLEU scores for all translation pairs with English as source and target language, respectively. (c) and (d) report MetricX-24 scores in the same setup. Above each figure is the topline score (in an instructed zero-shot setting).

**Steering** To validate the fact that the identified heads are sufficient to induce translation behavior, we construct steering vectors from their averaged activations under clean few-shot contexts and apply them during generation with an instruction-free zero-shot prompt. Figure 3 presents MT performance as a function of the proportion of language heads and translation heads steered. The results demonstrate that steering a remarkably small subset of heads (approximately 1% each of language and translation heads) enables the model to achieve translation quality comparable to that obtained with explicit task instructions. This finding confirms that the identified heads encode compact, transferable representations of the MT task that can be used to elicit translation without any in-context demonstrations or task specification. Notably, both components are necessary for successful translation: steering only language heads or only translation heads results in degraded performance, as evidenced by the low scores along the axes of each heatmap (3a,3c). This complementarity confirms our hypothesis that MT decomposes into target language identification and sentence equivalence, with each function requiring its dedicated set of attention heads. We provide examples of failure cases when steering language heads only or translation heads only

when translating from English to French in Appendix B.3.1.

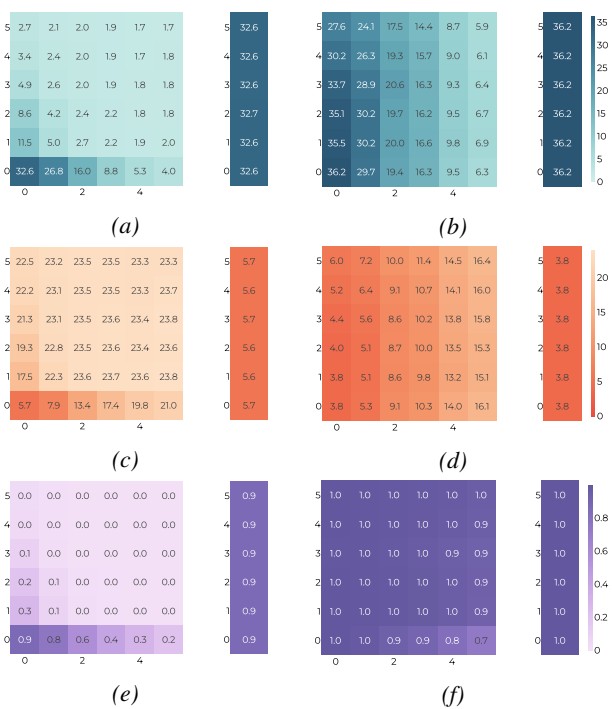

*(a)* *(b)*

*(c)* *(d)*

*(e)* *(f)*

*Figure 4.* MT performance of GEMMA-3-12B-PT when ablating $n\%$ translation heads (x-axis) and $m\%$ of language heads (y-axis) in an instructed zero-shot setup. (a) and (b) report average BLEU scores for all translation pairs with English as source and target language, respectively. (c) and (d) report MetricX-24 scores and (e) and (f) report the target language accuracy in the same setup. Results are compared against ablating $j\%$ of randomly selected heads (column matrix on the right).

**Head Ablation**   While the steering experiments demonstrate that the identified heads are sufficient to induce translation, we now assess whether they are necessary by ablating their activations during inference with the instructed zero-shot prompt. Figure 4 presents MT performance as a function of the proportion of language heads and translation heads ablated. The results reveal that removing as few as 1% of these heads substantially degrades translation quality, confirming their critical role in MT. Interestingly, we observe an asymmetric pattern depending on the translation direction. For translation pairs with English as the source language (Figures 4a and 4c), ablating language heads induces a sharp performance collapse, whereas ablating translation heads leads to a more gradual decline. The target language accuracy plots (Figure 4e) reveal the underlying cause: ablating even 1% of language heads renders the model unable to generate in the target language, explaining the precipitous drop in translation quality. The LLM switches between multiple languages within the same sentence and this considerably impacts string-matching metrics such as BLEU. When it comes to translation heads, it is more nuanced: there is a lot

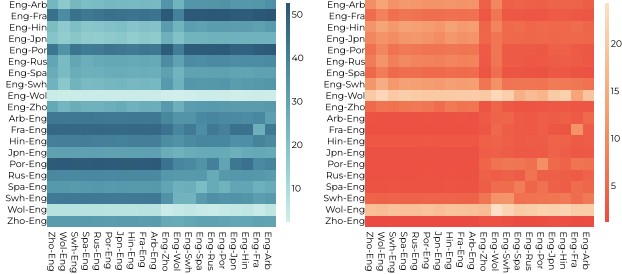

*Figure 5.* MT performance of GEMMA-3-12B-PT in a instruction-free zero-shot prompt when steering 1% of language heads and translation heads by varying the language pairs used to create the language agnostic sentence equivalence (x-axis) and the target language identity (y-axis) steering vectors. We report average BLEU (left) and MetricX-24 (right) scores with the target language identity as reference.

of repetition of the source but also translation attempts that are just bad quality. On the other hand, for translation pairs with English as the target language (Figures 4b and 4d), translation head ablation causes drastic degradation while language head ablation produces a more progressive decline. Figure 4f shows that when translating into English, the model continues to generate in the correct target language regardless of whether language heads or translation heads are ablated, but translation quality degrades substantially according to both BLEU and MetricX. This asymmetry likely reflects the model's English-centric pre-training: generating English text is a strong default behavior that persists even under ablation, whereas generating non-English languages critically depends on language heads to override this default. Comparison with the random ablation baseline confirms that these effects are specific to the identified heads rather than a consequence of general capacity reduction.

**Equivalence vectors transferability.**   In this experiment, we investigate whether steering an LLM with language vectors obtained for a direction $\texttt{Src}_1\texttt{-Tgt}_1$ and sentence equivalence vector of a difference direction $\texttt{Src}_2\texttt{-Tgt}_2$ would still induce translation into $\texttt{Tgt}_1$. Here we consider GEMMA-3-12B-PT and 20 directions (10 from English and 10 into English). For each direction, we steer 1% of language heads and 1% of translation heads and analyze the impact of varying the direction ($\texttt{Src}_2\texttt{-Tgt}_2$) associated with the translation heads. BLEU and MetricX scores are shown in Figure 5. The translation scores stays approximately the same for the rows of each matrix, indicating that equivalence vectors derived from alternative directions can work just as well as the direction of interest (provided by the language vectors). Low-resource target languages such as Wolof are an exception, because LLMs difficulty generating such languages (Zebaze et al., 2025b) is directly reflected in the quality of their representations of directions involving it.

*Table 1.* First 20 tokens obtained by decoding the steering vectors from the top-1 *Translation Head* and top-1 *Language Head* of GEMMA-3-4B-PT for five translation directions. For each direction, we project the mean head output onto the vocabulary space and report the highest-ranked tokens.

| Language Pair | Translation | Language |
|---|---|---|
| Eng-Fra | í ś ū Pérez = ₱ Rö   ization [ ī Pokémon ó чи у    īg Ré | French à pour dé Quebec Québec french é France É ou és rés ém Ré le en é Dé ès |
| Eng-Swh | = í impactful ū 에서 ś And и    y за в чи XNUMX ó [ And | Tanzania African Africa Kenya Kenyan ya Zambia Africans view Mp ( y . Malawi in Rwanda Kenyans m Nigeria |
| Eng-Zho | в у В Sociedad Chién за ной Trường Employees Employee Nguyễn الو ' ском ⊏ об Bronx Тру Yıld | （， ： 其他 ； 对 可以 做 （很 后 其他 或 从 上 此外 两 应 。 |
| Eng-Arb | = ś    ₱ í чи    [    And And Grü 에서 ū 有 Pérez Rö impactful | ه عم ـ نأ يف ن ط ش ح ر نا ق نم ص ب ا كـ ا لـ ع |
| Eng-Por | í ś ū = ₱ Pérez ī impactful ización y Pokémon czek B    и Rö ì | Brazilian Portuguese Brazil brazilian Portugal brasil Brasil portug São brazil Recife Chilean Urugu por Oliveira brasile João soccer ou Brazilian |

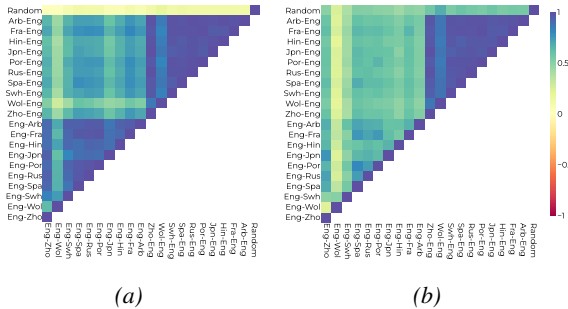

*(a)*        *(b)*

*Figure 6.* Cosine similarity of mean head outputs across translation directions for GEMMA-3-12B-PT, computed in a few-shot setup. For each translation direction, we extract the average activation output of the top-1 Translation Head (a) and top-1 Language Head (b) identified through activation patching, then compute pairwise cosine similarities.

To probe this geometrically, we report in Figure 6 the pairwise cosine similarities between head outputs across directions for GEMMA-3-12B-PT. For each direction, we extract the mean output of the top-1 *Translation head* and the top-1 *Language head* over the clean few-shot prompts, yielding one representative vector per head and direction. To account for model anisotropy, we use as a baseline a vector obtained by averaging the output of the same heads over 1000 prompts randomly sampled from FineWebEdu (Lozhkov et al., 2024). For equivalence vectors (Figure 6a), similarities between language pairs generally fall in the range 0.7–0.9, while similarity to the baseline stays close to 0.0, confirming that these vectors capture structure rather than head-level artifacts. Pairs that share either a source or target language reach similarities around 0.9. For language vectors (Figure 6b), the highest similarities correspond to pairs with matching target languages; otherwise, similarities collapse to the baseline, with a few exceptions such as Japanese–Chinese, Portuguese–Spanish, and Spanish–French, which plausibly reflect shared script or linguistic proximity. In both analyses, directions involving Wolof consistently exhibit the lowest similarities, trailing other pairs by roughly 0.3 points. We attribute this to the model's poor performance in Wolof, which itself reflects the limited pretraining data

available for this LRL. Moreover, when decoding (unembedding) equivalence vectors, we observe that the decoded tokens span a wide variety of scripts and languages – including Arabic, Chinese, English, French, Korean, Russian and Vietnamese for GEMMA-3-4B-PT in Table 1 – with consistent overlap across multiple translation directions. In contrast, decoding language vectors predominantly yields tokens in the target language of the corresponding direction (Appendix B.6). Specifically, for GEMMA-3-4B-PT, decoding language vectors produces tokens closely associated with the target language (e.g., *French, Québec, and France* for French; *Tanzania, Africa, and Kenya* for Swahili; and *Brazilian and Portuguese* for Portuguese). These results further support our transferability hypothesis and clarify the semantic meaning captured by our task vectors.

## 6. Discussion and Ablation studies

We run ablation studies to better understand the implications of our findings. Since few-shot MT performance generally improves with additional in-context demonstrations, does more demonstrations help to build better (more expressive) task vectors? We also ablate potentially sensitive hyperparameters such as the amplification factor and the token position for head identification (see Section 3.1). Choosing the token position at which to perform activation patching is a crucial design decision, since sentence-level outputs, unlike word-level outputs, introduce additional degrees of freedom that remain largely underexplored. Finally, we compare our heads with important heads reported by previous works and classify them into language heads and translation heads whenever possible.

**Impact of the number of shots**   We investigate the sensitivity of our head identification procedure and steering to the number of in-context demonstrations $k$. The identified heads remain remarkably consistent across shot counts, with the exception of the zero-shot setting, for which all deltas are low—an expected outcome given the absence of demonstrations from which to infer task structure. This stability suggests that even a single demonstration suffices to

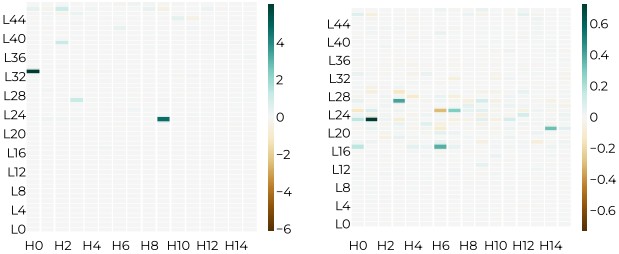

Figure 7. Log probability delta per layer and attention head in GEMMA-3-12B-PT under language (left) and translation (right) corruption for English→French, for the 1-shot scenario.

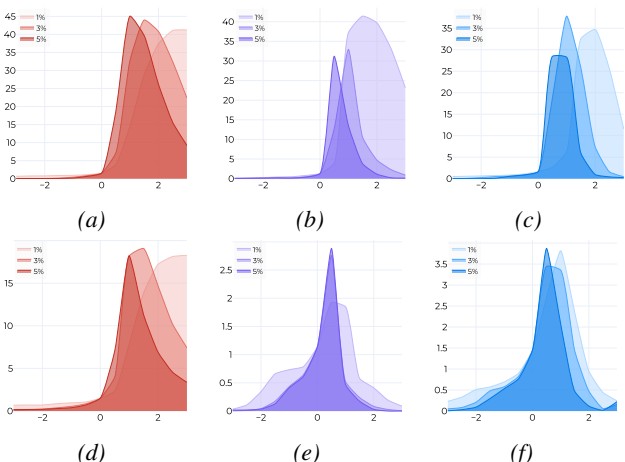

Figure 8. Effect of the amplification factor on steering-induced MT performance. We report BLEU scores under an instruction-free zero-shot setup for three model families: GEMMA-3-4B-PT (a, d), QWEN3-4B-BASE (b, e), and LLAMA-3.2-3B (c, f) for the English to French (a-c) and English to Swahili (d-f) translation direction. The x-axis denotes the amplification factor $\alpha$, while each curve corresponds to a different proportion of language heads and translation heads steered.

accurately identify and model the MT task. Figure 7 shows the activation patching results for GEMMA-3-12B-PT on English→French under language and translation corruptions for $k = 1$ shot. The corresponding figures for all values of $k \in \{0, 1, 20\}$ are provided in Appendix C.1 (Figures 68 and 69). We further investigate the result of steering 1% of language heads and 1% of translation heads with vectors obtained with different values of $k$ and show the BLEU scores in Figure 70. The BLEU with steering consistently follow the pattern of the BLEU obtained with few-shot prompting. However we note a little drop of performance for high values of $k$, which can be thought as a ceiling we face when trying to "compress" the information given by $k$ demonstrations within the same number of vectors as $k$ grows bigger.

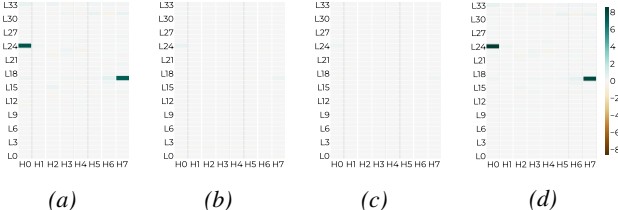

Figure 9. Log probability delta per layer and attention head in GEMMA-3-4B-PT under language corruption for English→French. We report activation patching results (a-d) for fixed positions {0, 2}, random and KL-based (ours), respectively.

**Impact of the amplification factor** We run an ablation study to understand the impact of the amplification factor $\alpha$ on the MT performance during steering. In this experiment, we consider the English→French direction, with three models GEMMA-4B-PT, QWEN3-4B and LLAMA-3.2-3B. We summarize the BLEU scores in Figure 8. Negative amplification factors are extremely disruptive, while the best range for the amplification factor falls within 0.5 and 3. Interestingly, the fewer heads we intervene on, the greater the amplification factor should be for better MT performance. However, a too big factor can ultimately break the LLM.

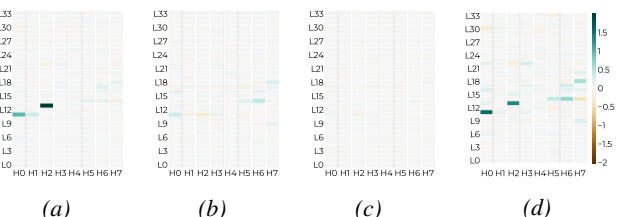

Figure 10. Log probability delta per layer and attention head in GEMMA-3-4B-PT under translation corruption for English→French. We report activation patching results (a-d) for fixed positions 0, 2, random and KL-based (ours), respectively.

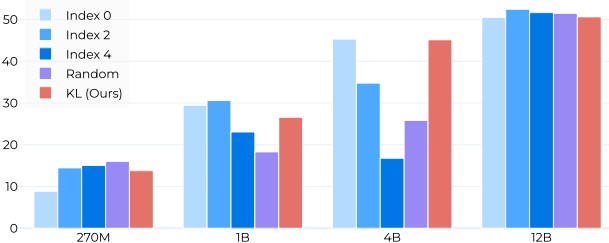

Figure 11. Effect of the studied token index on MT performance for English→French when steering with 5% of the language and translation heads. We report BLEU scores under an instruction-free zero-shot setup across four Gemma-3 models. Each bar corresponds to a different token selection strategy: fixed positions ($i \in \{0, 2, 4\}$), random selection, and our KL-based method.

**Impact of the studied token position** We also study the impact of the selection token strategy on the identified heads.

We consider three alternatives: prompts without any ground truth tokens ($i = 0$), prompts with a few tokens of a ground truth (we choose $i = 2$), and random selection i.e. uniformly choosing $i \in [0, |y| - 1]$ and considering the first $i$ tokens of $y$. We compare each strategy to our KL-based approach on the identification of language heads (Figure 9) and translation heads (Figure 10). Interestingly, token position zero works well. In Figure 9, it identifies the same top 2 attention heads that most prioritize language identification ($L_{24}, H_0$) and ($L_{17}, H_7$) as the KL-method. This is intuitively plausible because except in a few cases (punctuation, proper nouns, etc.), the first token can be enough to know which language to generate into and so its distribution there can be used to identify language heads. Surprisingly, the overlap between our KL-method and the usual first-token approach persists when identifying translation heads. Even index 2 and random selection have similar top-heads as index 0, for translation and language. This means that many token positions in the target $y$ are reasonable choices for head identification as they will give similar heads. As shown in Figure 11, the sensitivity to token position selection varies with model scale. GEMMA-3-12B-PT achieves consistent performance across all strategies. However, as model capacity decreases, the selection strategy becomes increasingly consequential, with smaller models showing greater variance across positions. Notably, the first token position tends to perform well for mid-sized models, while smaller models benefit more from later or randomly selected positions. Our KL-based method provides a principled, adaptive criterion that achieves competitive performance across scales without requiring manual tuning, making it a practical default when the optimal position is unknown *a priori*.

**Classification of previous works' heads**  Zhang et al. (2025) identified important heads for word-level MT with LLAMA-2-7B for English→Chinese. While they could not give a classification for their heads, we found, by applying analyzing the same model in our setup, that four of their top-5 heads ([30, 18], [14, 10], [12, 17], [16, 26]) were classified as language heads in our framework. Their eighth head ([15, 9]) was our third translation head. However, their most important head ([31, 8]) was nowhere to be found in our classification as its $\Delta_l$ was close to zero. Similarly, using LLAMA-2-7B, Wang et al. (2024) identified 12 heads as being consistently important for MT across multiple languages. Three of their top-5 heads ([9, 25], [12, 15] and [12, 28]) belong to our top-5 English→Chinese translation heads. These overlaps suggest that our setup seamlessly extends the word-to-word setup and our method recovers heads found via different approaches. Finally, the token alignment heads identified by Liu et al. (2026) for QWEN-1.7B-BASE were not found in ours and we attribute this to the difference of objective in our head identification process.

# 7. Conclusion

We explored MI tools to better understand how LLMs carry out the MT task, with a particular focus on attention heads. We proposed a two-aspect decomposition of MT: across a broad range of experiments, we find that certain attention heads primarily drive the choice of the target language, while others focus on meaning equivalence. We find that language and translation heads are largely consistent across translation directions, with minimal overlap between the two groups. Building on these findings, we construct task vectors, which we use to steer a small subset of heads to substantially improve MT performance under prompts without task and language indications. Notably, equivalence vectors transfer across language directions with minimal performance loss performance. Through ablation, we demonstrate that these heads play a causal role: disabling them results in failure modes that reflect their associated functions. Lastly, we show that analyzing the first target token prediction is sufficient to identify important heads for sentence-level MT, extending observations made for word-level MT.

Overall, these results indicate that our decomposition is an effective way of analyzing MT, enabling the isolation of head-level specializations and the construction of direction-agnostic equivalence vectors. Our work represents a step toward a deeper understanding of MT through the lens of MI beyond word-to-word mappings and opens up interesting directions for future research on cross-lingual task representations and understanding.

# Acknowledgments

This work was partly funded by Rachel Bawden and Benoît Sagot's chairs in the PRAIRIE institute funded by the French national agency ANR (Agence Nationale de la Recherche) under the reference ANR-19-P3IA-0001 and as part of the "Investissements d'avenir" programme, and by Djamé Seddah and Benoît Sagot's chairs in its follow-up, PRAIRIE-PSAI, funded by the ANR under the reference ANR-23-IACL-0008 as part of the "France 2030" strategy. It was also partly funded by the ANR project TraLaLaM (ANR-23-IAS1-0006) and by the BPI project Code Commons. This work was granted access to computing HPC and storage resources by GENCI at IDRIS thanks to the grants 2025-AD011016564 and 2025-AD011012254R5 on the supercomputer Jean Zay's CSL, A100, and H100 partitions and thanks to the grant GCDA1016807 on the supercomputer DALIA's B200 partition. We thank Maxence Lasbordes for the helpful discussions during the preparation of this work.

# Impact Statement

This paper presents work whose goal is to advance the field of Machine Translation with Large Language Models. There

are many potential societal consequences of our work, none which we feel must be specifically highlighted here.

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

# A. Reproducibility details

## A.1. Models, Datasets and Tools

In Table 2, we list the links to the relevant resources used for experiments.

*Table 2.* Links to datasets, benchmarks and models.

| Datasets | |
| --- | --- |
| FLORES-200 | https://huggingface.co/datasets/facebook/flores |
| *Models evaluated* | |
| GEMMA-3-270M | https://huggingface.co/google/gemma-3-270m |
| GEMMA-3-1B-PT | https://huggingface.co/google/gemma-3-1b-pt |
| GEMMA-3-4B-PT | https://huggingface.co/google/gemma-3-4b-pt |
| GEMMA-3-12B-PT | https://huggingface.co/google/gemma-3-12b-pt |
| GEMMA-3-27B-PT | https://huggingface.co/google/gemma-3-27b-pt |
| QWEN3-0.6B-BASE | https://huggingface.co/Qwen/Qwen3-0.6B-Base |
| QWEN3-1.7B-BASE | https://huggingface.co/Qwen/Qwen3-1.7B-Base |
| QWEN3-4B-BASE | https://huggingface.co/Qwen/Qwen3-4B-Base |
| LLAMA-3.2-1B | https://huggingface.co/meta-llama/Llama-3.2-1B |
| LLAMA-3.2-3B | https://huggingface.co/meta-llama/Llama-3.2-3B |
| LLAMA-2-7B | https://huggingface.co/meta-llama/Llama-2-7b |
| *Metrics* | |
| MetricX24-XXL | https://huggingface.co/google/metricx-24-hybrid-xxl-v2p6-bfloat16 |
| XCOMET-XXL | https://huggingface.co/Unbabel/XCOMET-XXL |
| BLEU | https://github.com/mjpost/sacrebleu |
| CHRF++ | https://github.com/mjpost/sacrebleu |
| FastText | https://huggingface.co/facebook/fasttext-language-identification |
| *Tools* | |
| NNsight (Fiotto-Kaufman et al., 2025) | https://nnsight.net/ |

# B. Additional Results

## B.1. Activation Patching

**Instance construction.** We construct multiple triplets consisting of one clean prompt and two corrupted prompts. Formally, we shuffle the dataset and partition it into $\lfloor \frac{N}{k+1} \rfloor$ non-overlapping groups of $k + 1$ sentence pairs. For the $j$-th group, we use the first $k$ sentences (along with their translations) as in-context demonstrations and the final sentence as the query. We then construct three prompts: a clean prompt $c_{clean}^{k,j}$ and two corrupted prompts $c_{lang}^{k,j}$ and $c_{MT}^{k,j}$.

**Results of head identification.** Following the experiments in Section 5, we provide detailed activation patching results across models and translation pairs. For GEMMA-3-12B-PT, we report results for all 20 translation directions, comprising English from and into 10 languages: Arabic, Chinese, French, Hindi, Japanese, Portuguese, Russian, Spanish, Swahili, and Wolof. For the remaining models, we report a representative subset of 5 directions: English into French, Chinese, Arabic, Russian, and Swahili. Complete results are organized by model family: Gemma-3 (Appendix B.1.1), Qwen3 (Appendix B.1.2), and LLaMA-3.2 (Appendix B.1.3).

### B.1.1. GEMMA-3

We report activation patching results for the Gemma-3 model family across four scales. For each model, we present log probability deltas under translation corruption and language corruption. GEMMA-3-270M: Figures 12 and 13; GEMMA-3-1B-PT: Figures 14 and 15; GEMMA-3-4B-PT: Figures 16 and 17; GEMMA-3-12B-PT: Figures 18 and 19; GEMMA-3-27B-PT: Figures 20 and 21. Across all translation directions, we observe that language heads and translation heads largely overlap, with the same heads being consistently solicited across different directions. Smaller models, such as GEMMA-3-270M, exhibit a larger number of highly involved heads, as well as several heads with negative deltas, particularly under translation corruption. While we cannot conclusively determine that these heads are non-functional, we attribute this behavior to the difficulty small models face in distinguishing subtle differences between clean and corrupted

prompts (in particular when the target language is the same). As model size increases, these negative deltas largely disappear, and we instead observe a sparse set of strongly positive (green) heads. For GEMMA-3-12B-PT, translation directions involving low-resource languages (English→Wolof and Wolof→English) yield positive but smaller deltas; nevertheless, the identified heads substantially overlap with those found for high-resource directions. We also observe smaller deltas when identifying language heads for directions where English is the target language, which we attribute to the English-centric nature of Gemma's training data. Finally, GEMMA-3-27B-PT exhibits trends similar to those of GEMMA-3-12B-PT.

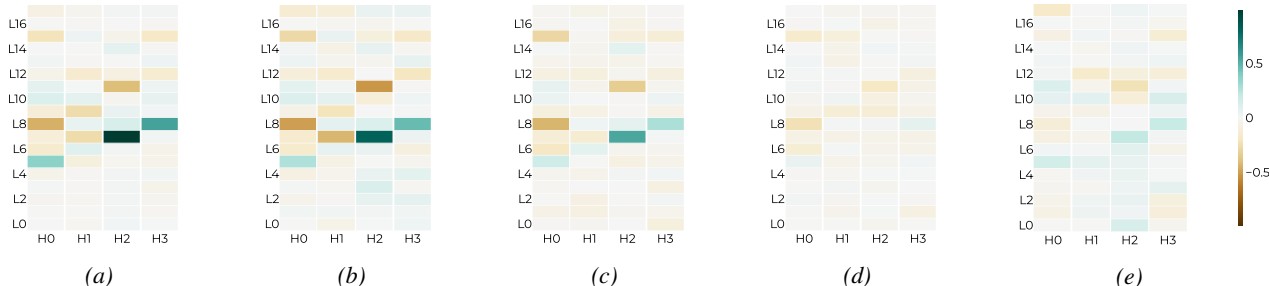

*Figure 12.* Log probability delta per layer and attention head in GEMMA-3-270M under translation corruption. (a) represent the mean delta across the 20 translations directions, while (b), (c), (d), (e) represent the delta for the translation pair English to French, Chinese, Arabic and Swahili respectively.

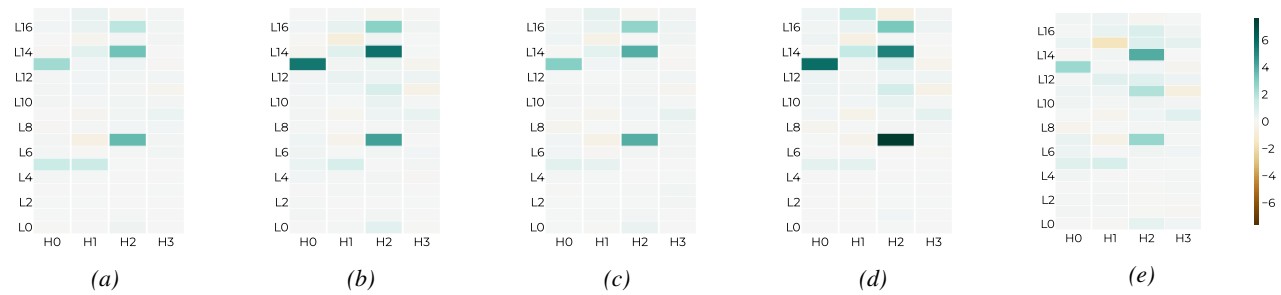

*Figure 13.* Log probability delta per layer and attention head in GEMMA-3-270M under language corruption. (a) represent the mean delta across the 20 translations directions, while (b), (c), (d), (e) represents the delta for the translation pair English to French, Chinese, Arabic and Swahili respectively.

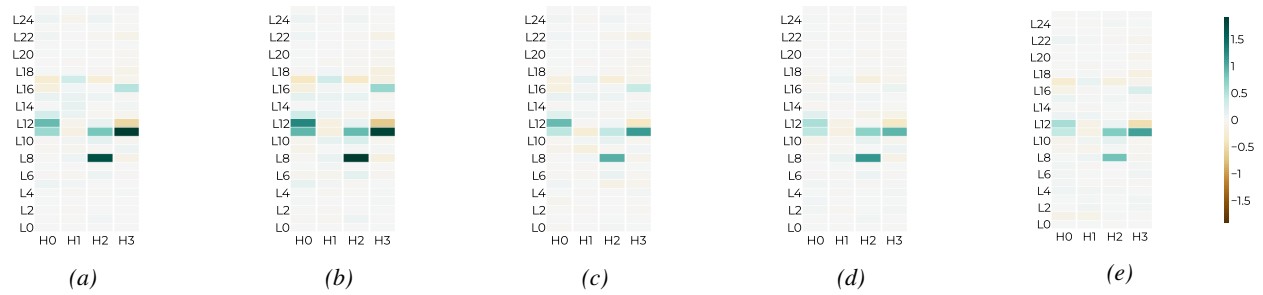

*Figure 14.* Log probability delta per layer and attention head in GEMMA-3-1B-PT under translation corruption. (a) represent the mean delta across the 20 translations directions, while (b), (c), (d), (e) represent the delta for the translation pair English to French, Chinese, Arabic and Swahili respectively.

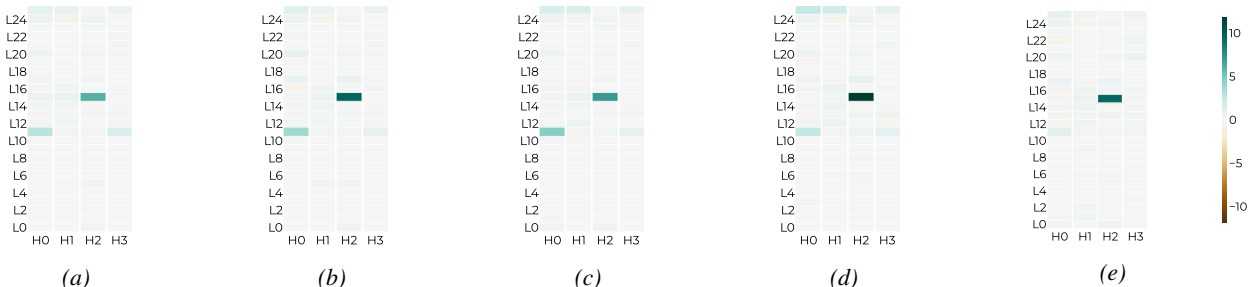

*Figure 15.* Log probability delta per layer and attention head in GEMMA-3-1B-PT under language corruption. (a) represent the mean delta across the 20 translations directions, while (b), (c), (d), (e) represents the delta for the translation pair English to French, Chinese, Arabic and Swahili respectively.

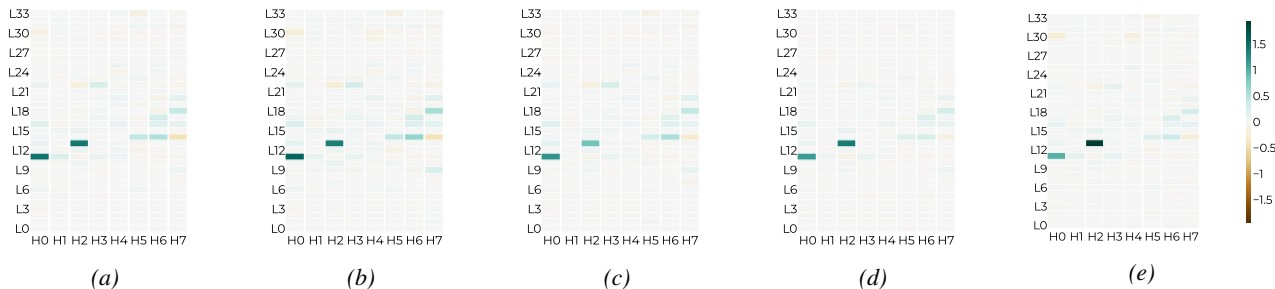

*Figure 16.* Log probability delta per layer and attention head in GEMMA-3-4B-PT under translation corruption. (a) represent the mean delta across the 20 translations directions, while (b), (c), (d), (e) represent the delta for the translation pair English to French, Chinese, Arabic and Swahili respectively.

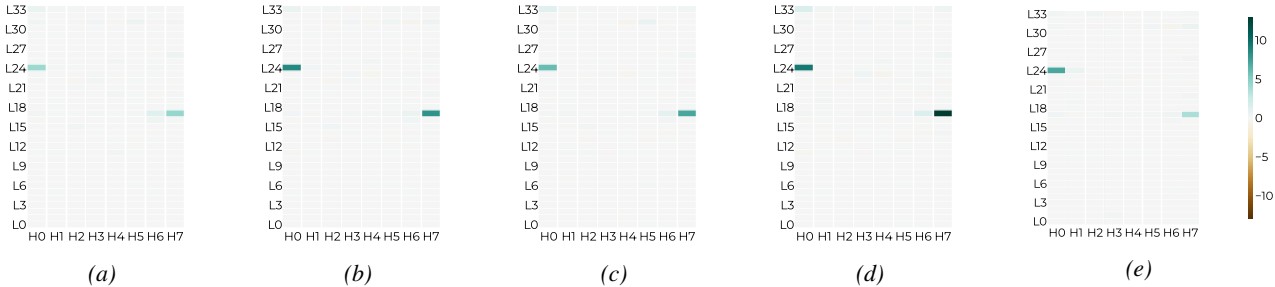

*Figure 17.* Log probability delta per layer and attention head in GEMMA-3-4B-PT under language corruption. (a) represent the mean delta across the 20 translations directions, while (b), (c), (d), (e) represents the delta for the translation pair English to French, Chinese, Arabic, Russian and Swahili respectively.

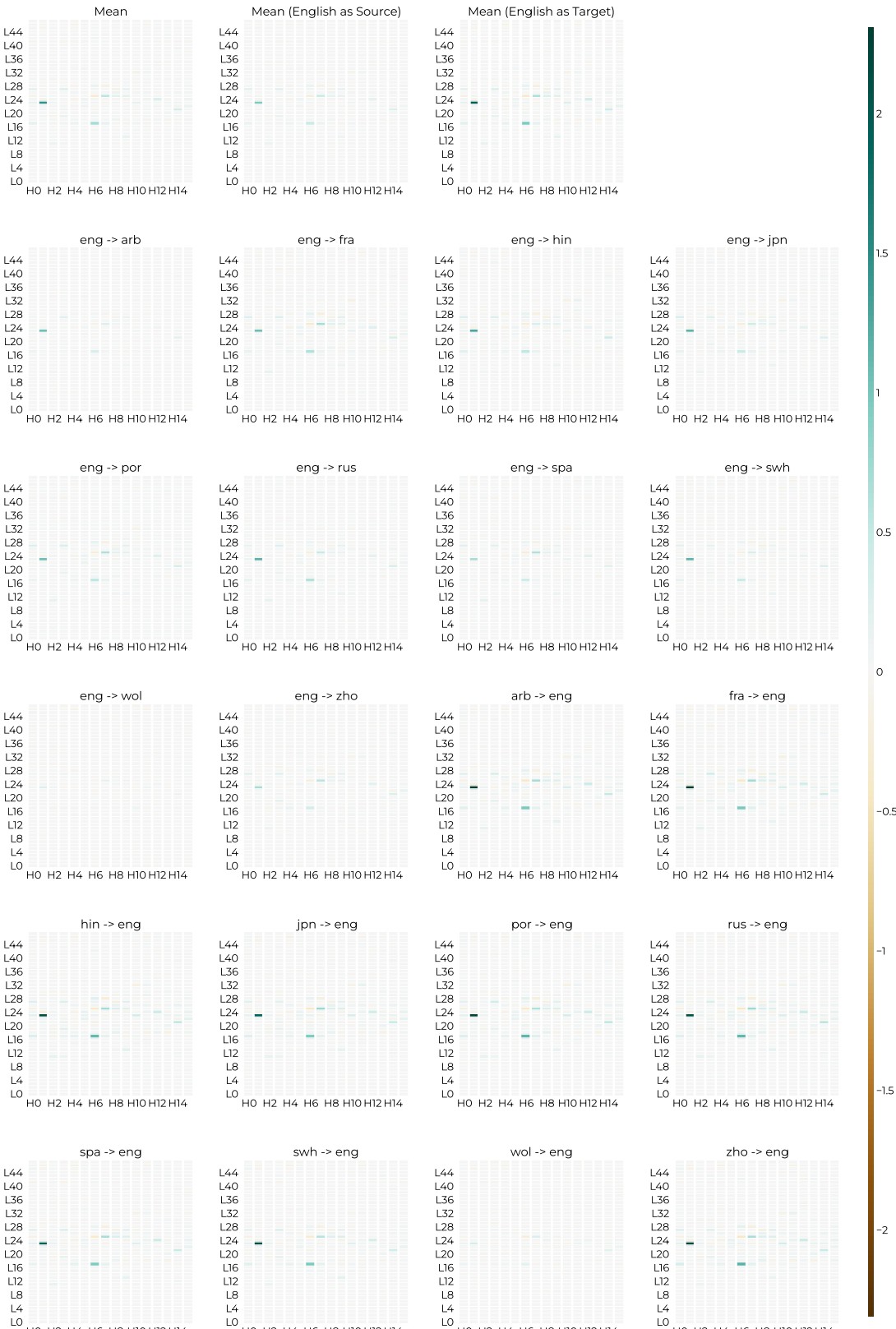

*Figure 18.* Activation Patching result under translation corruption for GEMMA-3-12B-PT

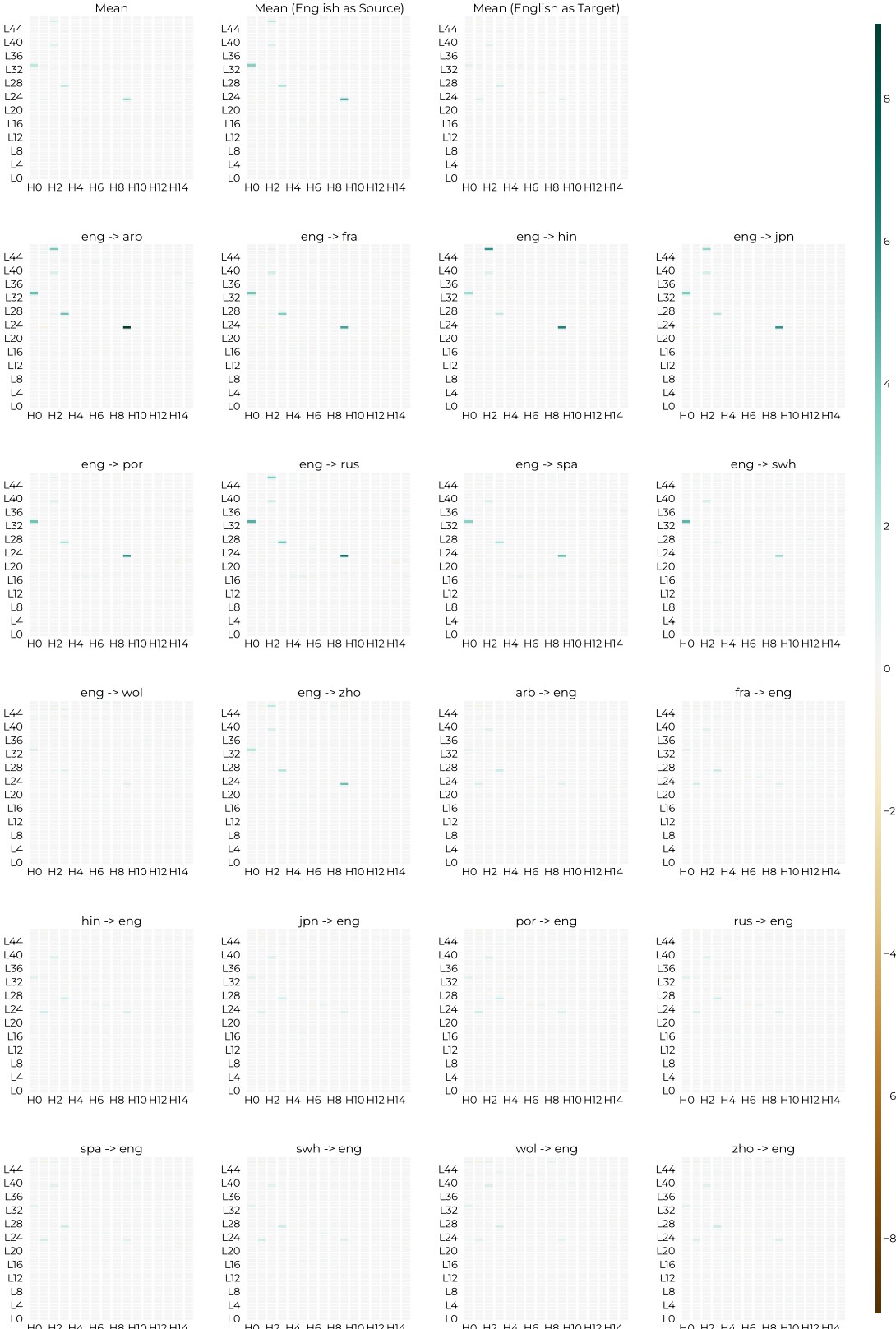

*Figure 19.* Activation Patching result under language corruption for GEMMA-3-12B-PT

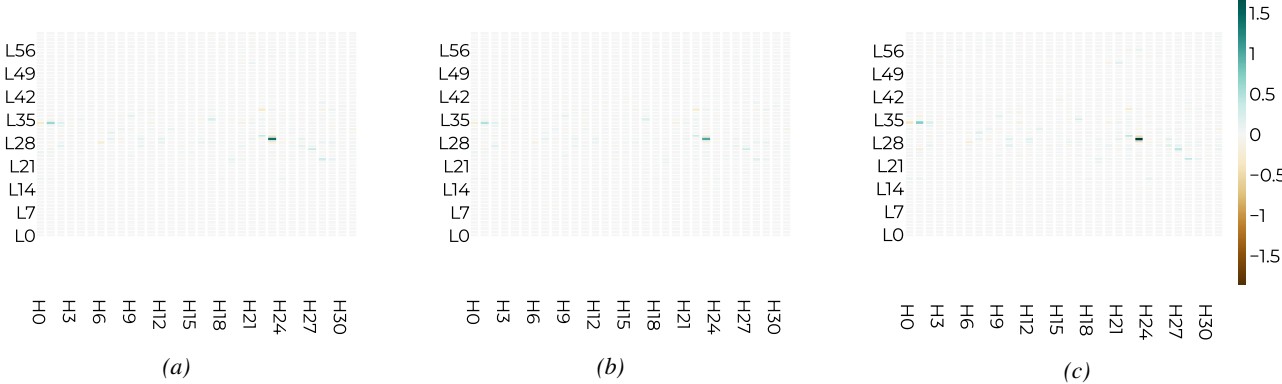

*Figure 20.* Log probability delta per layer and attention head in GEMMA-3-27B-PT under translation corruption. (a) represent the mean delta across the translation direction English→French and French→English. While (b) and (c) represent the delta for the translation pair English to French and French to English, respectively.

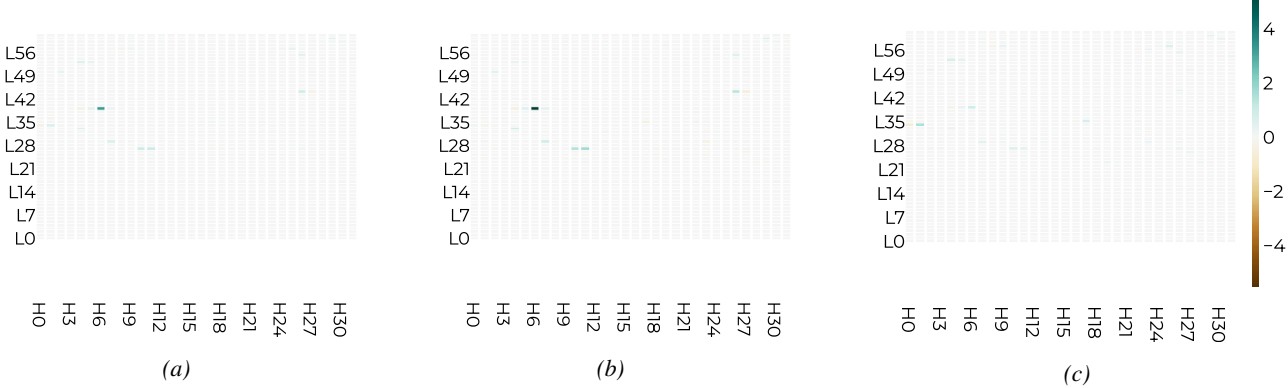

*Figure 21.* Log probability delta per layer and attention head in GEMMA-3-27B-PT under language corruption. (a) represent the mean delta across the translation direction English→French and French→English. While (b) and (c) represent the delta for the translation pair English to French and French to English, respectively.

B.1.2. QWEN-3

We report activation patching results for the Qwen3 model family across three scales. For each model, we present log probability deltas under translation corruption and language corruption. QWEN3-0.6B-BASE: Figures 22 and 23; QWEN3-1.7B-BASE: Figures 24 and 25; QWEN3-4B-BASE: Figures 26 and 27. Similar to GEMMA-3-270M-PT, QWEN3-0.6B-BASE exhibits negative deltas under translation corruption, which gradually disappear as model size increases. Within the Qwen family, we consistently observe a single head that stands out for language identification across French, Chinese, and Arabic. However, since Qwen models struggle to generate Swahili, Figures 22e, 23e, and 24e show that almost no head clearly stands out for this direction. This effect is particularly pronounced under translation corruption, where the model may fail to recognize that the corrupted prompt contains incorrect source–translation pairings.

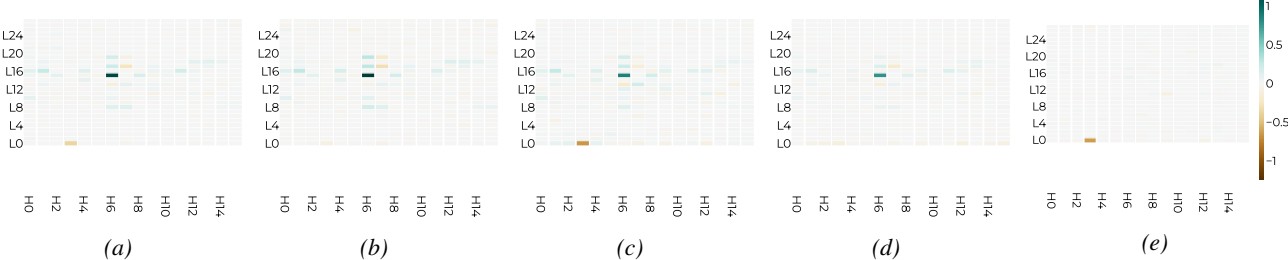

*Figure 22.* Log probability delta per layer and attention head in QWEN3-0.6B-BASE under translation corruption. (a) represent the mean delta across the 20 translations directions, while (b), (c), (d), (e) represent the delta for the translation pair English to French, Chinese, Arabic and Swahili respectively.

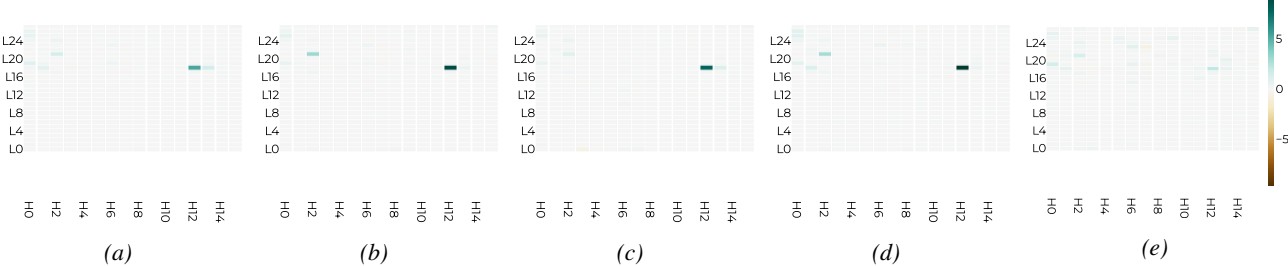

*Figure 23.* Log probability delta per layer and attention head in QWEN3-0.6B-BASE under language corruption. (a) represent the mean delta across the 20 translations directions, while (b), (c), (d), (e) represents the delta for the translation pair English to French, Chinese, Arabic and Swahili respectively.

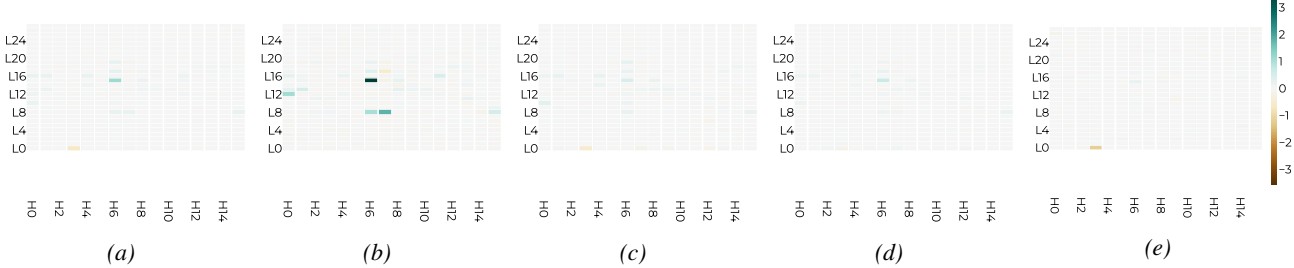

*Figure 24.* Log probability delta per layer and attention head in QWEN3-1.7B-BASE under translation corruption. (a) represent the mean delta across the 20 translations directions, while (b), (c), (d), (e) represent the delta for the translation pair English to French, Chinese, Arabic and Swahili respectively.

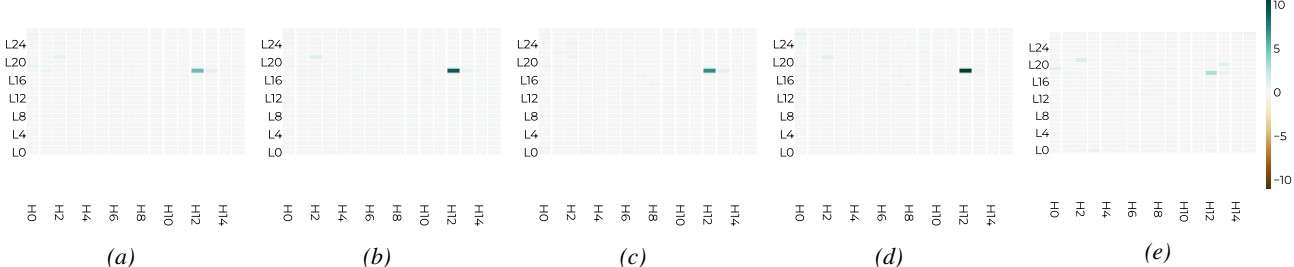

*Figure 25.* Log probability delta per layer and attention head in QWEN3-1.7B-BASE under language corruption. (a) represent the mean delta across the 20 translations directions, while (b), (c), (d), (e) represents the delta for the translation pair English to French, Chinese, Arabic and Swahili respectively.

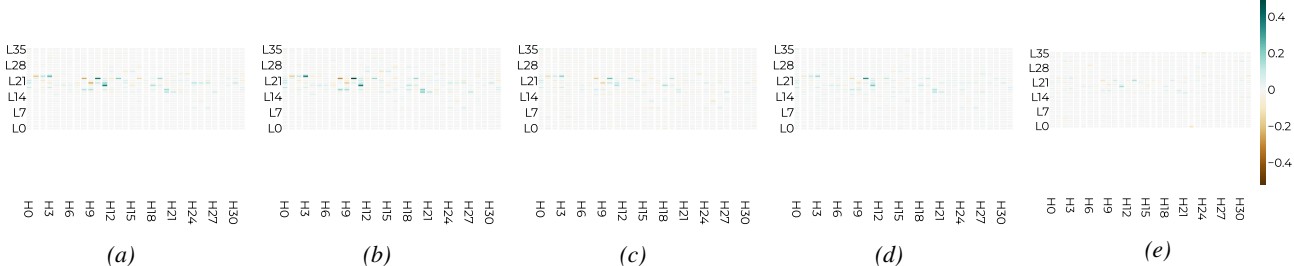

*Figure 26.* Log probability delta per layer and attention head in QWEN3-4B-BASE under translation corruption. (a) represent the mean delta across the 20 translations directions, while (b), (c), (d),(e) represent the delta for the translation pair English to French, Chinese, Arabic and Swahili respectively.

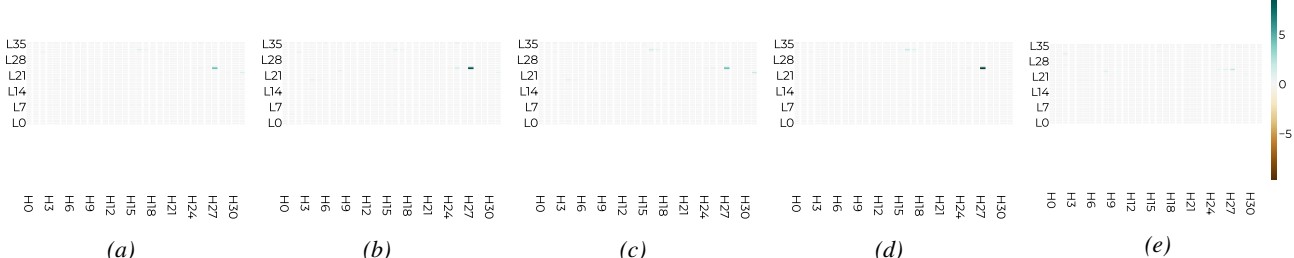

*Figure 27.* Log probability delta per layer and attention head in QWEN3-4B-BASE under language corruption. (a) represent the mean delta across the 20 translations directions, while (b), (c), (d), (e) represents the delta for the translation pair English to French, Chinese, Arabic and Swahili respectively.

### B.1.3. LLAMA-3.2

We report activation patching results for the Llama-3.2 model family across two scales. For each model, we present log probability deltas under translation corruption and language corruption. LLAMA-3.2-1B: Figures 28 and 29; LLAMA-3.2-3B: Figures 30 and 31. Here, we observe the same behavior as in the QWEN3 models.

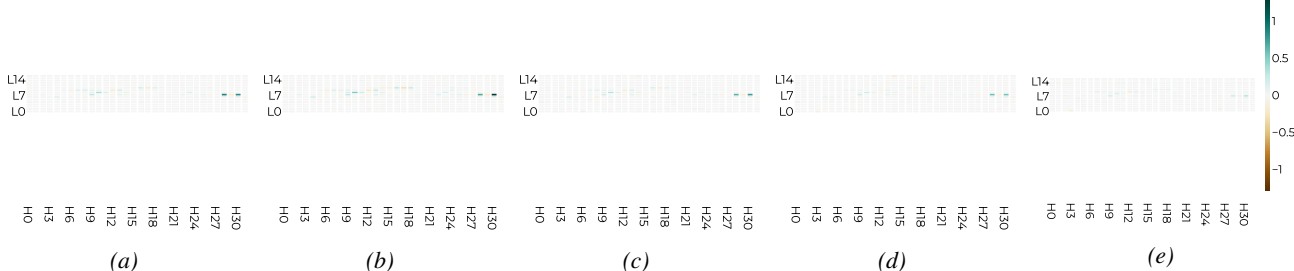

*Figure 28.* Log probability delta per layer and attention head in LLAMA-3.2-1B under translation corruption. (a) represent the mean delta across the 20 translations directions, while (b), (c), (d), (e) represent the delta for the translation pair English to French, Chinese, Arabic and Swahili respectively.

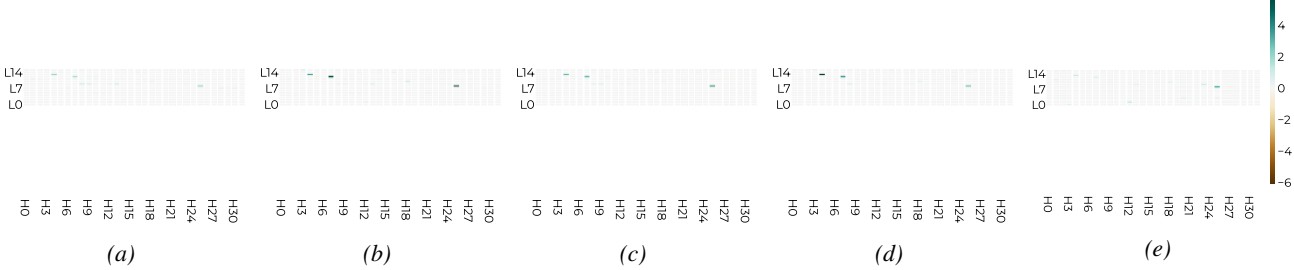

*Figure 29.* Log probability delta per layer and attention head in LLAMA-3.2-1B under language corruption. (a) represent the mean delta across the 20 translations directions, while (b), (c), (d), (e) represents the delta for the translation pair English to French, Chinese, Arabic and Swahili respectively.

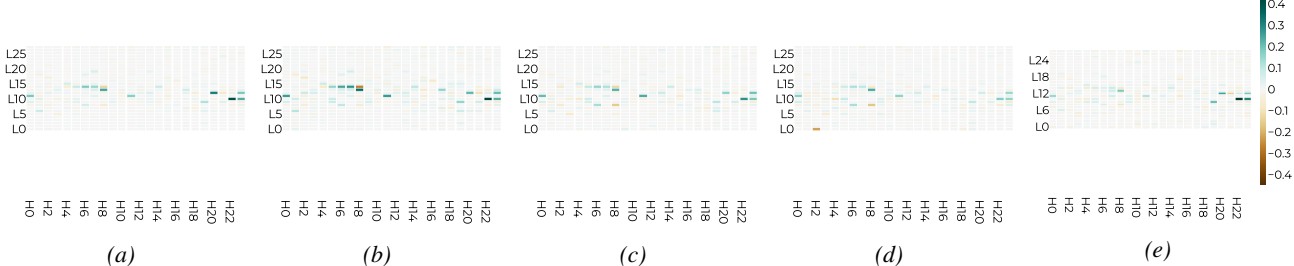

*Figure 30.* Log probability delta per layer and attention head in LLAMA-3.2-3B under translation corruption. (a) represent the mean delta across the 20 translations directions, while (b), (c), (d), (e) represent the delta for the translation pair English to French, Chinese, Arabic and Swahili respectively.

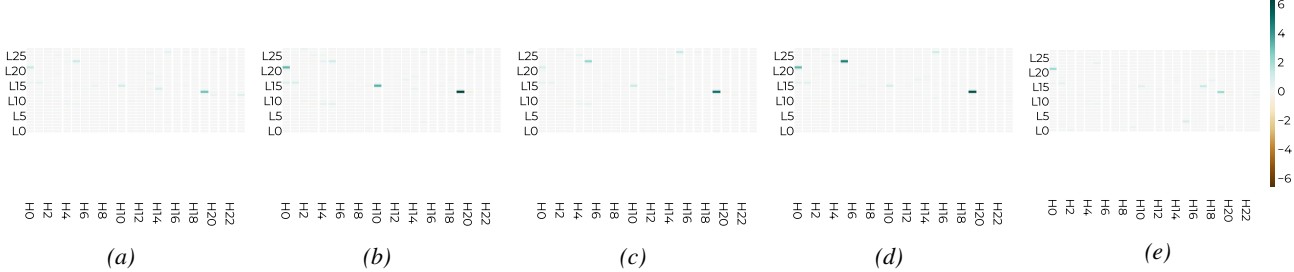

*Figure 31.* Log probability delta per layer and attention head in LLAMA-3.2-3B under language corruption. (a) represent the mean delta across the 20 translations directions, while (b), (c), (d), (e) represents the delta for the translation pair English to French, Chinese, Arabic and Swahili respectively.

## B.2. Jaccard Similarity between language heads and translation heads

To quantify the degree of overlap between the two sets of heads identified through activation patching (Section 3.1), we compute the Jaccard index between the top 5% of language heads and the top 5% of translation heads for all studied translation pairs. We report results for the Gemma-3 (Appendix B.2.1), Qwen-3 (Appendix B.2.2), and Llama-3.2 (Appendix B.2.3) model families.

Across all models and translation directions, the Jaccard indices remain consistently low, typically below 0.2, indicating that language heads and translation heads constitute mostly disjoint sets. These findings provide quantitative support for our proposed decomposition of MT into distinct subtasks mediated by separate attention heads.

### B.2.1. GEMMA-3

We report the Jaccard index for the Gemma-3 model family across four scales: GEMMA-3-270M (Figure 32), GEMMA-3-1B-PT (Figure 33), GEMMA-3-4B-PT (Figure 34), and GEMMA-3-12B-PT (Figure 35).

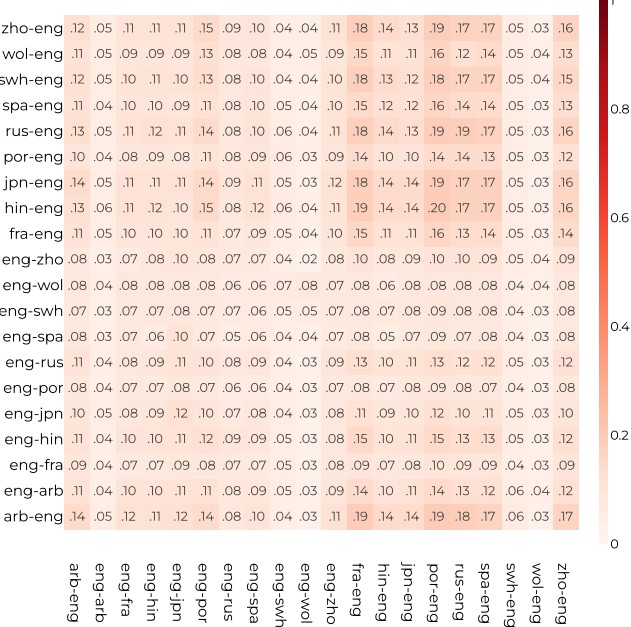

*Figure 32.* Jaccard index between the top 5% of language heads (y-axis) and the top 5% of translation heads (x-axis) identified through activation patching in GEMMA-3-270M.

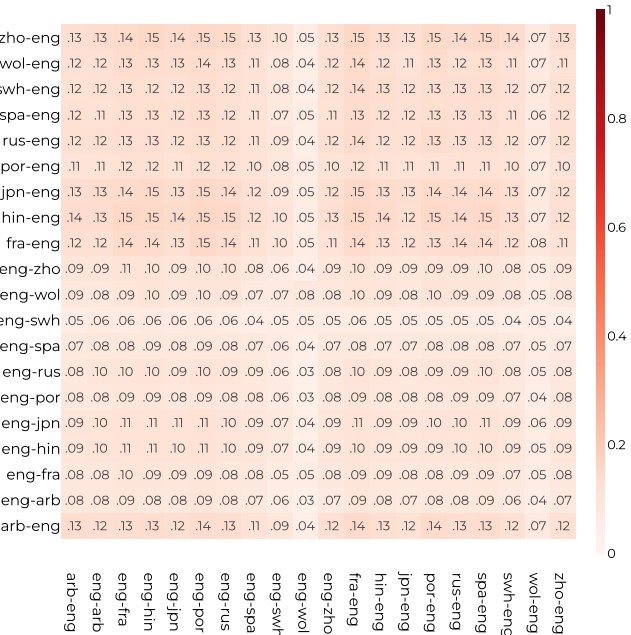

*Figure 33.* Jaccard index between the top 5% of language heads (y-axis) and the top 5% of translation heads (x-axis) identified through activation patching in GEMMA-3-1B.

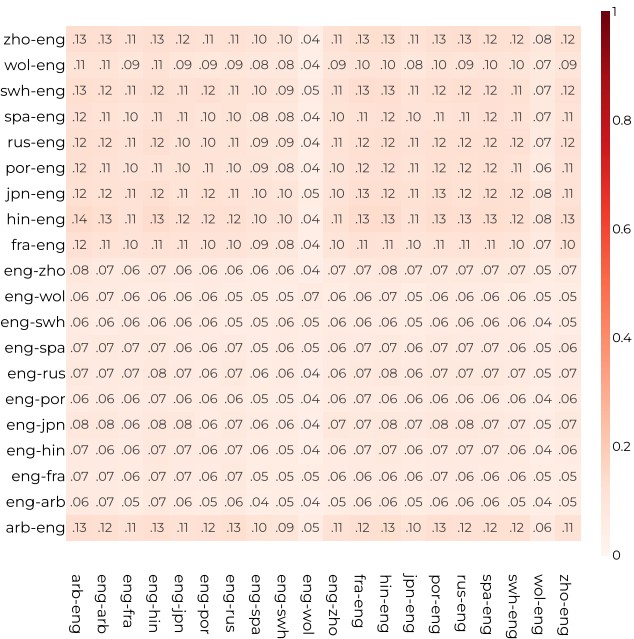

*Figure 34.* Jaccard index between the top 5% of language heads (y-axis) and the top 5% of translation heads (x-axis) identified through activation patching in GEMMA-3-4B-PT.

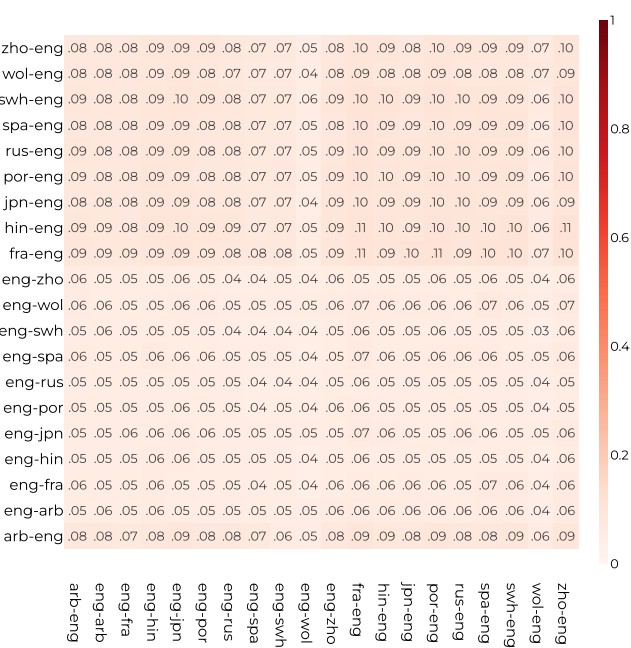

*Figure 35.* Jaccard index between the top 5% of language heads (y-axis) and the top 5% of translation heads (x-axis) identified through activation patching in GEMMA-3-12B-PT.

### B.2.2. QWEN-3

We report the Jaccard index for the Qwen-3 model family across three scales: QWEN3-0.6B-BASE (Figure 36), QWEN3-1.7B-BASE (Figure 37), and QWEN3-4B-BASE (Figure 38).

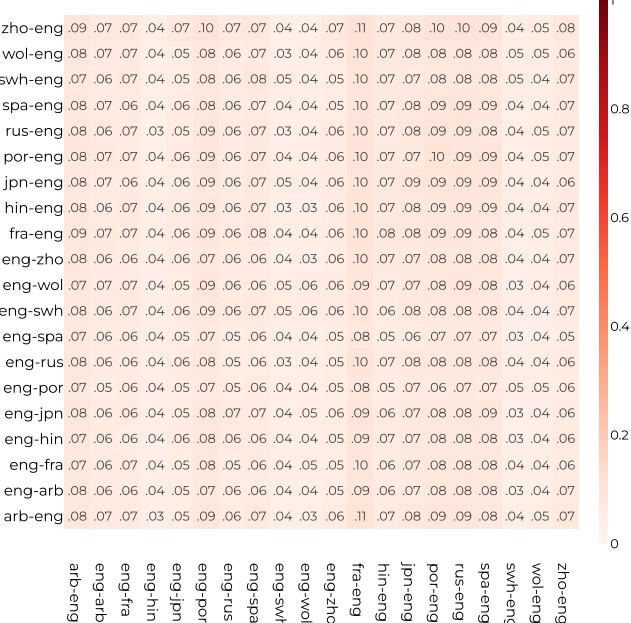

*Figure 36.* Jaccard index between the top 5% of language heads (y-axis) and the top 5% of translation heads (x-axis) identified through activation patching in QWEN3-0.6B-BASE.

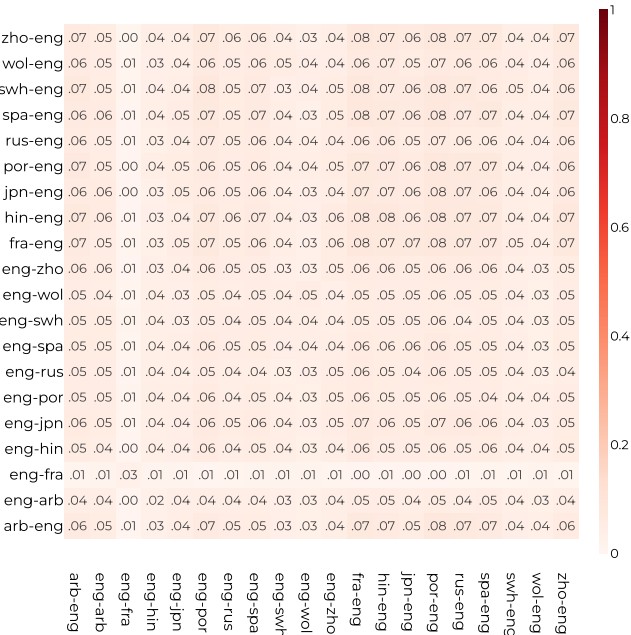

*Figure 37.* Jaccard index between the top 5% of language heads (y-axis) and the top 5% of translation heads (x-axis) identified through activation patching in QWEN3-1.7B-BASE.

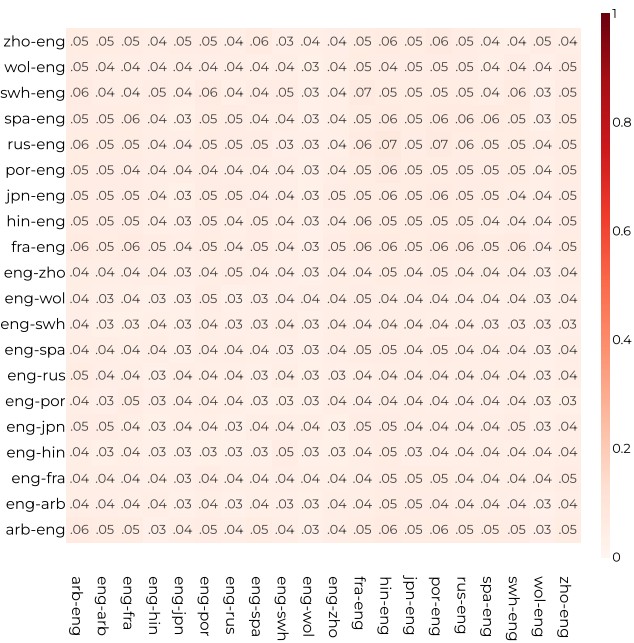

*Figure 38.* Jaccard index between the top 5% of language heads (y-axis) and the top 5% of translation heads (x-axis) identified through activation patching in QWEN3-4B-BASE. We report

B.2.3. LLAMA-3.2

We report the Jaccard index for the Llama-3.2 model family across two scales: LLAMA-3.2-1B (Figure 39) and LLAMA-3.2-3B (Figure 40).

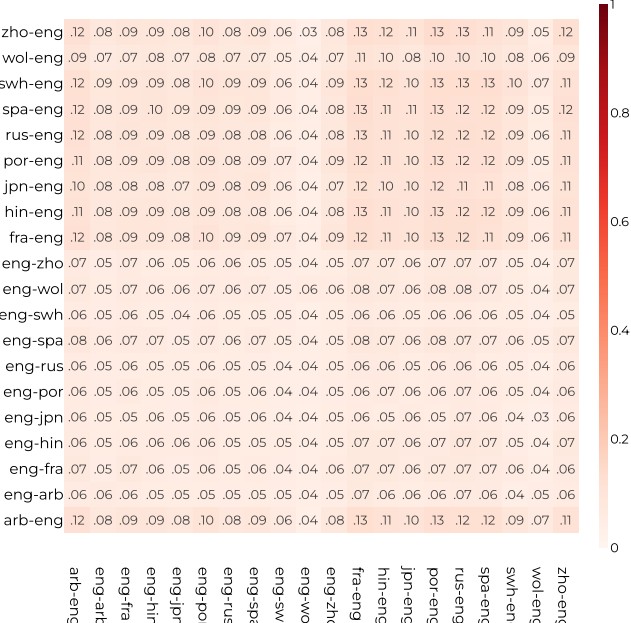

*Figure 39.* Jaccard index between the top 5% of language heads (y-axis) and the top 5% of translation heads (x-axis) identified through activation patching in LLAMA-3.2-1B.

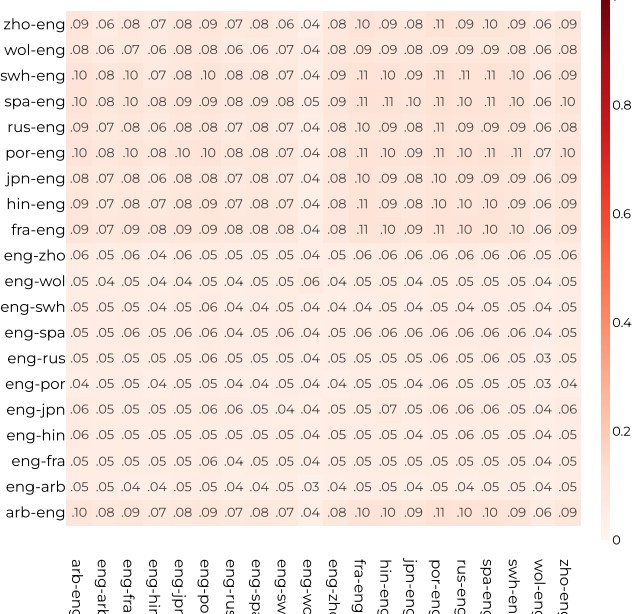

*Figure 40.* Jaccard index between the top 5% of language heads (y-axis) and the top 5% of translation heads (x-axis)identified through activation patching in LLAMA-3.2-3B.

## B.3. Steering

Following the experiments in Section 5, we provide detailed steering results across models and translation pairs. We first present a qualitative analysis of failure modes when steering with only language heads or only translation heads (Appendix B.3.1), illustrating the distinct functional roles of each head type. We then report quantitative results using five metrics: BLEU, chrF++, MetricX-24, MetricX-24 QE, and XCOMET. For GEMMA-3-12B-PT, we report results for all 20 translation directions, comprising English from and into 10 languages: Arabic, Chinese, French, Hindi, Japanese, Portuguese, Russian, Spanish, Swahili, and Wolof. For the remaining models, we report average scores aggregated over translation pairs with English as the source language and English as the target language, respectively. Complete quantitative results are organized by model family: GEMMA-3 (Appendix B.3.2), QWEN3 (Appendix B.3.3), and LLAMA-3.2 (Appendix B.3.4).

### B.3.1. QUALITATIVE ANALYSIS

To complement our quantitative evaluation, we provide illustrative examples of failure modes when steering with only language heads or only translation heads. Table 3 presents representative outputs from GEMMA-3-12B-PT on English→French translation. These examples highlight the functional separation between head types: steering exclusively with translation heads preserves semantic equivalence but fails to maintain target language identity, while steering exclusively with language heads produces text in the correct language but without translating the source content.

*Table 3.* Illustrative failure modes when steering GEMMA-3-12B-PT with only 1% of translation heads or 1% of language heads for English→French translation. When steering exclusively with translation heads, the model produces semantically adequate translations but frequently generates in an incorrect target language (e.g., Chinese or Spanish), indicating that sentence equivalence is preserved while target language identity is lost. Conversely, when steering exclusively with language heads, the model consistently outputs French text but often fails to produce a translation of the query, instead generating unrelated content (e.g., "Vrai"). These complementary failure patterns provide further evidence that translation heads and language heads encode functionally distinct components of the MT task.

| Query | Reference | Steering with translation heads | Steering with language heads |
| --- | --- | --- | --- |
| Soon after the outbreak of hostilities, Britain initiated a naval blockade of Germany. | Peu après le début des hostilités, la Grande-Bretagne a mis en place un blocus naval contre l'Allemagne. | Peu après le début des hostilités, la Grande-Bretagne a mis en place un blocus naval de l'Allemagne. | Vrai |
| "We will endeavour to cut carbon dioxide emissions per unit of GDP by a notable margin by 2020 from the 2005 level," Hu said. | « Nous nous efforcerons de réduire les émissions de dioxyde de carbone par unité de PIB de manière significative d'ici 2020 par rapport au niveau de 2005 », a déclaré M. Hu. | "我们将在2020年前努力使单位国内生产总值二氧化碳排放量比2005年水平大幅下降。"胡说。 | "Nous nous efforcerons de réduire les émissions de dioxyde de carbone par unité de PIB d'un taux notable d'ici 2020 par rapport au niveau de 2005", a déclaré Hu. |
| He did not set a figure for the cuts, saying they will be made based on China's economic output. | Il n'a pas chiffré ces réductions, disant qu'elles seront basées sur la production économique chinoise. | 他没有设定削减的数字，他说削减将根据中国的经济产出进行。 | Il n'a pas fixé de chiffre pour les coupes, disant qu'elles seront faites en fonction de la production économique de la Chine. |
| The outbreak has prompted the Indian government to undertake such measures as deployment of pig catchers in seriously affected areas, distributing thousands of mosquito curtains and spraying pesticides. | L'épidémie a incité le gouvernement indien à prendre des mesures telles que le déploiement de collecteurs de porcs dans les zones gravement touchées, la distribution de milliers de moustiquaires et la pulvérisation de pesticides. | La epidemia ha llevado al gobierno indio a tomar medidas como el despliegue de cazadores de cerdos en las zonas más afectadas, la distribución de miles de mosquiteras y la pulverización de pesticidas. | L'épidémie a incité le gouvernement indien à prendre des mesures telles que le déploiement de chasseurs de porcs dans les zones les plus touchées, la distribution de milliers de moustiquaires et la pulvérisation de pesticides. |
| Before The Simpsons Simon had worked on several shows in various positions. | Avant les Simpsons, Simon avait travaillé sur plusieurs séries, occupant divers postes. | Avant The Simpsons Simon avait travaillé sur plusieurs émissions dans diverses positions. | Vrai |

### B.3.2. GEMMA-3

We report steering results for the Gemma-3 model family across four scales. For GEMMA-3-12B-PT, we present results for all 20 translation directions. For the smaller models, we report average scores aggregated over translation pairs with English as the source language and English as the target language: GEMMA-3-270M (Figure 41), GEMMA-3-1B-PT (Figure 42), GEMMA-3-4B-PT (Figure 43) and GEMMA-3-12B-PT (Figure 44-48). Steering GEMMA-3-270M substantially outperforms the instructed zero-shot baseline across all metrics. The baseline performs poorly because the model is too small to reliably follow instructions, and its zero-shot multilingual capabilities remain limited. Larger Gemma models (e.g., GEMMA-3-1B-PT and GEMMA-3-4B-PT) do not suffer from this baseline issue; nevertheless, our steering approach achieves performance competitive with the baseline, particularly when translating into English. This aligns with the well-known observation that translating into English is easier than translating out of it, which also facilitates performance recovery through steering in this direction. For GEMMA-3-12B-PT, steering only 1% of language heads and 1% of translation heads matches or outperforms the instructed baseline. With this proportion of heads, we obtain an average English→X performance of 34.9 BLEU and 4.7 MetricX, compared to 37.51 BLEU and 4.06 MetricX for standard 5-shot prompting with GEMMA-3-12B-PT (Table 5). The gap of only 2.6 BLEU and 0.6 MetricX points highlights the expressiveness and effectiveness of our task vectors.

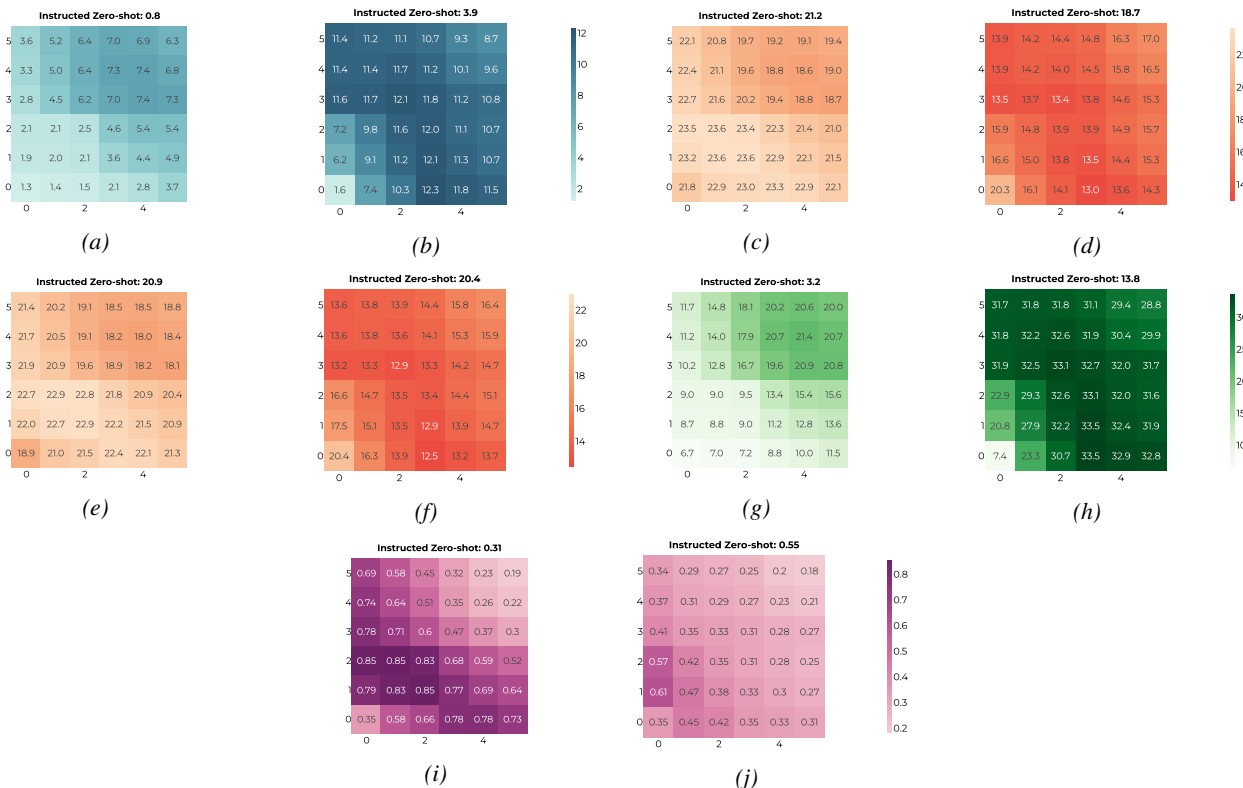

*Figure 41.* Steering-induced MT performance for GEMMA-3-270M under an instruction-free zero-shot setup. We report BLEU (a, b), MetricX-24 (c, d), MetricX-24 QE (e, f), chrF++ (g, h) and XCOMET (i, j) when steering $n\%$ of translation heads (x-axis) and $m\%$ of language heads (y-axis). Subfigures (a, c, e, g, i) report results for translation pairs with English as the source language, while (b, d, f, h, j) report results for translation pairs with English as the target language.

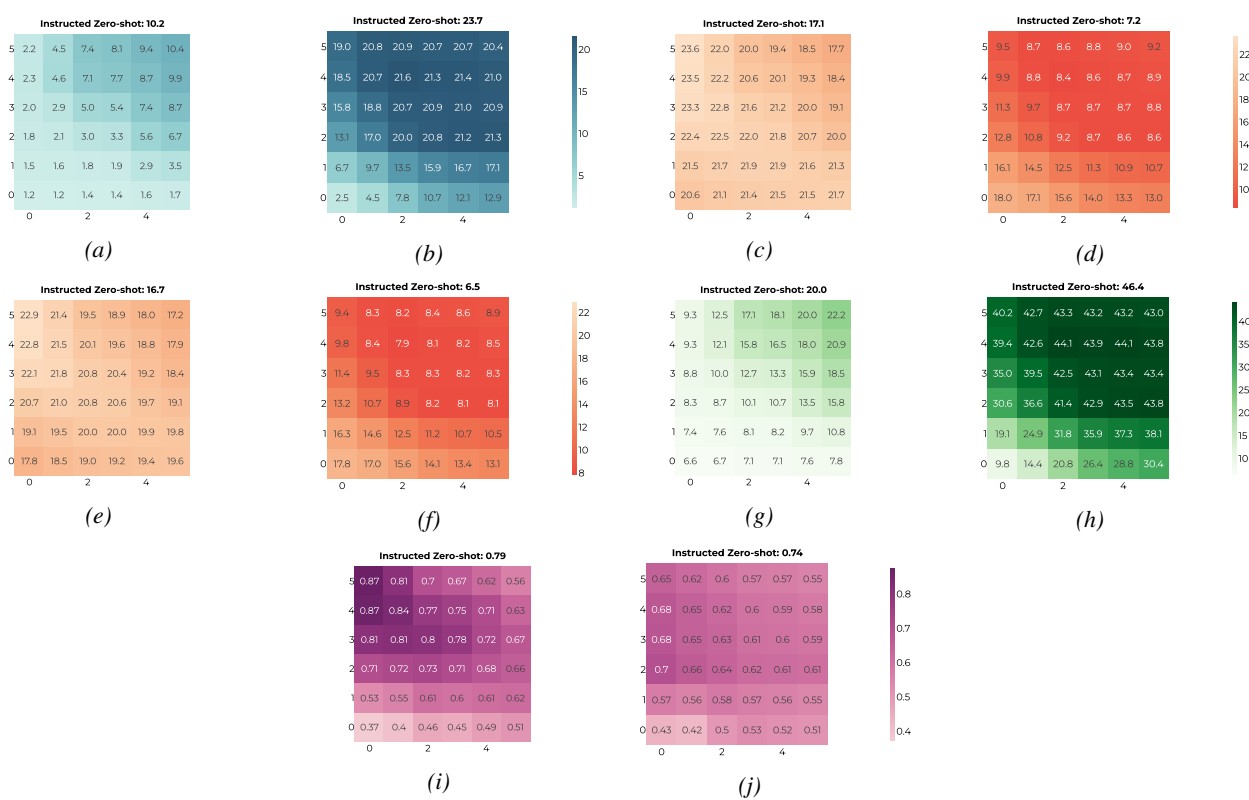

*Figure 42.* Steering-induced MT performance for GEMMA-3-1B-PT under an instruction-free zero-shot setup. We report BLEU (a, b), MetricX-24 (c, d), MetricX-24 QE (e, f), chrF++ (g, h) and XCOMET (i, j) when steering $n\%$ of translation heads (x-axis) and $m\%$ of language heads (y-axis). Subfigures (a, c, e, g, i) report results for translation pairs with English as the source language, while (b, d, f, h, j) report results for translation pairs with English as the target language.

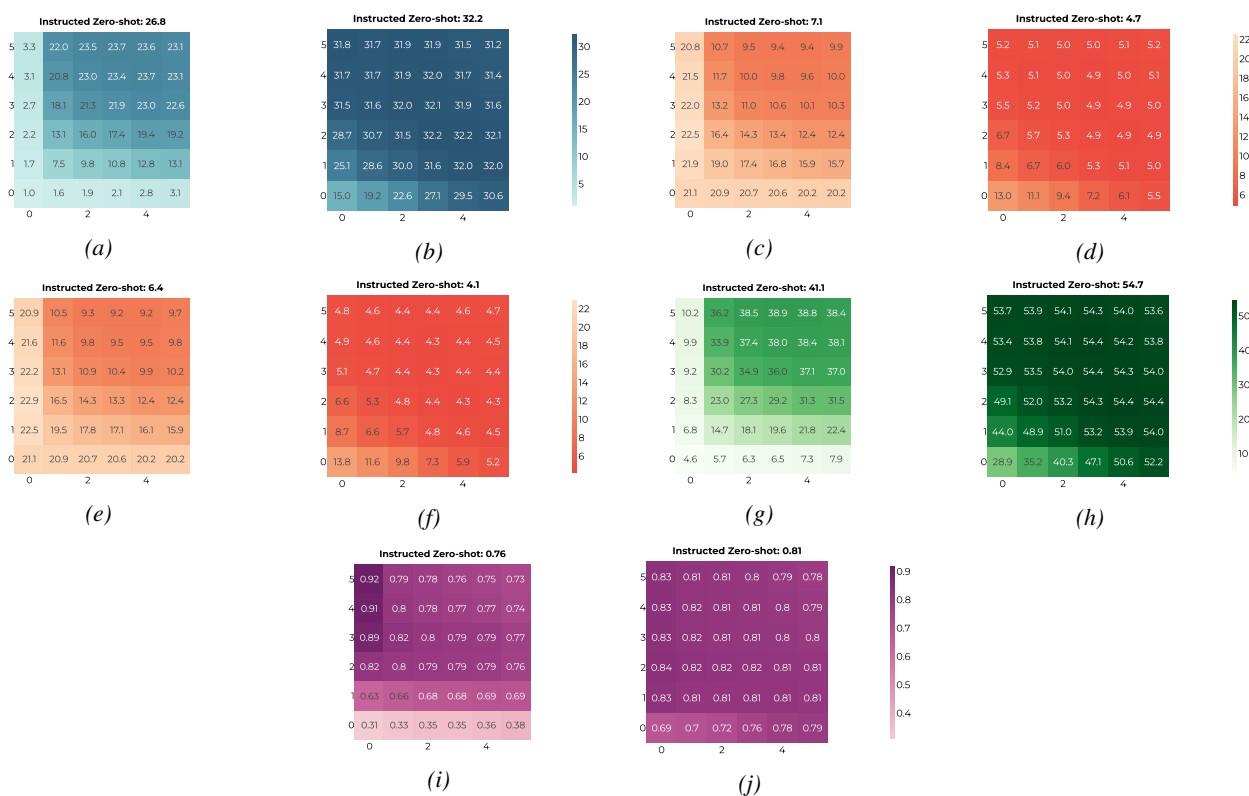

*Figure 43.* Steering-induced MT performance for GEMMA-3-4B-PT under an instruction-free zero-shot setup. We report BLEU (a, b), MetricX-24 (c, d), MetricX-24 QE (e, f), chrF++ (g, h) and XCOMET (i, j) when steering $n\%$ of translation heads (x-axis) and $m\%$ of language heads (y-axis). Subfigures (a, c, e, g, i) report results for translation pairs with English as the source language, while (b, d, f, h, j) report results for translation pairs with English as the target language.

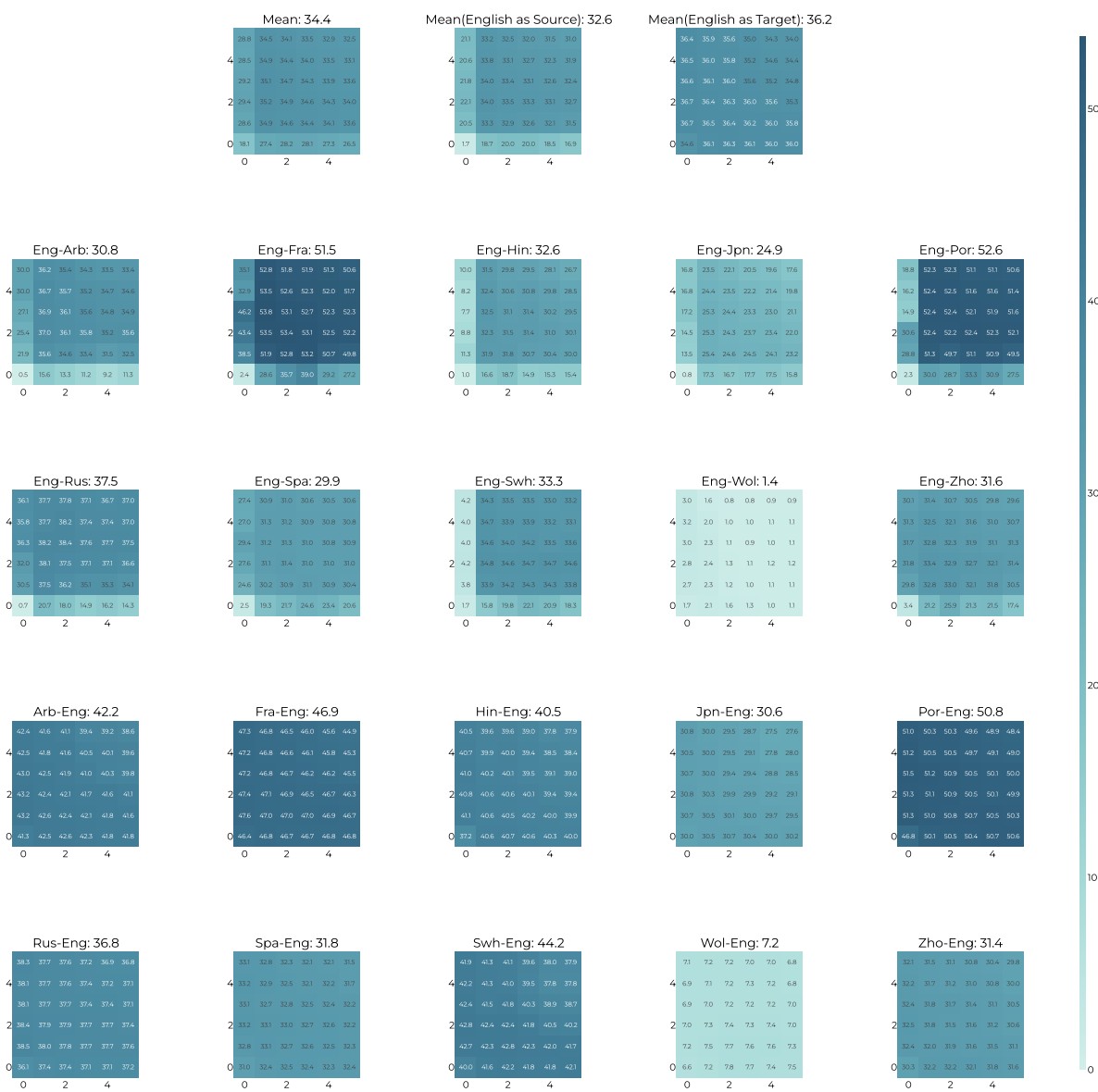

*Figure 44.* Steering-induced MT performance (BLEU) for GEMMA-3-12B-PT under an instruction-free zero-shot setup across all 20 translation directions. Each heatmap corresponds to a specific translation pair, with the x-axis and y-axis denoting the proportion of translation heads and language heads steered, respectively. The score reported next to each translation pair indicates the performance obtained in an instructed zero-shot setup, serving as a topline for comparison.

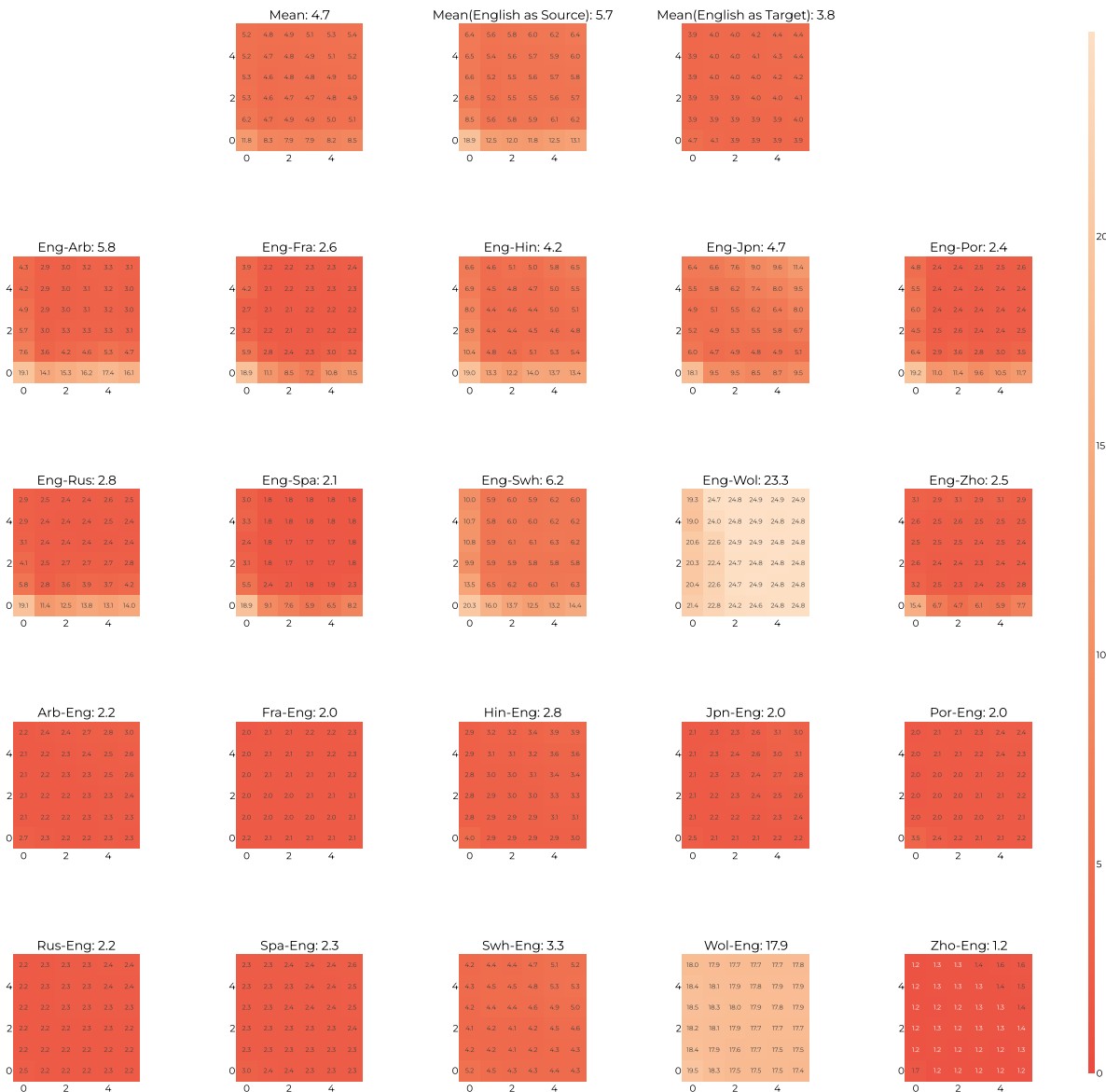

*Figure 45.* Steering-induced MT performance (MetricX-24) for GEMMA-3-12B-PT under an instruction-free zero-shot setup across all 20 translation directions. Each heatmap corresponds to a specific translation pair, with the x-axis and y-axis denoting the proportion of translation heads and language heads steered, respectively. The score reported next to each translation pair indicates the performance obtained in an instructed zero-shot setup, serving as a topline for comparison.

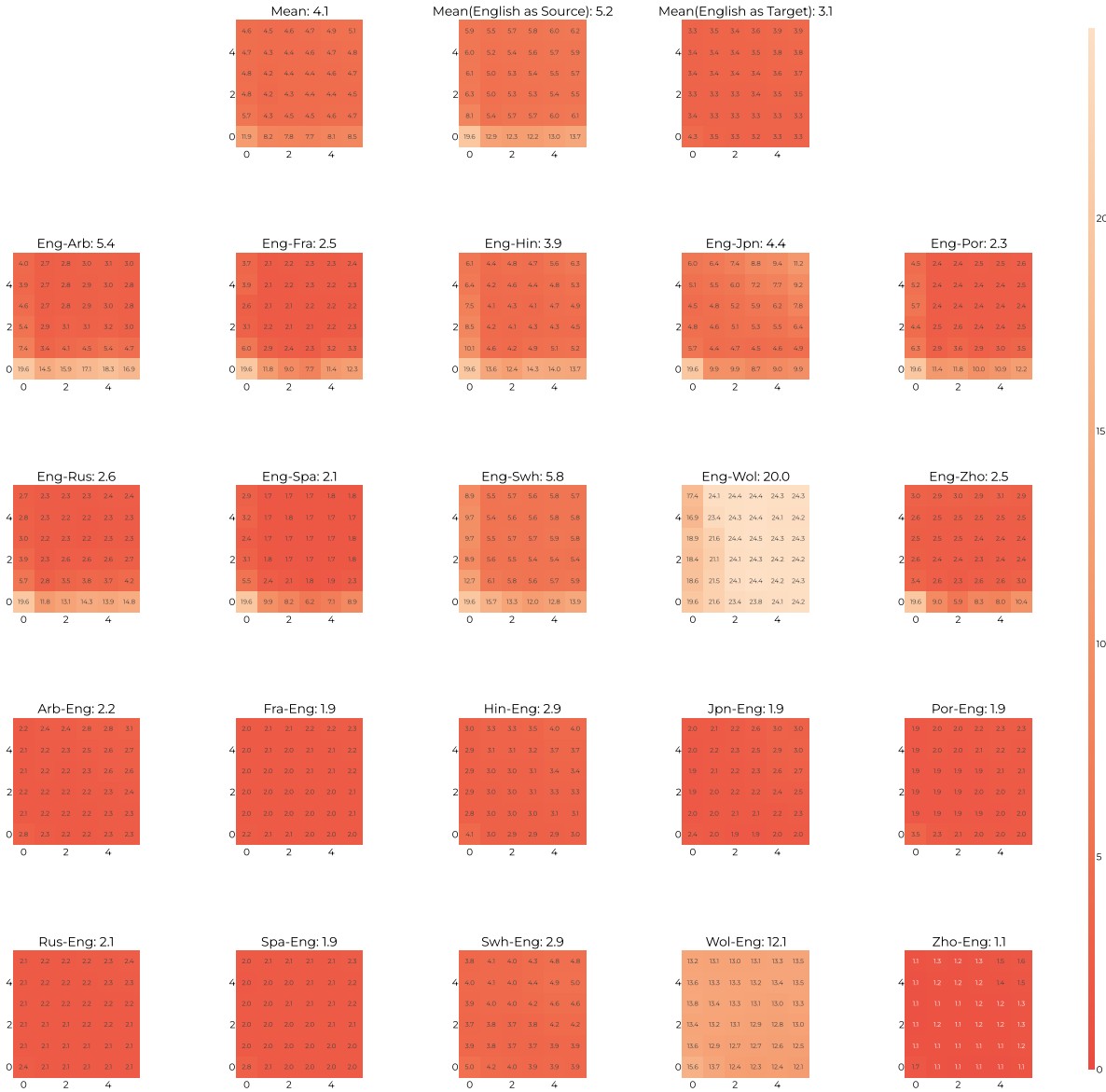

*Figure 46.* Steering-induced MT performance (MetricX-24 QE) for GEMMA-3-12B-PT under an instruction-free zero-shot setup across all 20 translation directions. Each heatmap corresponds to a specific translation pair, with the x-axis and y-axis denoting the proportion of translation heads and language heads steered, respectively. The score reported next to each translation pair indicates the performance obtained in an instructed zero-shot setup, serving as a topline for comparison.

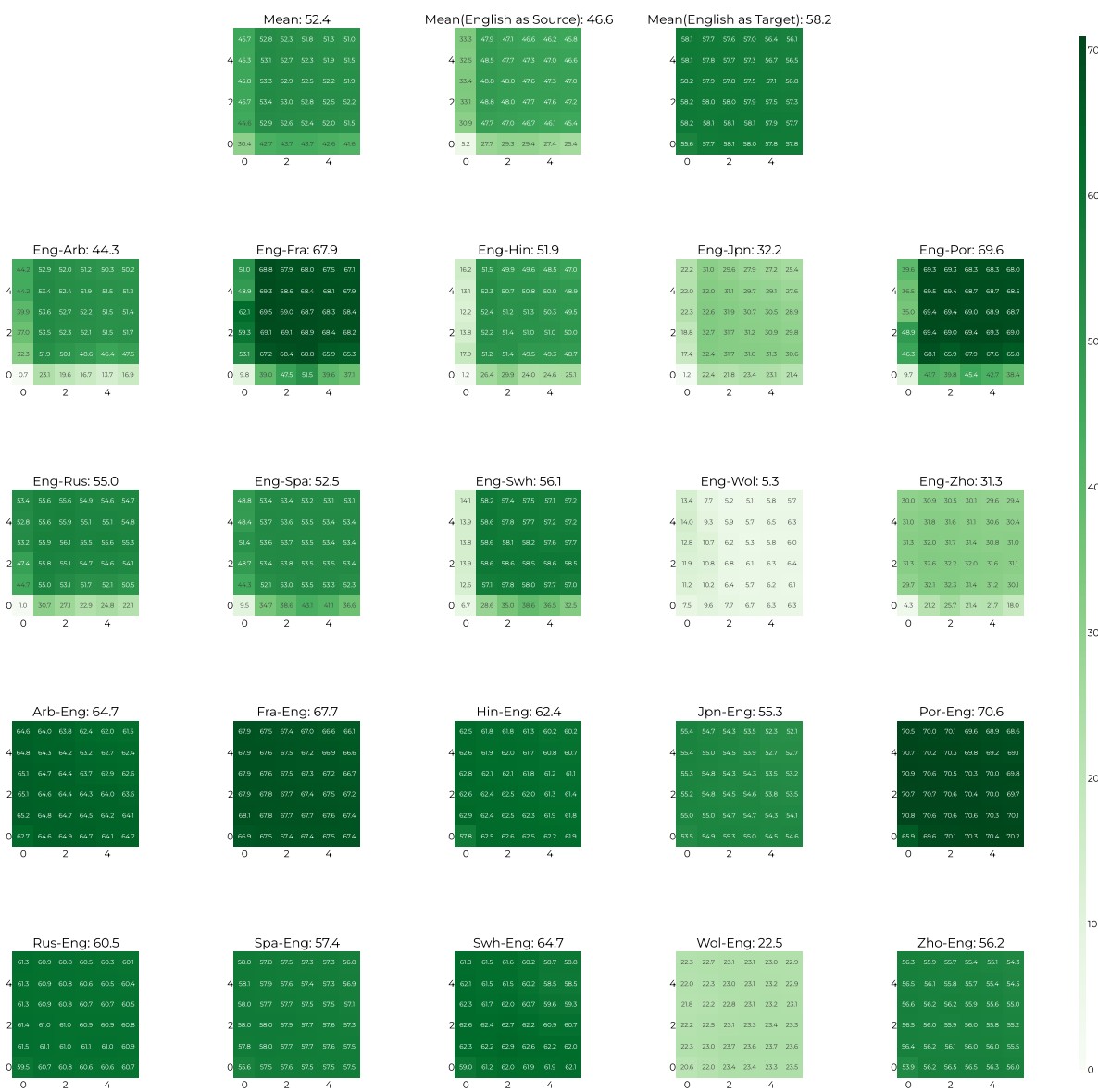

*Figure 47.* Steering-induced MT performance (chrF++) for GEMMA-3-12B-PT under an instruction-free zero-shot setup across all 20 translation directions. Each heatmap corresponds to a specific translation pair, with the x-axis and y-axis denoting the proportion of translation heads and language heads steered, respectively. The score reported next to each translation pair indicates the performance obtained in an instructed zero-shot setup, serving as a topline for comparison.

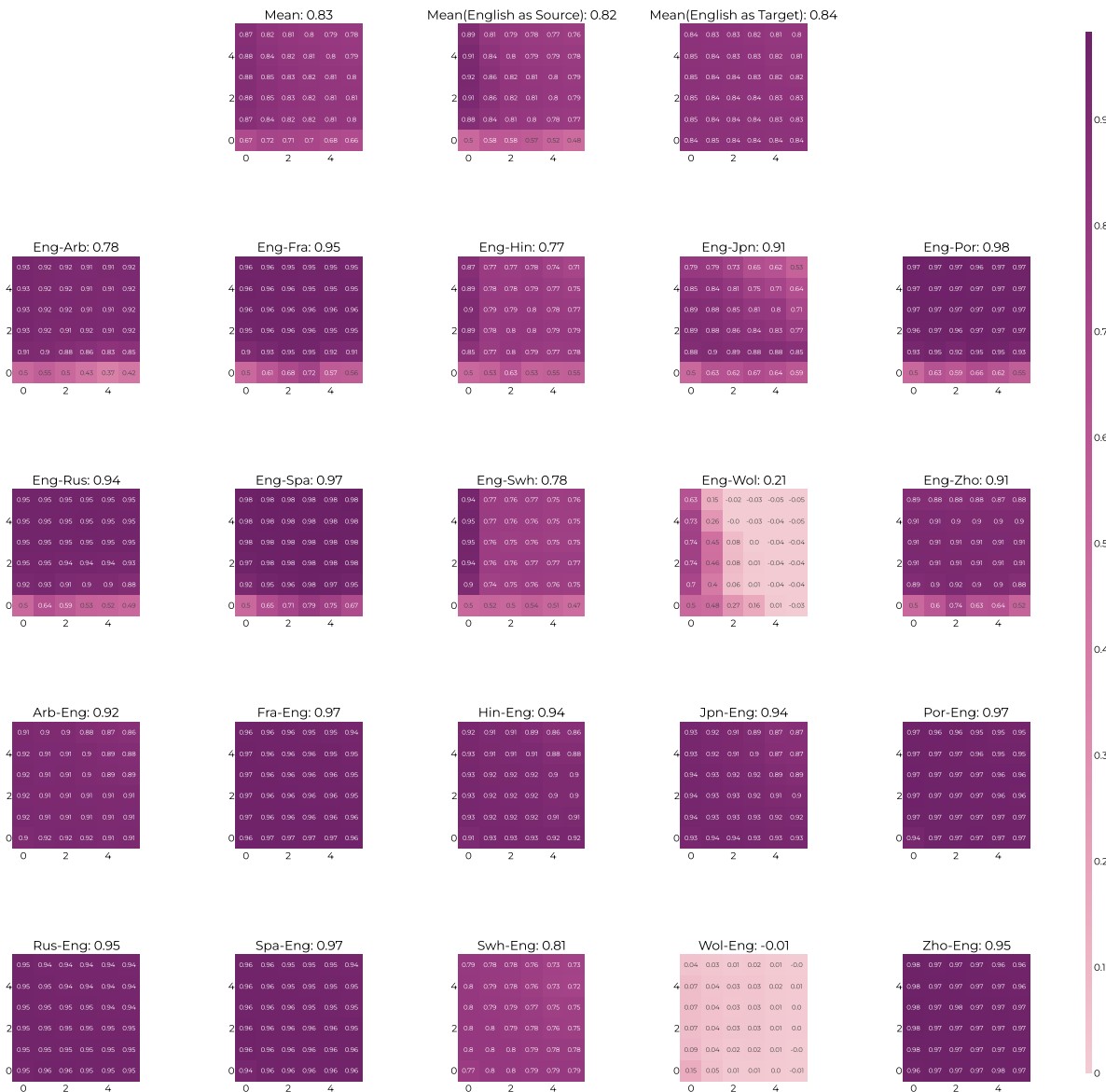

*Figure 48.* Steering-induced MT performance (COMET) for GEMMA-3-12B-PT under an instruction-free zero-shot setup across all 20 translation directions. Each heatmap corresponds to a specific translation pair, with the x-axis and y-axis denoting the proportion of translation heads and language heads steered, respectively. The score reported next to each translation pair indicates the performance obtained in an instructed zero-shot setup, serving as a topline for comparison.

### B.3.3. QWEN-3

We report steering results for the Qwen3 model family across three scales. For each model, we present average scores aggregated over translation pairs with English as the source language and English as the target language: QWEN3-0.6B-BASE (Figure 49), QWEN3-1.7B-BASE (Figure 50), and QWEN3-4B-BASE (Figure 51).

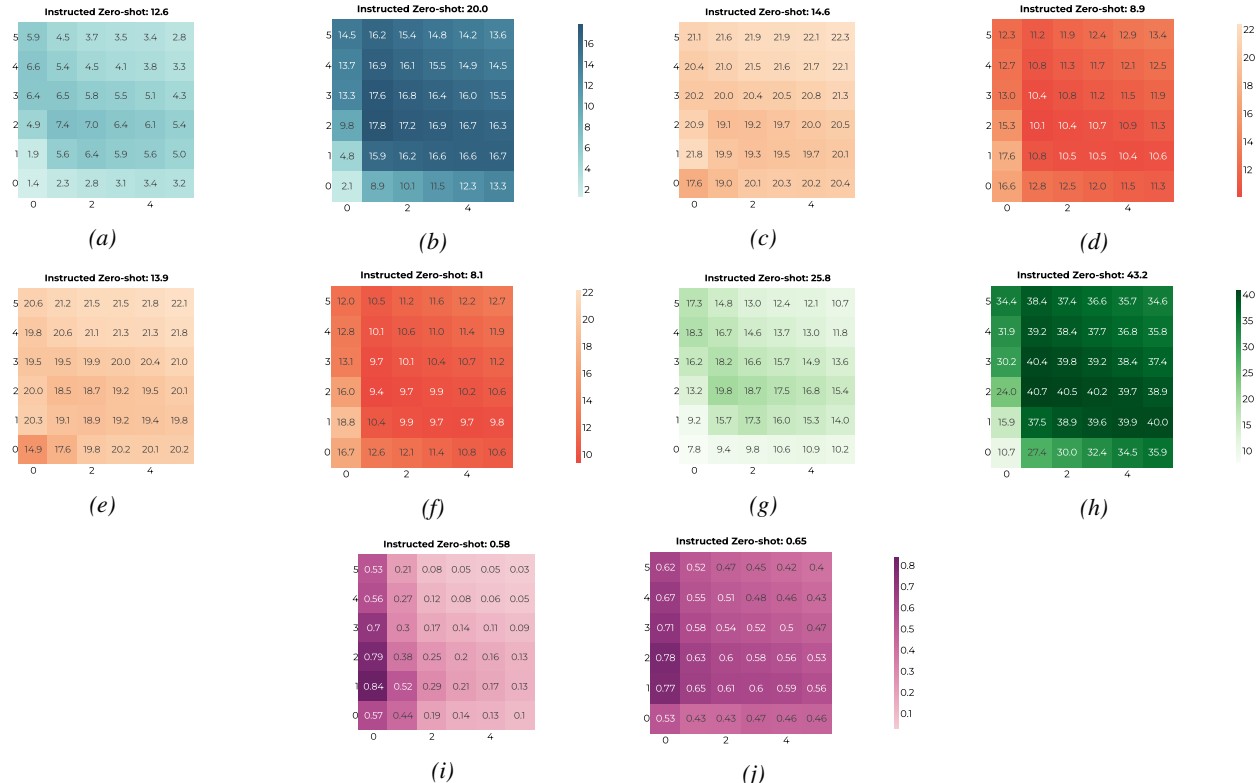

*Figure 49.* Steering-induced MT performance for QWEN3-0.6B-BASE under an instruction-free zero-shot setup. We report BLEU (a, b), MetricX-24 (c, d), MetricX-24 QE (e, f), chrF++ (g, h) and XCOMET (i, j) when steering $n\%$ of translation heads (x-axis) and $m\%$ of language heads (y-axis). Subfigures (a, c, e, g, i) report results for translation pairs with English as the source language, while (b, d, f, h, j) report results for translation pairs with English as the target language.

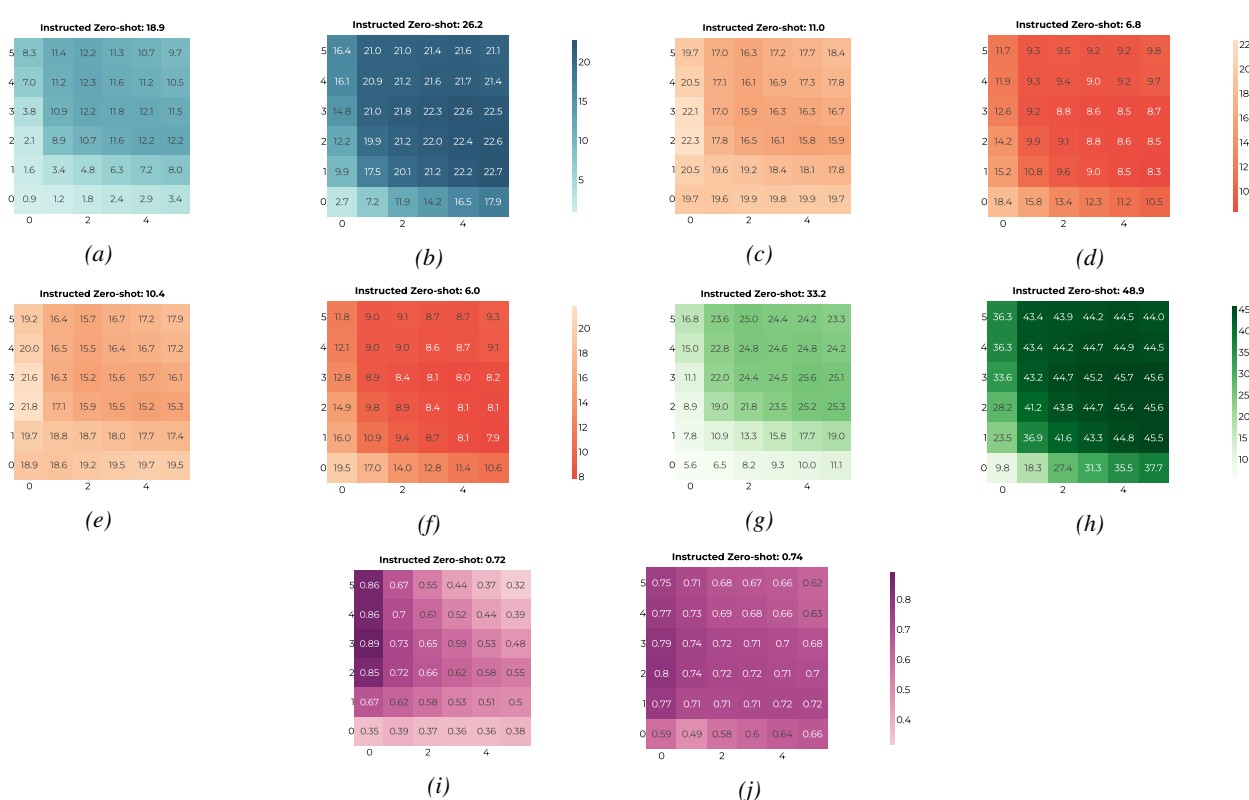

*Figure 50.* Steering-induced MT performance for QWEN3-1.7B-BASE under an instruction-free zero-shot setup. We report BLEU (a, b), MetricX-24 (c, d), MetricX-24 QE (e, f), chrF++ (g, h) and XCOMET (i, j) when steering $n\%$ of translation heads (x-axis) and $m\%$ of language heads (y-axis). Subfigures (a, c, e, g, i) report results for translation pairs with English as the source language, while (b, d, f, h, j) report results for translation pairs with English as the target language.

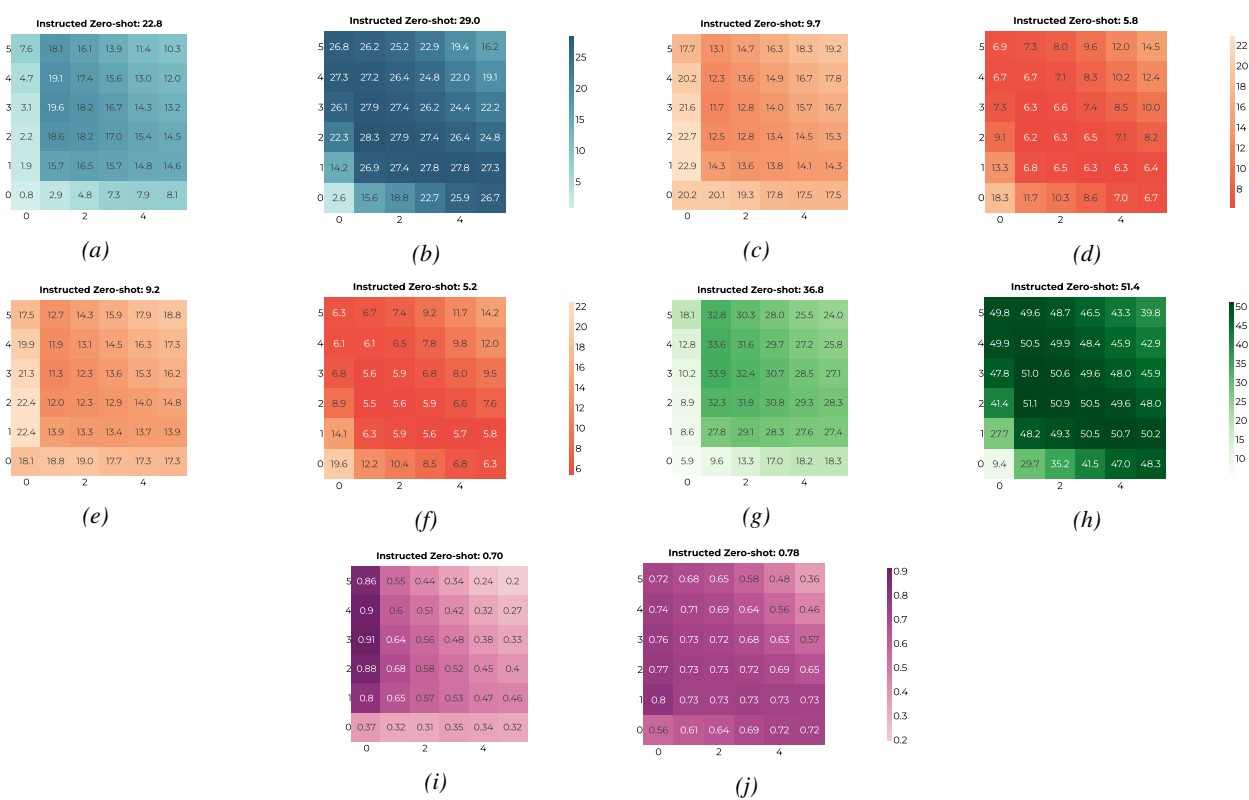

*Figure 51.* Steering-induced MT performance for QWEN3-4B-BASE under an instruction-free zero-shot setup. We report BLEU (a, b), MetricX-24 (c, d), MetricX-24 QE (e, f), chrF++ (g, h) and XCOMET (i, j) when steering $n\%$ of translation heads (x-axis) and $m\%$ of language heads (y-axis). Subfigures (a, c, e, g, i) report results for translation pairs with English as the source language, while (b, d, f, h, j) report results for translation pairs with English as the target language.

### B.3.4. LLAMA-3.2

We report steering results for the Llama-3.2 model family across two scales. For each model, we present average scores aggregated over translation pairs with English as the source language and English as the target language: LLAMA-3.2-1B (Figure 52) and LLAMA-3.2-3B (Figure 53).

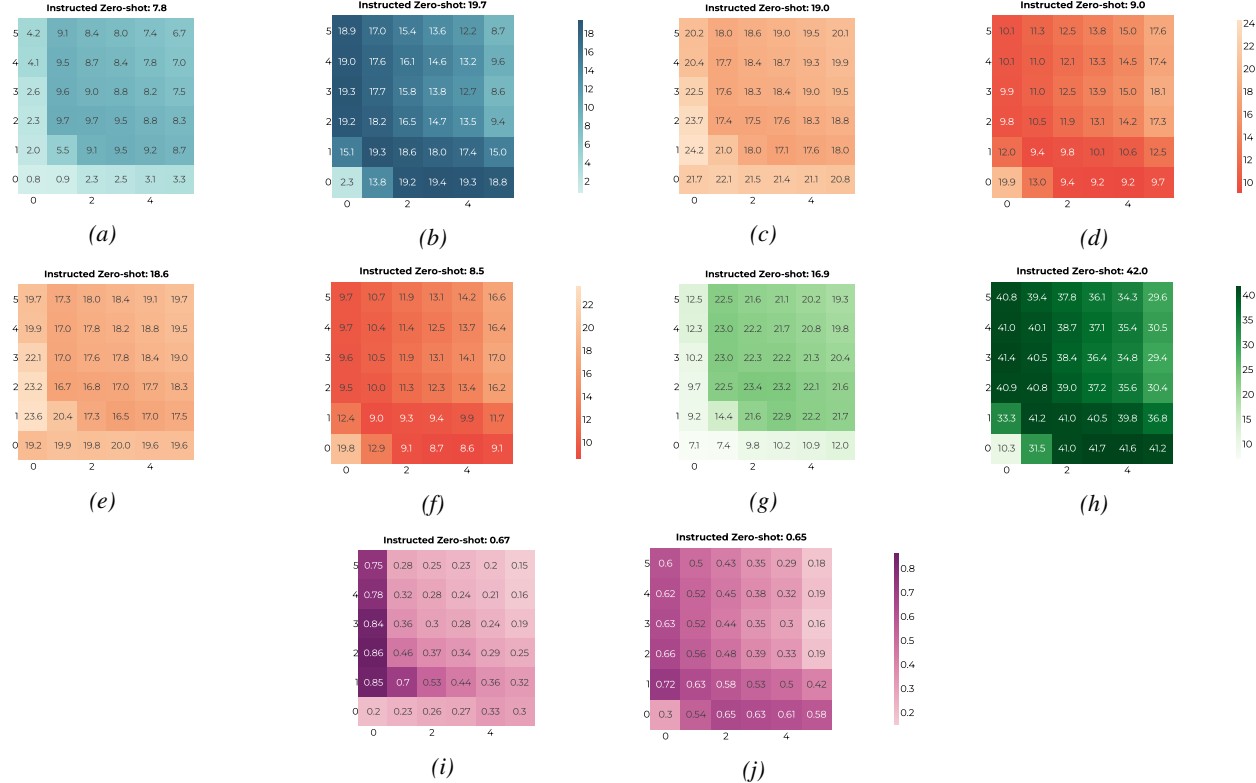

*Figure 52.* Steering-induced MT performance for LLAMA-3.2-1B under an instruction-free zero-shot setup. We report BLEU (a, b), MetricX-24 (c, d), MetricX-24 QE (e, f), chrF++ (g, h) and XCOMET (i, j) when steering $n\%$ of translation heads (x-axis) and $m\%$ of language heads (y-axis). Subfigures (a, c, e, g i) report results for translation pairs with English as the source language, while (b, d, f, h, j) report results for translation pairs with English as the target language.

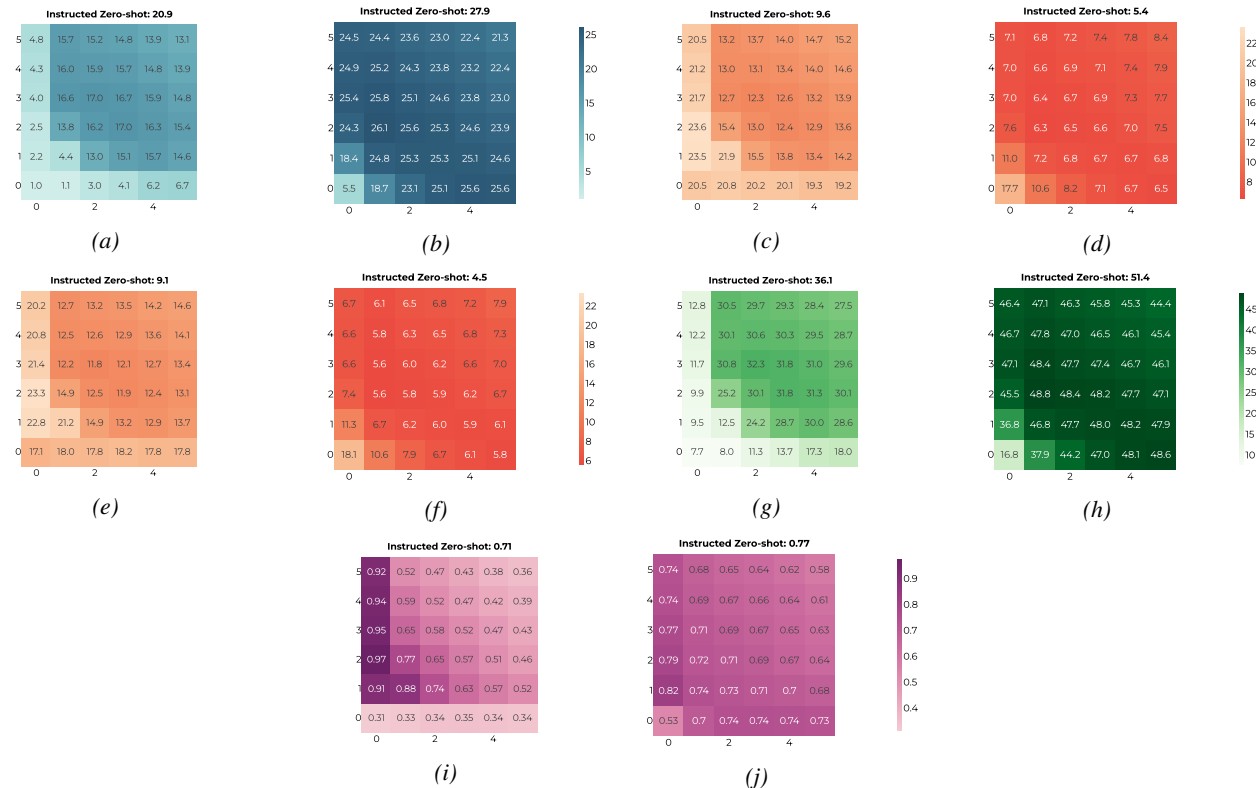

*Figure 53.* Steering-induced MT performance for LLAMA-3.2-3B under an instruction-free zero-shot setup. We report BLEU (a, b), MetricX-24 (c, d), MetricX-24 QE (e, f), chrF++ (g, h) and XCOMET (i, j) when steering $n\%$ of *translation heads* (x-axis) and $m\%$ of *language heads* (y-axis). Subfigures (a, c, e, g, i) report results for translation pairs with English as the source language, while (b, d, f, h, j) report results for translation pairs with English as the target language.

## B.4. Ablation

Following the experiments in Section 5, we provide detailed ablation results across models and translation pairs. We first present a qualitative analysis of failure modes when ablating *language heads* or *translation heads* (Appendix B.4.1), illustrating the distinct functional roles of each head type. We then report quantitative results using six metrics: BLEU, MetricX-24, MetricX-24 QE, chrF++, XCOMET and the target language accuracy. For GEMMA-3-12B-PT, we report results for all 20 translation directions, comprising English from and into 10 languages: Arabic, Chinese, French, Hindi, Japanese, Portuguese, Russian, Spanish, Swahili, and Wolof. For the remaining models, we report average scores aggregated over translation pairs with English as the source language and English as the target language, respectively. Complete quantitative results are organized by model family: Gemma-3 (Appendix B.4.2), Qwen3 (Appendix B.4.3), and LLaMA-3.2 (Appendix B.4.4).

### B.4.1. QUALITATIVE ANALYSIS

*Table 4.* Illustrative failure modes when ablating 1% of translation heads or 1% of language heads in GEMMA-3-12B-PT for English→French translation using an instructed zero-shot prompt. When ablating language heads, the model exhibits language switching, producing outputs that interleave French with English or other languages, indicating a degraded ability to maintain target language identity. Conversely, when ablating translation heads, the model consistently generates in French but produces translations of lower quality, omitting information or failing to preserve the full semantic content of the source. These distinct failure patterns corroborate that language heads are primarily responsible for target language identification, while *translation heads* focus on semantic equivalence.

| Query | Reference | Ablating translation heads | Ablating language heads |
|---|---|---|---|
| "We will endeavour to cut carbon dioxide emissions per unit of GDP by a notable margin by 2020 from the 2005 level," Hu said. | « Nous nous efforcerons de réduire les émissions de dioxyde de carbone par unité de PIB de manière significative d'ici 2020 par rapport au niveau de 2005 », a déclaré M. Hu. | « Nous nous efforcerons de réduire les émissions de dioxyde de carbone par unité de PIB de manière significative d'ici 2020 par rapport au niveau de 2005 », a déclaré M. Hu. | "Nous nous efforcerons de réduire les émissions de dióxido de carbono per unit of GDP by a notable margin by 2020 from the 2005 level," Hu said. |
| "We now have 4-month-old mice that are non-diabetic that used to be diabetic," he added. | « Nous avons à présent des souris de 4 mois qui ne sont pas diabétiques alors qu'elles l'étaient auparavant », a-t-il ajouté. | "Nous avons maintenant des souris de 4 mois qui ne sont plus diabétiques." | "Nous avons maintenant des souris de 4 mois qui ne sont plus diabétiques, mais qui l'étaient." |
| Science's main goal is to figure out the way the world works through the scientific method. This method in fact guides most scientific research. | L'objectif principal de la science est de comprendre la façon dont le monde fonctionne grâce à la méthode scientifique. Cette méthode guide en fait la plupart des recherches scientifiques. | La science a pour but de comprendre le monde qui nous entoure. | The scientific method is a process that scientists use to answer questions about the world around them. It is a way of thinking that helps scientists to understand the world around them. |
| Thousands of years ago, a man called Aristarchus said that the Solar System moved around the Sun. | Il y a quelques milliers d'années, un homme appelé Aristarque a affirmé que le système solaire se déplaçait autour du Soleil. | Il y a des milliers d'années, un homme appelé Aristarchus a dit que le système solaire tournait autour du soleil. | Il y a des milliers d'années, un homme appelé Aristarchus said that the Solar System moved around the Sun. |

### B.4.2. GEMMA-3

We report ablation results for the Gemma-3 model family across four scales. For GEMMA-3-12B-PT, we present results for all 20 translation directions. For the smaller models, we report average scores aggregated over translation pairs with English as the source language and English as the target language: GEMMA-3-270M (Figure 54), GEMMA-3-1B-PT (Figure 55), GEMMA-3-4B-PT (Figure 56), and GEMMA-3-12B-PT (Figures 57–62).

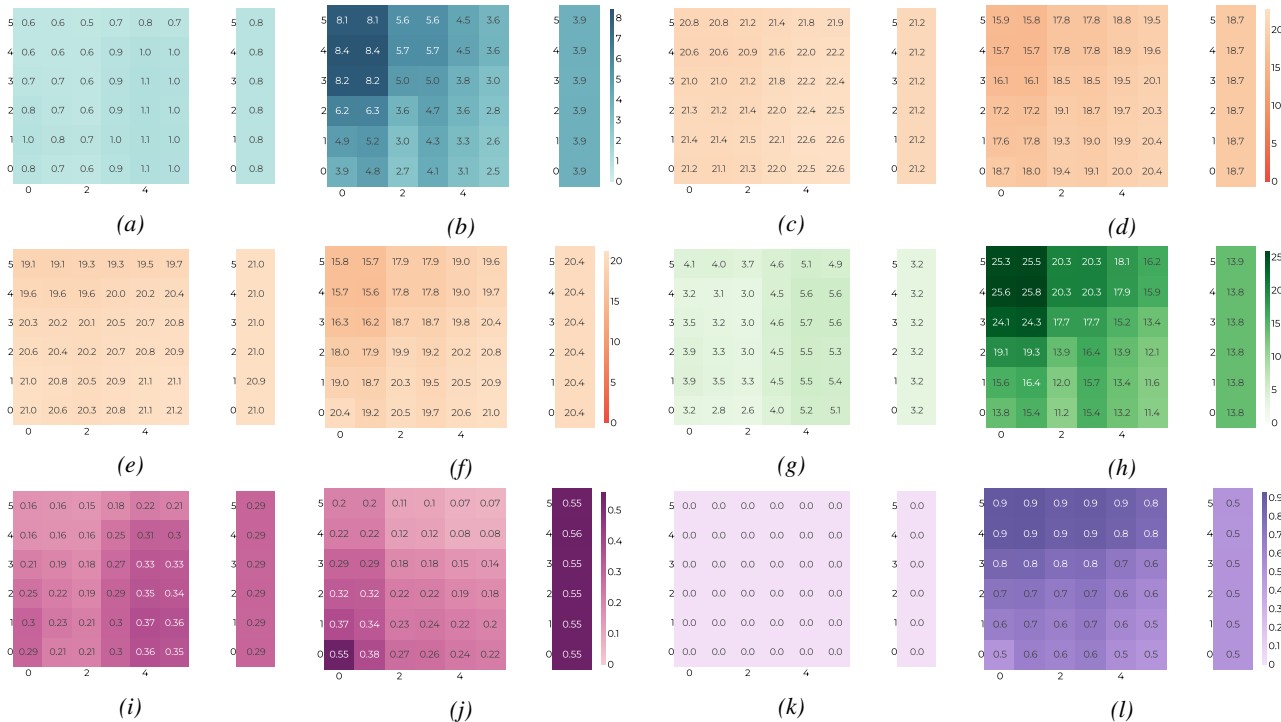

*Figure 54.* MT performance under head ablation for GEMMA-3-270M using an instructed zero-shot setup. We report BLEU (a, b), MetricX-24 (c, d), MetricX-24 QE (e, f), chrF++ (g, h), XCOMET (i, j), and target language accuracy (k, l) when ablating and $n\%$ of translation heads (x-axis) and $m\%$ of language heads (y-axis) . Subfigures (a, c, e, g, i, k) report results for translation pairs with English as the source language, while (b, h, d, f, j, l) report results for translation pairs with English as the target language.

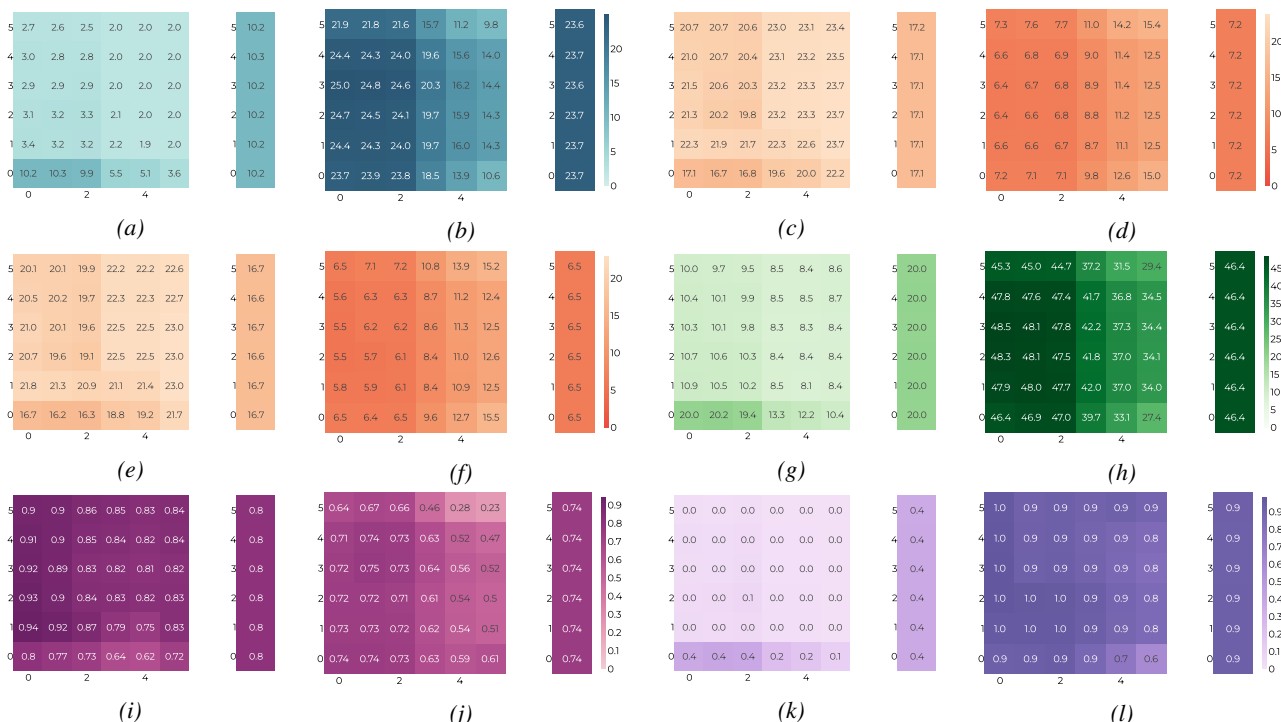

*Figure 55.* MT performance under head ablation for GEMMA-3-1B-PT using an instructed zero-shot setup. We report BLEU (a, b), MetricX-24 (c, d), MetricX-24 QE (e, f), chrF++ (g, h), XCOMET (i, j), and target language accuracy (k, l) when ablating and $n\%$ of translation heads (x-axis) and $m\%$ of language heads (y-axis) . Subfigures (a, c, e, g, i, k) report results for translation pairs with English as the source language, while (b, h, d, f, j, l) report results for translation pairs with English as the target language.

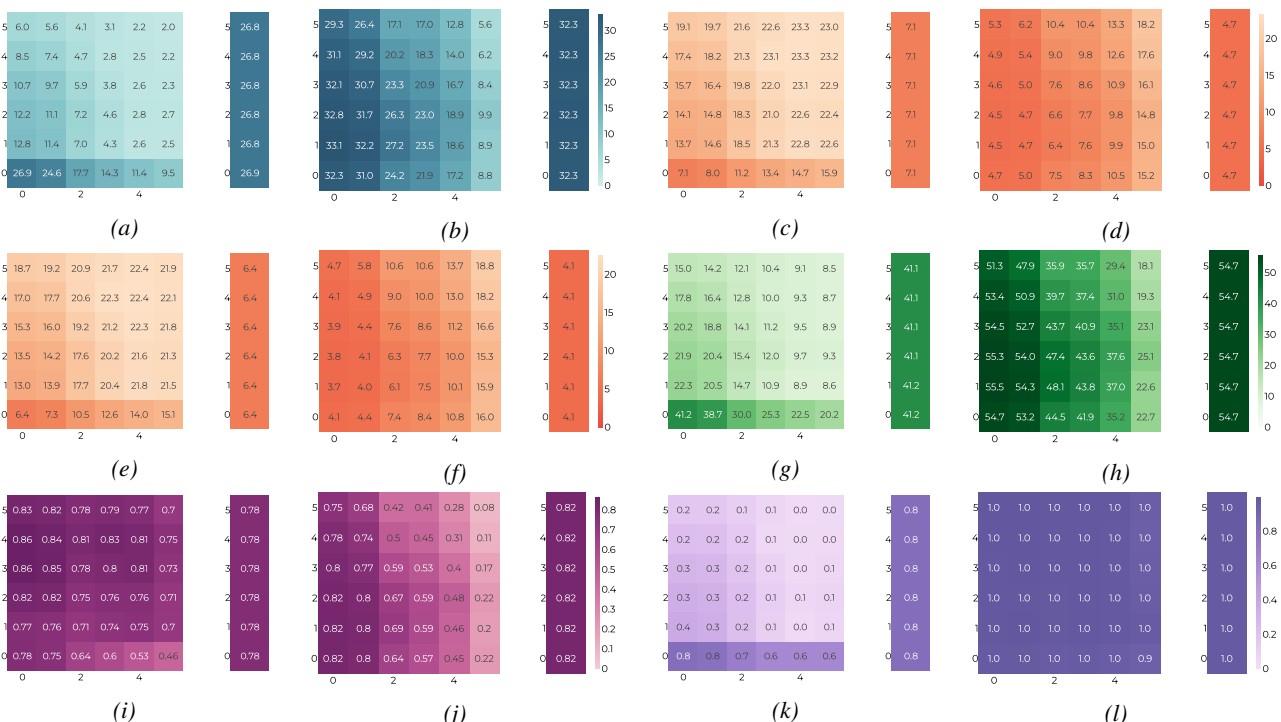

*Figure 56.* MT performance under head ablation for GEMMA-3-4B-PT using an instructed zero-shot setup. We report BLEU (a, b), MetricX-24 (c, d), MetricX-24 QE (e, f), chrF++ (g, h), XCOMET (i, j), and target language accuracy (k, l) when ablating and $n\%$ of translation heads (x-axis) and $m\%$ of language heads (y-axis) . Subfigures (a, c, e, g, i, k) report results for translation pairs with English as the source language, while (b, h, d, f, j, l) report results for translation pairs with English as the target language.

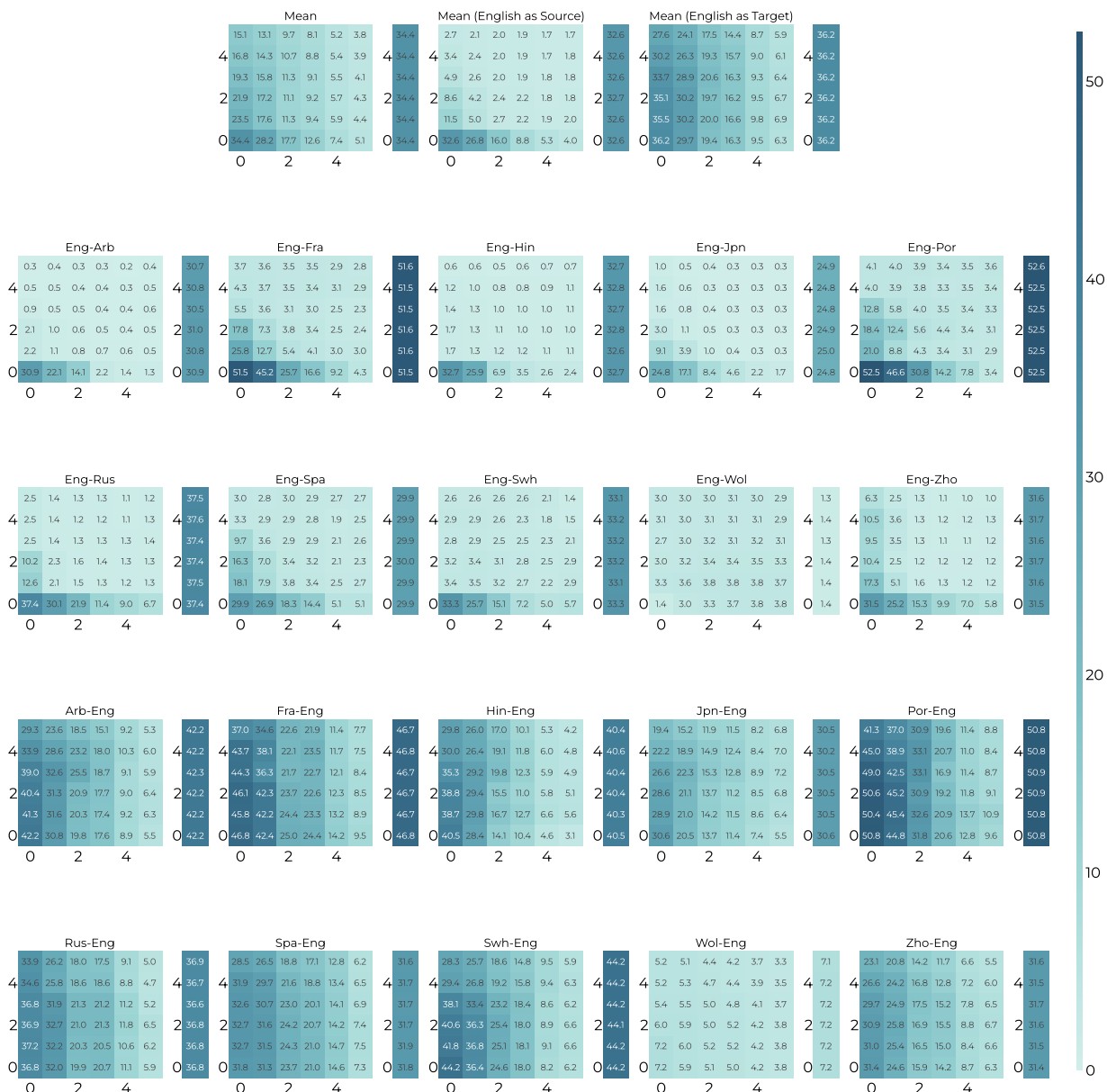

*Figure 57.* MT performance under head ablation (BLEU) for GEMMA-3-12B-PT across all 20 translation directions using an instructed zero-shot setup. Each heatmap shows performance when ablating $n\%$ of *Translation Heads* (x-axis) and $m\%$ of *Language Heads* (y-axis) for a specific translation pair. Results are compared against ablating j% of randomly selected heads.

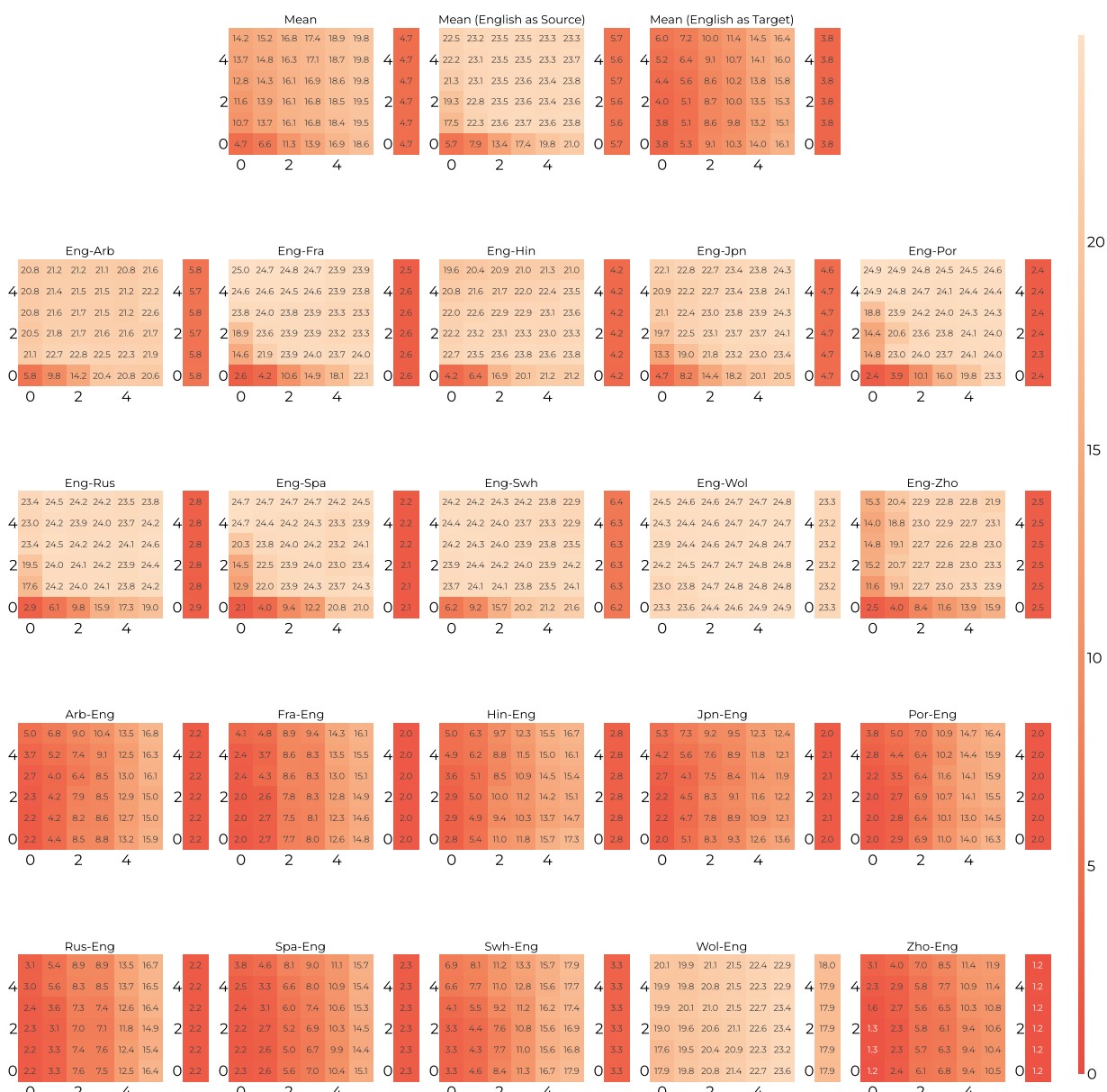

*Figure 58.* MT performance under head ablation (MetricX-24) for GEMMA-3-12B-PT across all 20 translation directions using an instructed zero-shot setup. Each heatmap shows performance when ablating $n\%$ of *Translation Heads* (x-axis) and $m\%$ of *Language Heads* (y-axis) for a specific translation pair. Results are compared against ablating j% of randomly selected heads.

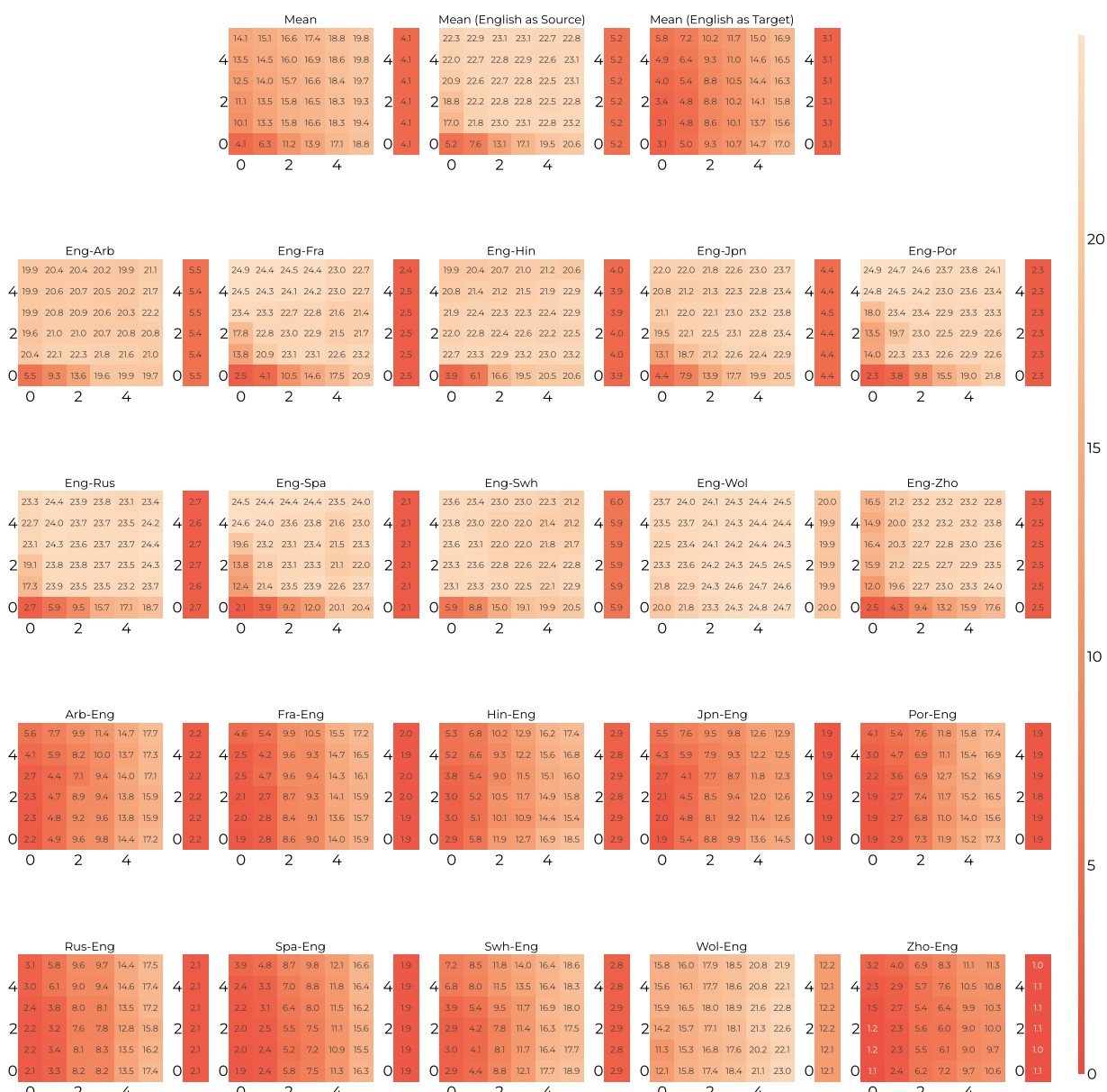

*Figure 59.* MT performance under head ablation (MetricX-24 QE) for GEMMA-3-12B-PT across all 20 translation directions using an instructed zero-shot setup. Each heatmap shows performance when ablating $n\%$ of *Translation Heads* (x-axis) and $m\%$ of *Language Heads* (y-axis) for a specific translation pair. Results are compared against ablating j% of randomly selected heads.

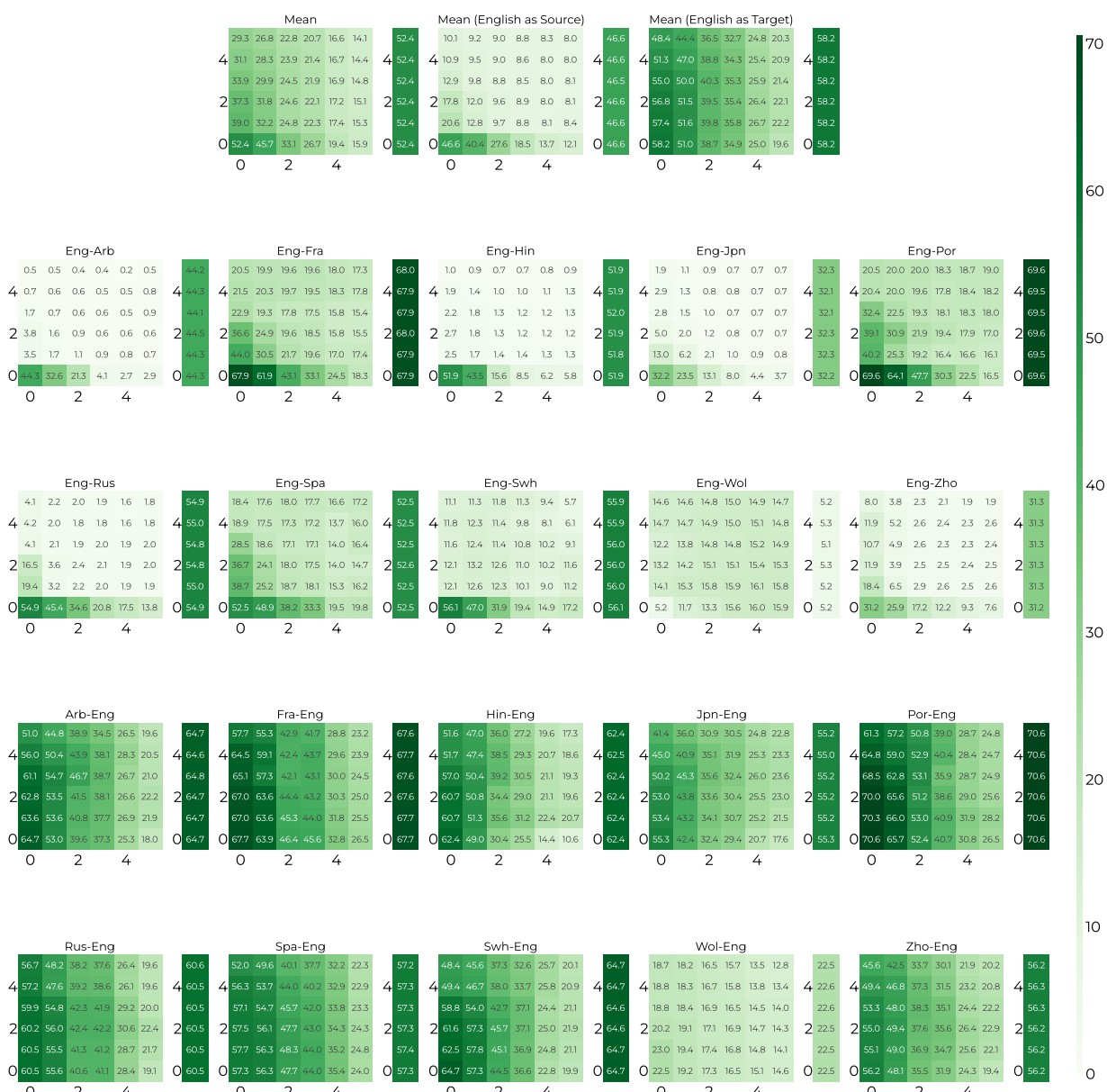

*Figure 60.* MT performance under head ablation (CHRF++) for GEMMA-3-12B-PT across all 20 translation directions using an instructed zero-shot setup. Each heatmap shows performance when ablating $n\%$ of *Translation Heads* (x-axis) and $m\%$ of *Language Heads* (y-axis) for a specific translation pair. Results are compared against ablating j% of randomly selected heads.

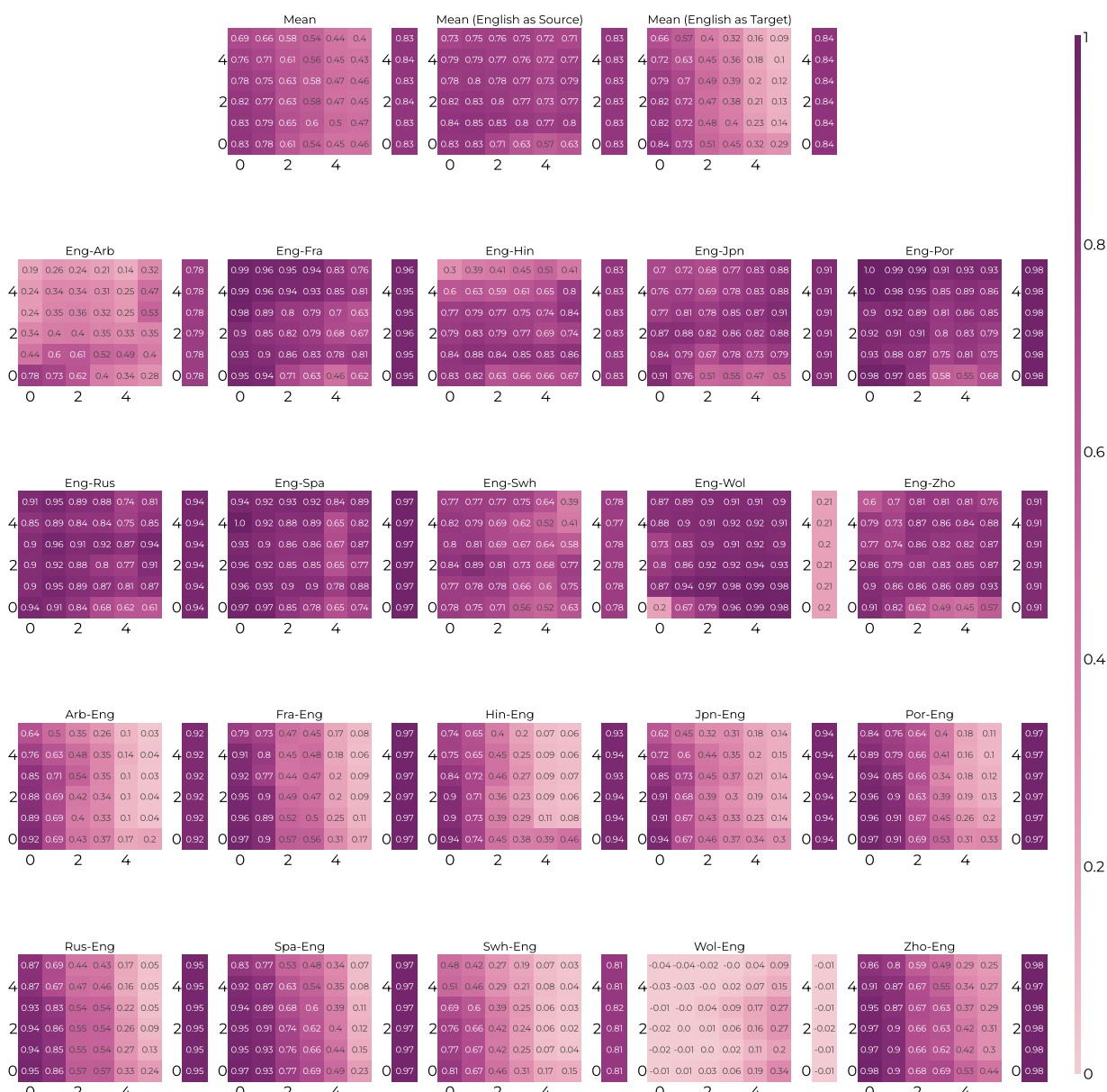

*Figure 61.* MT performance under head ablation (COMET) for GEMMA-3-12B-PT across all 20 translation directions using an instructed zero-shot setup. Each heatmap shows performance when ablating $n\%$ of *Translation Heads* (x-axis) and $m\%$ of *Language Heads* (y-axis) for a specific translation pair. Results are compared against ablating j% of randomly selected heads.

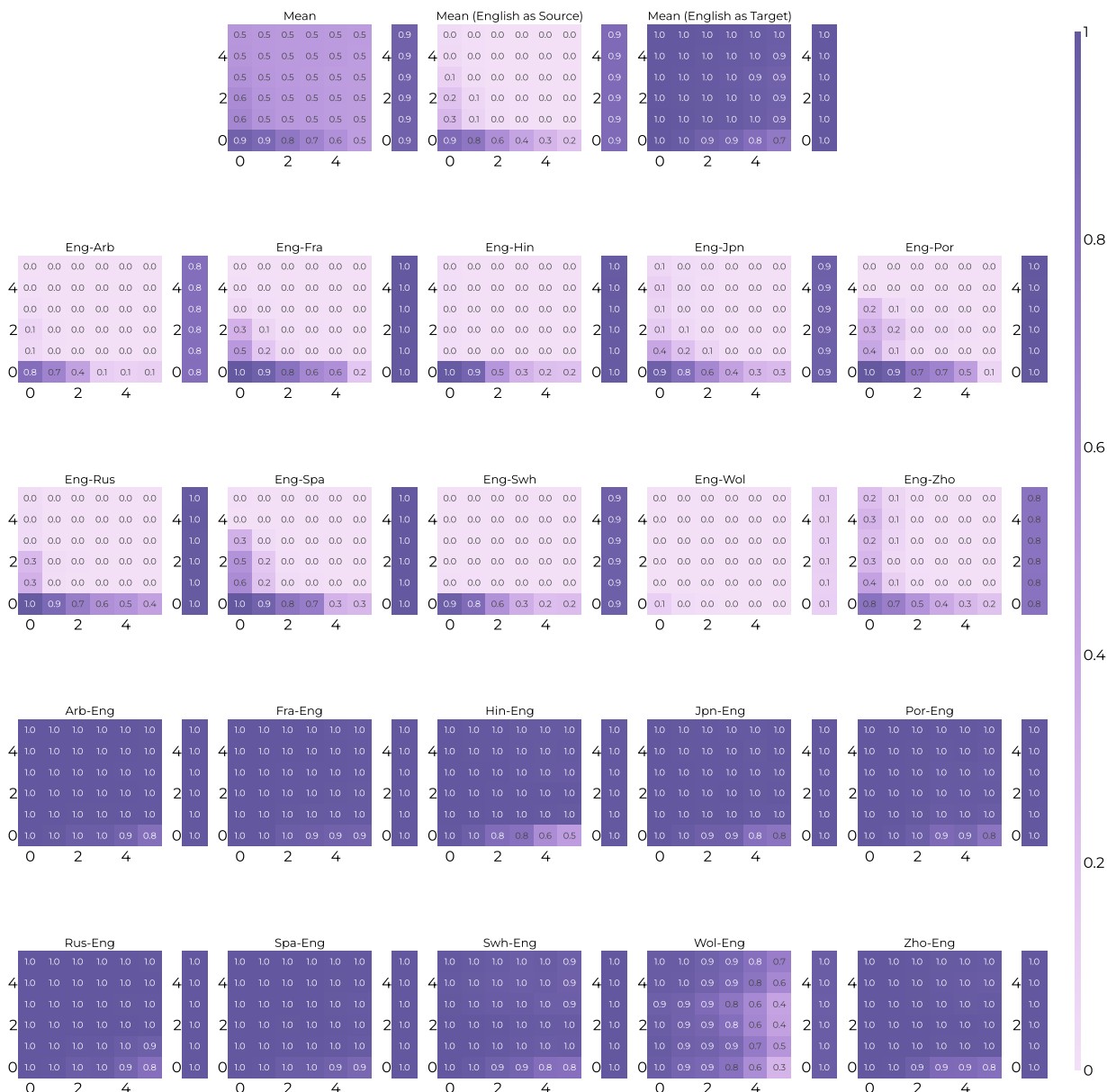

*Figure 62.* Target language accuracy under head ablation for GEMMA-3-12B-PT across all 20 translation directions using an instructed zero-shot setup. Each heatmap shows performance when ablating $n\%$ of *Translation Heads* (x-axis) and $m\%$ of *Language Heads* (y-axis) for a specific translation pair. Results are compared against ablating j% of randomly selected heads.

### B.4.3. QWEN-3

We report ablation results for the Qwen-3 model family across three scales. For each model, we present average scores aggregated over translation pairs with English as the source language and English as the target language: QWEN3-0.6B-BASE (Figure 63), QWEN3-1.7B-BASE (Figure 64) and QWEN3-4B-BASE (Figure 65).

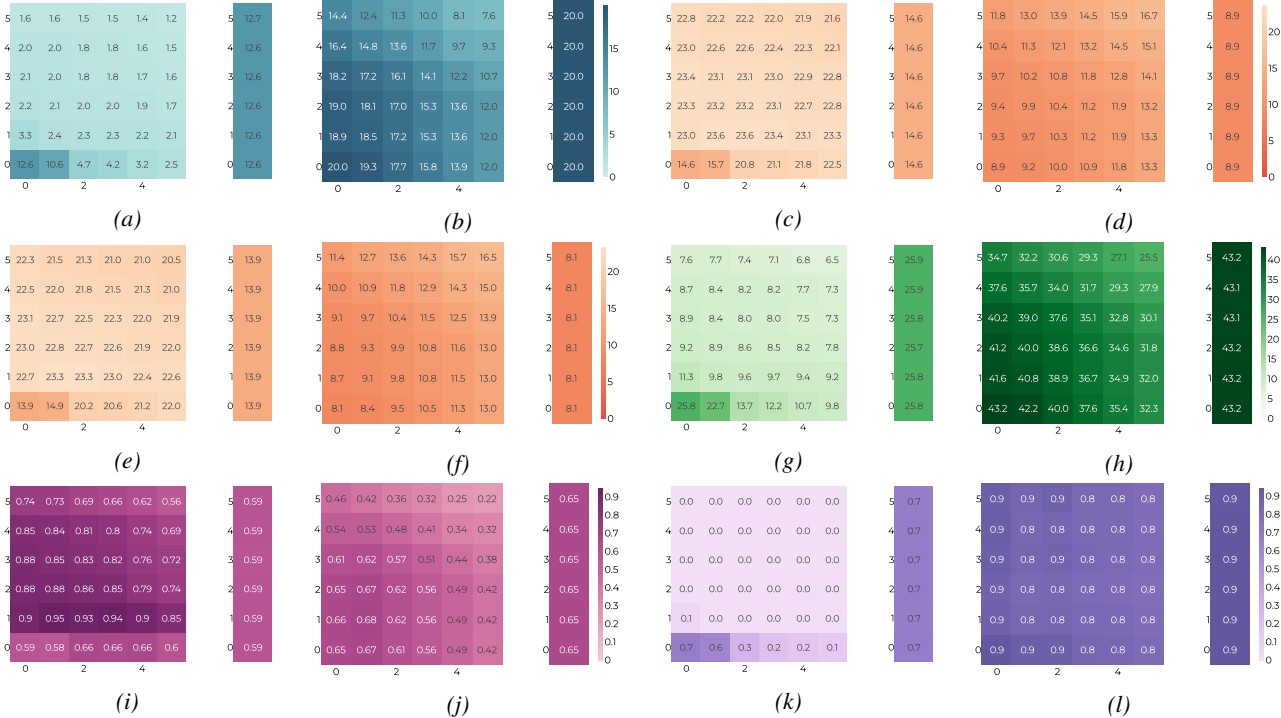

*Figure 63.* MT performance under head ablation for QWEN3-0.6B-BASE using an instructed zero-shot setup. We report BLEU (a, b), MetricX-24 (c, d), MetricX-24 QE (e, f), chrF++ (g, h), XCOMET (i, j), and target language accuracy (k, l) when ablating and $n\%$ of translation heads (x-axis) and $m\%$ of language heads (y-axis) . Subfigures (a, c, e, g, i, k) report results for translation pairs with English as the source language, while (b, h, d, f, j, l) report results for translation pairs with English as the target language.

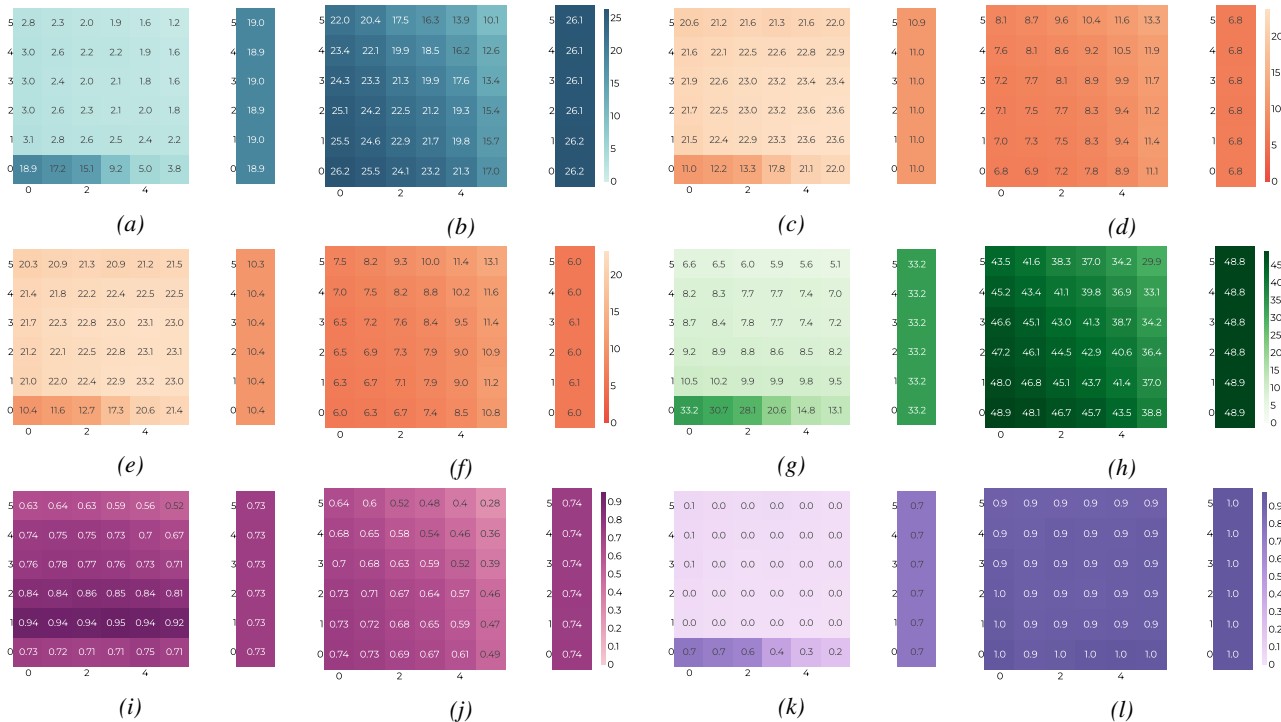

*Figure 64.* MT performance under head ablation for QWEN3-1.7B-BASE using an instructed zero-shot setup. We report BLEU (a, b), MetricX-24 (c, d), MetricX-24 QE (e, f), chrF++ (g, h), XCOMET (i, j), and target language accuracy (k, l) when ablating and $n\%$ of translation heads (x-axis) and $m\%$ of language heads (y-axis) . Subfigures (a, c, e, g, i, k) report results for translation pairs with English as the source language, while (b, h, d, f, j, l) report results for translation pairs with English as the target language.

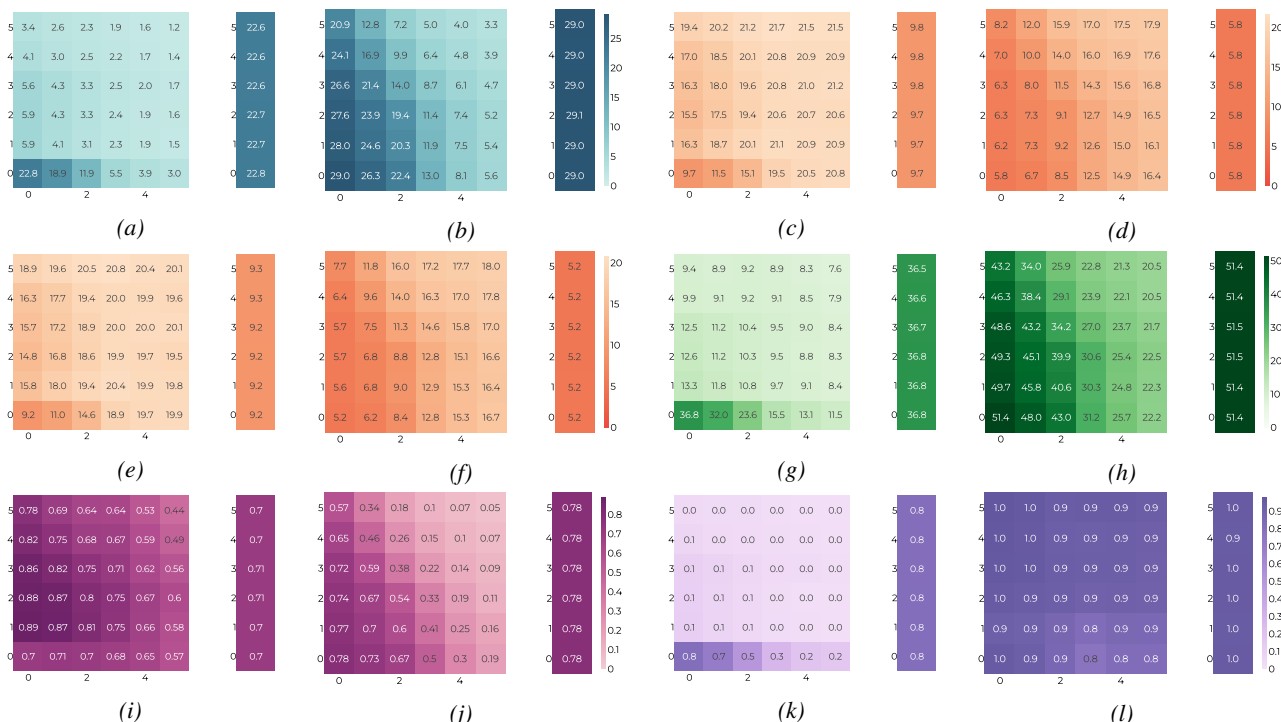

*Figure 65.* MT performance under head ablation for QWEN3-4B-BASE using an instructed zero-shot setup. We report BLEU (a, b), MetricX-24 (c, d), MetricX-24 QE (e, f), chrF++ (g, h), XCOMET (i, j), and target language accuracy (k, l) when ablating and $n\%$ of translation heads (x-axis) and $m\%$ of language heads (y-axis) . Subfigures (a, c, e, g, i, k) report results for translation pairs with English as the source language, while (b, h, d, f, j, l) report results for translation pairs with English as the target language.

## B.4.4. LLAMA-3.2

We report ablation results for the Llama-3.2 model family across two scales. For each model, we present average scores aggregated over translation pairs with English as the source language and English as the target language: LLAMA-3.2-1B (Figure 66) and LLAMA-3.2-3B (Figure 67).

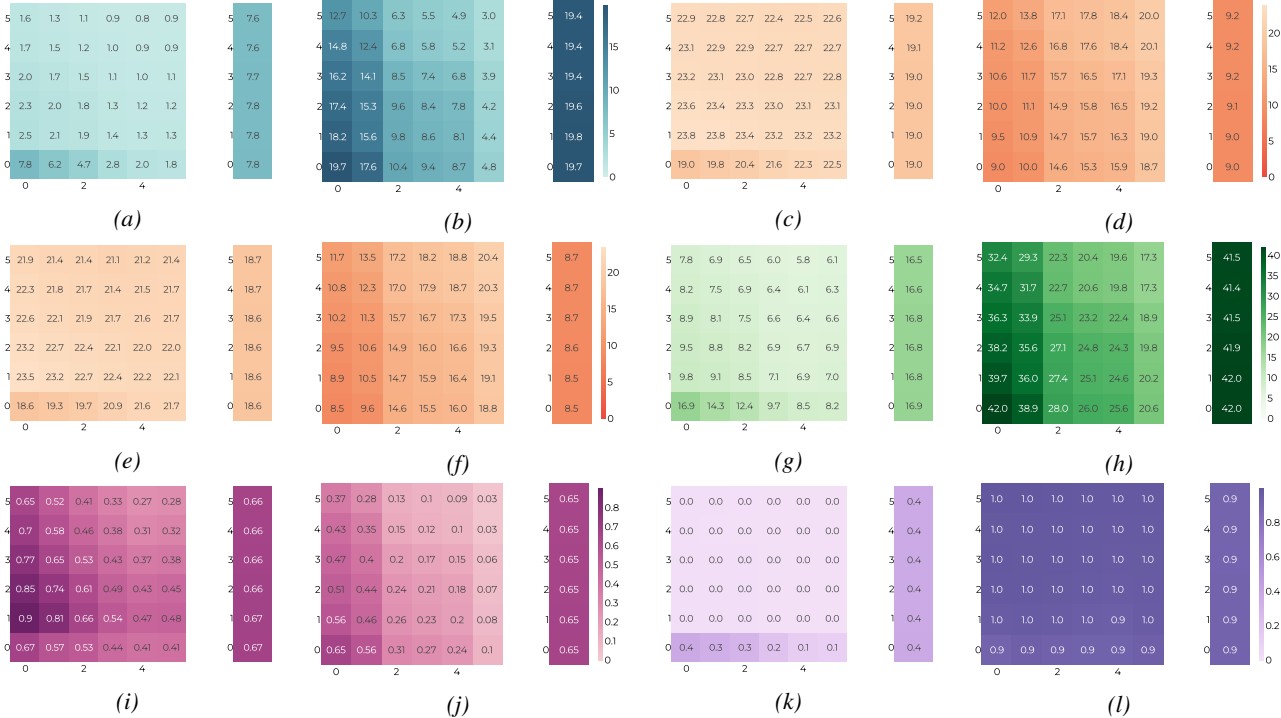

*Figure 66.* MT performance under head ablation for LLAMA-3.2-1B using an instructed zero-shot setup. We report BLEU (a, b), MetricX-24 (c, d), MetricX-24 QE (e, f), chrF++ (g, h), XCOMET (i, j), and target language accuracy (k, l) when ablating and $n\%$ of translation heads (x-axis) and $m\%$ of language heads (y-axis) . Subfigures (a, c, e, g, i, k) report results for translation pairs with English as the source language, while (b, h, d, f, j, l) report results for translation pairs with English as the target language.

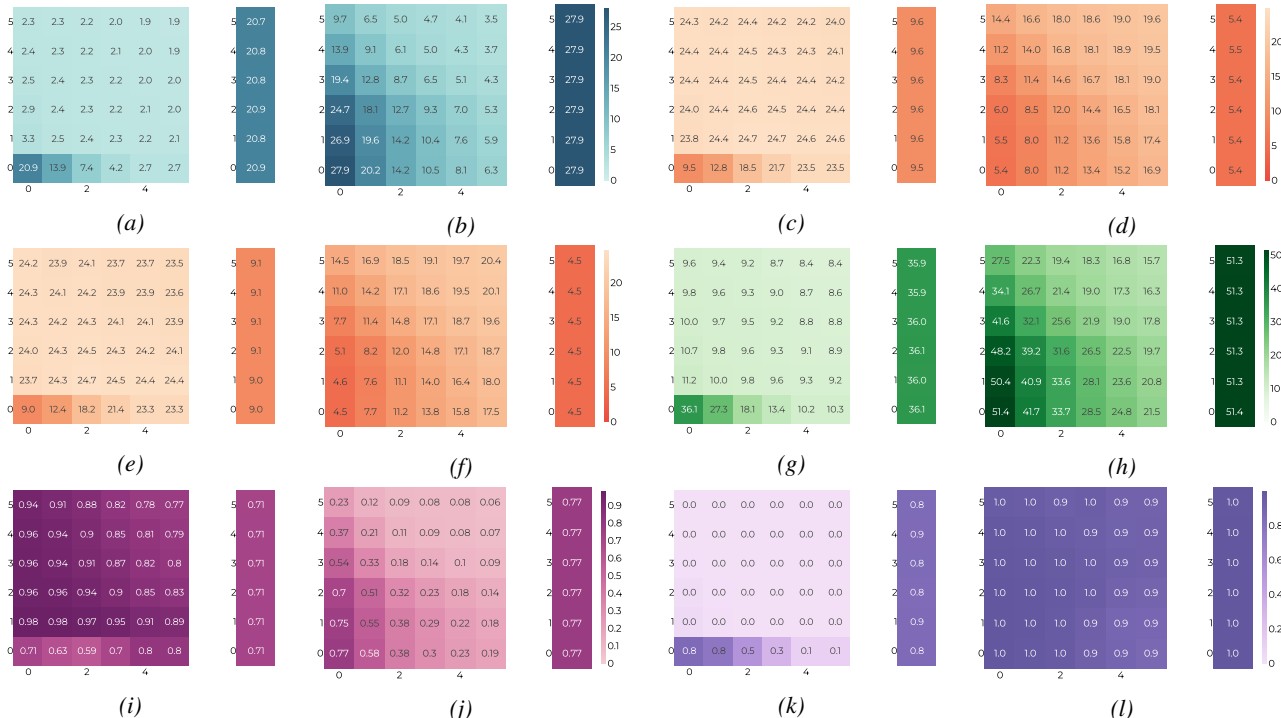

*Figure 67.* MT performance under head ablation for LLAMA-3.2-3B using an instructed zero-shot setup. We report BLEU (a, b), MetricX-24 (c, d), MetricX-24 QE (e, f), chrF++ (g, h), XCOMET (i, j), and target language accuracy (k, l) when ablating and $n\%$ of translation heads (x-axis) and $m\%$ of language heads (y-axis) . Subfigures (a, c, e, g, i, k) report results for translation pairs with English as the source language, while (b, h, d, f, j, l) report results for translation pairs with English as the target language.

## B.5. Few-shot scores

Following experiments on steering in Section 5, we provide an additional topline in a few-shot . Table 5 reports average scores across all studied translation directions.

*Table 5.* Few-shot MT performance on the *devtest* set of FLORES-200 using 5 randomly selected demonstrations from the *dev* set. We report BLEU, MetricX, MetricX QE, chrF++ and XCOMET scores averaged across all studied translation directions, serving as an additional topline for the steering experiments.

|  | BLEU | MetricX | MetricX QE | CHRF++ | XCOMET |
|---|---|---|---|---|---|
| GEMMA-3-12B-PT | 37.51 | 4.06 | 3.63 | 55.37 | 0.84 |
| GEMMA-3-4B-PT | 33.15 | 4.79 | 4.25 | 51.84 | 0.82 |
| GEMMA-3-1B-PT | 24.55 | 7.29 | 6.59 | 44.12 | 0.71 |
| GEMMA-3-270M | 12.15 | 13.55 | 12.79 | 30.66 | 0.39 |
| QWEN3-4B-BASE | 28.16 | 6.71 | 5.98 | 46.74 | 0.73 |
| QWEN3-1.7B-BASE | 23.64 | 8.15 | 7.34 | 42.41 | 0.67 |
| QWEN3-0.6B-BASE | 17.47 | 10.83 | 9.87 | 36.36 | 0.55 |
| LLAMA-3.2-3B | 26.74 | 6.55 | 5.78 | 46.02 | 0.73 |
| LLAMA-3.2-1B | 19.34 | 9.27 | 8.48 | 38.58 | 0.59 |

## B.6. Steering vector decoding

To gain insight into the information encoded by the identified heads, we decode the steering vectors by decoding the mean head outputs. We report the top 20 tokens for the top-1 translation head and top-1 language head across five translation directions for GEMMA-3-270M (Table 6), GEMMA-3-1B-PT (Table 7), GEMMA-3-4B-PT (Table 8) and GEMMA-3-12B-PT (Table 9). For models from 1B parameters onward, the equivalence vectors exhibit substantial linguistic diversity as their top tokens span multiple languages and scripts simultaneously, regardless of the translation direction. For instance, in

GEMMA-3-1B-PT, the English→French equivalence vector contains tokens from Russian, Arabic, Thai, and Chinese, among others. This multilingual mixing persists in GEMMA-3-12B-PT as equivalence vectors include Greek, German, Thai, Chinese, and Japanese tokens. GEMMA-3-270M does not exhibit clear patterns, with decoded tokens appearing largely noisy across both head types. In contrast, the language vectors show strong alignment to the intended target language. For GEMMA-3-1B-PT, the English→French language vector yields French tokens (*à*, *la*, *que*, *tout*, *des*), while the English→Portuguese vector produces Portuguese tokens (*uma*, *não*, *mais*, *você*). Similarly, in GEMMA-3-4B-PT, the English→Portuguese language vector decodes to tokens such as *Brazilian*, *Portuguese*, *Brazil*, and *São*. This target-language specificity holds across model scales and directions, with Chinese vectors decoding predominantly to Chinese characters and Arabic vectors to Arabic script.d

*Table 6.* First 20 tokens obtained by decoding the steering vectors from the top-1 *Translation Head* and top-1 *Language Head* of GEMMA-3-270M for five translation directions. For each direction, we project the mean head output onto the vocabulary space and report the highest-ranked tokens.

| Language Pair | Translation | Language |
|---|---|---|
| Eng-Fra | verticalLayout 事实上 Fach Sie correspond ز Zie Herz ﻰ ⌐ suspend Benutzer Consult ته Zij Understand Gü | verticalLayout ⌐ Herz Fach correspond Sie Benutzer ˙ ر 事实上 Zie ⌐ suspend Zij ﻰ ته người Understand |
| Eng-Swh | châ DataTypes Jenis Chartered polyfills AppModule Organisations NotFound GameManager disorderly vâ idmat retry-Writes Shareholders Chứng Travelling Topological Organis '| OnTrigger | châ DataTypes Jenis Chartered polyfills AppModule Organisations GameManager disorderly Chứng Shareholders Runtime '|NotFound Organis OnTrigger Topological Biodiversity retryWrites Bayes |
| Eng-Zho | 事实上 strconv wèi kors ernst qīng ⌐ yě daher jīn zhī 我国 fēng Benutzer getline řel zhōng shēng yú qí | 事实上 kors strconv wèi ernst ⌐ qīng yě Benutzer getline daher zhī jīn fēng strtok leistung ilberger zhōng řel ländischen |
| Eng-Arb | châ strconv cump şɔ $(' tind defineProperty Zie Portanto Understand Onun Corresponding BlogPost webdriver Bundan waxay homic polyfills fub verticalLayout | châ strconv cump şɔ $(' tind Portanto Corresponding defineProperty Zie Understand Onun kors BlogPost Bundan webdriver homic polyfills mores fub |
| Eng-Por | Benutzer Fach Corresponding correspond Portanto XNUMX strconv 事实上 kors Read Zie READ suspend mortal verticalLayout Medien caract 协同 (…) | Benutzer XNUMX Corresponding correspond Fach kors Portanto strconv suspend ⌐ ⌐ (…) verticalLayout READ 事实上 Read Medien caract Zie |

*Table 7.* First 20 tokens obtained by decoding the steering vectors from the top-1 *Translation Head* and top-1 *Language Head* of GEMMA-3-1B-PT for five translation directions. For each direction, we project the mean head output onto the vocabulary space and report the highest-ranked tokens.

| Language Pair | Translation | Language |
|---|---|---|
| Eng-Fra | на не в это علم того I The с за да а इस يف все The 这 on как In | à la que tout é des de mais qui dé ré dans É très ou les mé comme au pré |
| Eng-Swh | на того ใน в білเ ٷ tego нам يف мы рам toho он во ٳ ви フ เบ ของ этого | kwa kwenye katika una ambayo lakini uso ikuwa baada yake weza kuwa pamoja wakati tego Kenyans în kutoka và Una |
| Eng-Zho | 在 如果 一 不 的 да 大 на 这 По 有 新 这个 我们 当 比 高 не За | 在 和 一 以 中 有、 将 本 用 上 ， 地 被 向 与 不 明 的 他 |
| Eng-Arb | يف Ha Ha على He ش لاب لل ال Ha ال ف تساخ ف He علم كد ر دي ا عم | ن ر إ ق ول ف ت ج ي س خ ش خ مل من م ك ي عم م ملا أ يف |
| Eng-Por | на не в I за это да с все того Не при علم The د по те На За У | uma não mais você são que sua apenas pode pessoas somente por de seus está até fazendo qualquer seu muito |

*Table 8.* First 20 tokens obtained by decoding the steering vectors from the top-1 *Translation Head* and top-1 *Language Head* of GEMMA-3-4B-PT for five translation directions. For each direction, we project the mean head output onto the vocabulary space and report the highest-ranked tokens.

| Language Pair | Translation | Language |
|---|---|---|
| Eng-Fra | í ś ū Pérez = ₽ Rö    ization [ ī Pokémon ó чи y    īg Ré | French à pour dé Quebec Québec french é France É ou és rés ém Ré le en é Dé ès |
| Eng-Swh | = í impactful ū 에서 ś And и    y за в чи XNUMX ó [ And | Tanzania African Africa Kenya Kenyan ya Zambia Africans view Mp ( y . Malawi in Rwanda Kenyans m Nigeria |
| Eng-Zho | в y В Sociedad Chiến за ной Trường Employees Employee Nguyễn الاو ' ском ⊏ об Bronx Тру Yıld | （，：其他；对 可以 做 （很 后 其他 或 从 上 此外 两 应。 |
| Eng-Arb | = ś    ₽ í чи    [ And And Grü 에서 ū 有 Pérez Rö impactful | ه عم ـ نأ ي ف ن ط ش ح ج ر ن ا ق ن م ص ب ا ك أ ل ع |
| Eng-Por | í ś ū = ₽ Pérez ī impactful ización y Pokémon czek В    и Rö i | Brazilian Portuguese Brazil brazilian Portugal brasil Brasil portug São brazil Recife Chilean Urugu por Oliveira brasile João soccer ou Brazilian |

*Table 9.* First 20 tokens obtained by decoding the steering vectors from the top-1 *Translation Head* and top-1 *Language Head* of GEMMA-3-12B-PT for five translation directions. For each direction, we project the mean head output onto the vocabulary space and report the highest-ranked tokens.

| Language Pair | Translation | Language |
|---|---|---|
| Eng-Fra | Vegetation Planting στη To Party Plants Cryptocurrency Tooth Öffentlichkeit Rainfall Allowing 砦 ilikom 興 在 甑 Aircraft Skilled στην ของ | RÉ é ÈRE équ à ÉS Caractéristiques É exemple ré exercice ÉRI Rés még ÉE « mé ÉR él fonction |
| Eng-Swh | 砦 στη Plants Planting Party Workforce สำหรับ オフィス Mijn Servicios Desenvolvimento To Kijk Medications Vegetation Σε Tank Jobs Z Partei | ی それ semasa pemer ご ruangan umur pemas kuwa powodu bidang さ penampilan gadgets และ spowod これらの についての ストレ lanjut |
| Eng-Zho | 在 Myself στη To Planting 生地 我 Having Öffentlichkeit Plants Vegetation Tôi 在我 ใน 對 ของ Party στην 特定 Saya | 一 在（出所具有一，用：那分便像一不以 people |
| Eng-Arb | To Planting Vegetation Plants στη สำหรับ Cryptocurrency オフィス Medications Mijn Tôi Saya Про 奕 ณ Myself Servicios Kijk | كلذ شلا بت مل فو عم تسا فلا ع ب نم بلا ف ش نلا عت لذ م ملا ي |
| Eng-Por | Vegetation 砦 στη Planting 甑 Plants Party στην Öffentlichkeit 在我 を持つ 胃 在 奕 ২৩৪ 興 ของ यांच्या VIII 조선 | mesmo pessoas mencion também especific nível há produtos maneira só área você níveis tamanho compartilh geralmente período versão código áreas |

# C. Additional Experiments

## C.1. Impact of the number of shots

Figures 68 and 69 show the result of our ablation study on the impact of the number of the number of few-shot examples. They show the log probability deltas per layer and attention head in GEMMA-3-12B-PT under language and translation corruption respectively for the English to French translation pair. Activation patching results are reported for 0, 1 and 20 shots. Moreover, figure 70 shows the result of these experiments on steering-induced MT performance.

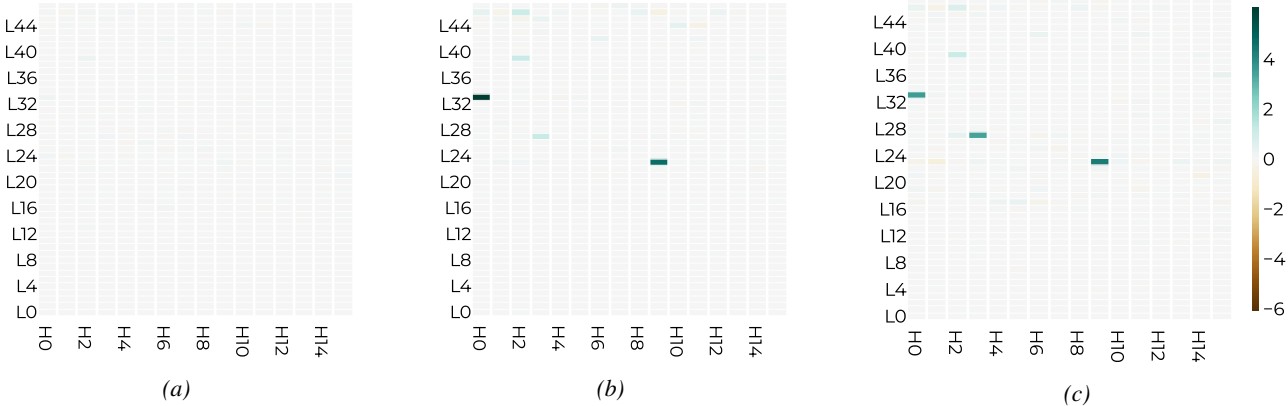

*Figure 68.* Log probability delta per layer and attention head in GEMMA-3-12B-PT under language corruption for the English to French translation pair. We report activation patching results (a-c) for 0, 1, 20 shots, respectively.

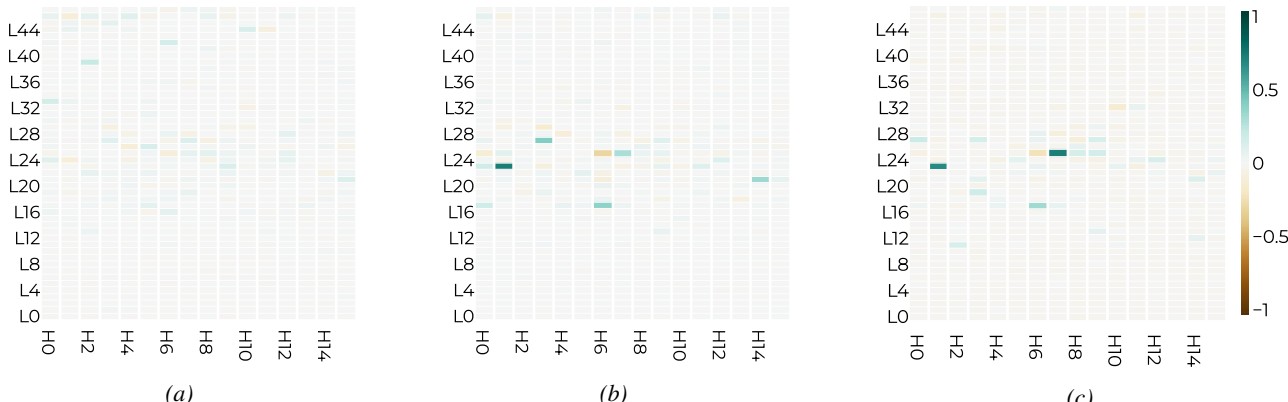

*Figure 69.* Log probability delta per layer and attention head in GEMMA-3-12B-PT under translation corruption for the English to French translation pair. We report activation patching results (a-c) for 0, 1, 20 shots, respectively.

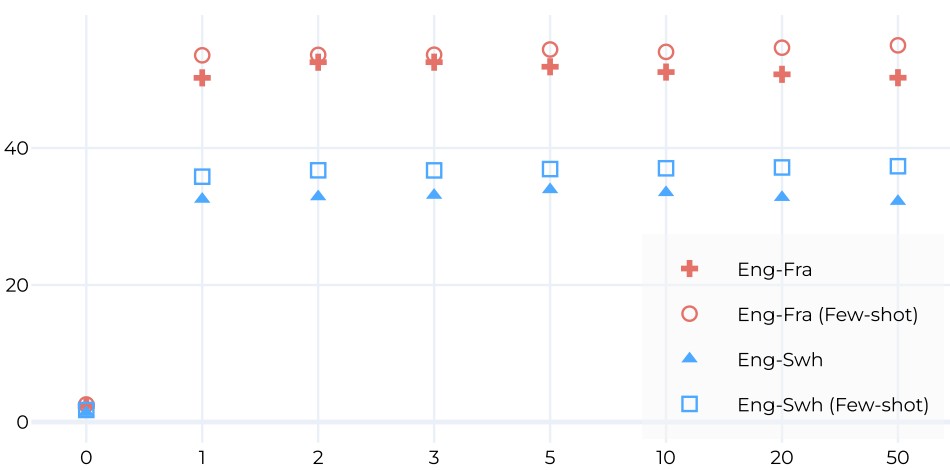

*Figure 70.* Effect of the number of few-shot examples on steering-induced MT performance. We report BLEU scores under an instruction-free zero-shot setup for GEMMA-3-12B-PT when steering $1\%$ of Language and translation heads for 0, 1, 2, 3, 5, 10, 20 and 50 shots. Results are compared against generation in a $n$-shot setup.

