# OpenReview forum: "Translation Heads: Disentangling meaning from language in LLM-based machine translation"
_ICML.cc/2026/Conference — ICML 2026 regular_

### Official Review · Reviewer_QiYd · 2026-03-10

**Soundness:** 3
**Presentation:** 3
**Significance:** 2
**Originality:** 2
**Overall Recommendation:** 3
**Confidence:** 4

**Summary:**

This paper studies the lack of mechanistic explanations for sentence level machine translation, which is still not well understood in large language models. The authors analyse MT through two subtasks: identifying the correct target language and preserving the meaning of the source sentence. They use activation patching together with a KL divergence method to choose the most informative token position, and they evaluate this across 20 translation directions and three model families.

Across Gemma 3, Qwen 3, and Llama 3.2, the main findings are: MT behavior is concentrated in about 1% of attention heads; the two subtasks rely on different groups of heads; and the meaning related vectors transfer well across languages. The paper also tests two types of interventions, steering and ablation. The results show that subtask specific vectors can achieve instruction free translation close to prompted performance, while removing the identified heads breaks the related functions. This provides a more systematic way to control task representations in multilingual systems.

**Compliance With Llm Reviewing Policy:**

Affirmed.

**Final Justification:**

I appreciate the authors’ effort to clarify my earlier concerns, their responses are not sufficiently convincing to resolve them. Therefore, I will maintain my current score.

**Key Questions For Authors:**

- Why was the analysis limited only to attention heads? Do other components of the Transformer architecture, such as the MLP (Multi Layer Perceptron) blocks, also play an important role in these functions?

**Limitations:**

Yes

**Strengths And Weaknesses:**

Strengths

- The study is well motivated, it aims to explore and explain the capabilities of LLMs in machine translation. The proposed approach based on activation patching is sound and intuitive.

- Experiments and analyses cover 20 translation directions and three open source model families. The evaluation is comprehensive which provides strong support for the study’s claims.

- The paper is well written and easy to follow.


Weaknesses

- The paper reduces MT to two subtasks, “target language identification” and “sentence equivalence.” This feels somewhat oversimplified, as machine translation usually involves more complex linguistic processes such as syntactic restructuring, handling tense and aspect, and discourse level coherence. It seems these two subtasks were chosen mainly because they are easier to manipulate experimentally (e.g., swapping target languages or replacing sentences), which may limit the depth of the analysis.

- Some of the patterns reported in this study appear similar to findings in earlier works. While this does provide a form of cross validation, it may reduce the excitement for readers. For example, Zhang et al. (2025) also observed that only a sparse subset of attention heads is critical for MT, and Zhan et al. (2023) pointed out that the first token plays a major role in multilingual translation behavior.

- For an analytical study, it would be more valuable to provide deeper or more fundamental insights. For instance, how can the findings in this paper directly help developers improve MT capabilities in LLMs, or assist practitioners who rely on LLMs for translation tasks?


[1] Zhang et al. Exploring the Translation Mechanism of Large Language Models. In NeurIPS 2025.

[2] Zhan et al. Prefix Text as a Yarn: Eliciting Non-English Alignment in Foundation Language Model. In ACL 2024 Findings.

---

> ### Author Rebuttal · Authors · 2026-03-31
>
> Dear reviewer QiYd, thank you for your review.
>
> > W1: The paper reduces MT to two subtasks, “target language identification” and “sentence equivalence.” This feels somewhat oversimplified
>
> Translating a sentence from a source language into a target language can be understood as producing an equivalent sentence to the input in the target language. This decomposition is not a reductive view but rather the very definition of the MT task. A key aim of the article is to find attention heads that are important for MT.  We draw a line between the goal of the task (i.e. equivalent sentence in target language) and the potential how (e.g. handling tense, syntax restructuring), which are inherently language- and sentence-dependent sub-processes naturally encompassed within sentence equivalence objective.
>
> This decomposition enables a deeper analysis of the task. It allows us to identify which attention heads contribute to different aspects of MT and to characterize their overlap. It also uncovers roles and behaviors that generalize across translation directions, which is central to our findings on both attention heads and task vectors.
>
> > W2: Some of the patterns reported in this study appear similar to findings in earlier works.
>
> The observation that a small number of attention heads play a significant role has been reported in prior work, including Zhang et al. (NeurIPS 2025), which we cite. However their focus is uniquely on word-level MT and they identify sentence-level MT as a natural direction for future research. Our paper builds on and extends these findings to the sentence-level setting: we are among the first to study in a comprehensive manner through the lens of mechanistic interpretability.
>
> The importance of the first token is well established in multilingual settings. For example, appending tokens in a given language to a prompt can induce a model to continue generation in that language. However, our findings go beyond this to show that analyzing the distribution of the first generated token (conditioned on a few-shot prompt) provides a principled way to identify which heads are most relevant for the task.  This is not a trivial finding at the sentence level, and  it was not obvious that the heads responsible for encoding language or sentence equivalence would remain consistent throughout the generated sequence. We provide empirical evidence of this functional continuity across token positions (Figure 8, 9 & 10), which we consider a key contribution. We also include steering experiments that highlight the functional importance of specific heads across a wide range of models and language directions. Furthermore, the task vectors we derive exhibit strong and interpretable properties: (i) equivalence vectors are consistent across directions (allowing transfer between language pairs), and (ii) language vectors decode into tokens characteristic of the target language (Appendix C.3), with similarities emerging across languages that share “similar” features (e.g., Spanish-Portuguese, Japanese-Chinese, Appendix C.2).
>
> > W3: For an analytical study, it would be more valuable to provide deeper or more fundamental insights. How can the findings in this paper directly help developers improve MT capabilities in LLMs, or assist practitioners who rely on LLMs for translation tasks?
>
> We want to highlight that this work is focused on interpretability. The idea is not directly to make LMs perform MT better but rather to see if we can build an understanding of how the task is performed internally. This work provides further evidence that cross-lingual transfer emerges during pretraining, with shared representations whose quality correlates with the amount of data available in each language. We show that this cross-linguality extends to the task level in Machine Translation. Additionally, our findings open up avenues for future research such as parameter-efficient fine-tuning through the selective use of attention heads, and studying the so-called “curse of multilinguality” in MT.
>
> > Q1: Why was the analysis limited only to attention heads? Do other components of the Transformer architecture, such as the MLP (Multi Layer Perceptron) blocks, also play an important role in these functions ?
>
> We do not investigate FFN neurons in this work, although we acknowledge this as an interesting direction for future research. Exploring them falls beyond the scope of this paper, given the breadth of experiments we conduct. We choose to focus on attention heads as they provide a natural and well-studied unit for analysis in mechanistic interpretability (MI). As noted in the paper, prior work has shown that attention heads play a central role in in-context learning setups (e.g., Olsson et al., 2022, In-Context Learning and Induction Heads).

---

> > ### Author Rebuttal · Reviewer_QiYd · 2026-04-04
> >
> > Thanks for authors clarifying my major questions. However, there still need additional experiments or justifications to resolve the concerns.

---

### Official Review · Reviewer_Sqb6 · 2026-03-11

**Soundness:** 3
**Presentation:** 2
**Significance:** 2
**Originality:** 3
**Overall Recommendation:** 3
**Confidence:** 3

**Summary:**

The paper anaylizes the attention head involvement in two aspects, target language identification and sentence meaning equivalence, of the machine translation (MT) process of LLMs, by constructing two corrupted in-context-learning (ICL) settings and compring them with the propoer setting, and calculating the change in log probability after applying activation patching. Results show that two small sets of attention heads are responsible for the two aspects respectively, and steering or ablating them can affect the MT performances across models and langauges.

**Compliance With Llm Reviewing Policy:**

Affirmed.

**Final Justification:**

The authors' rebuttal solves my concern raised in Weakness 1 and 2, and partly 3. The added results contributes to the soundness of the paper. As a result, I increased my soundness and overall rating.

**Key Questions For Authors:**

- Can you provide some output cases of the LLMs under the two corrupted settings?
- Other than attention heads, other model components such as FFN neurons can also affect the MT behavior. Have you investigated their roles in the task?

**Limitations:**

Yes

**Strengths And Weaknesses:**

## Strengths
- The paper identifies small sets of attention heads related to language identification and sentence equivalence.
- Experiments cover multiple LLMs and languages.

## Weaknesses
- The corrupted ICL settings are like "undefined behaviors" of LLMs. What are the models supposed to output when the MT demonstrates are messed up with languages or sentence meaning? Evidences that the LLMs are doing the language identification or sentence equivalence tasks should be provided, such as language usage accuracy and semantic similarity, rather than mere translation scores.
- In Figure 3, contradictive trends of BLEU and MetricX-24 scores can be observed, without further discussion on why the BLEU trend is perferred.
- The motivation of the paper is not clear enough. If you are interested in the two aspects of MT, why not studying transformer-based multilingual MT systems such as mBART, M2M, NLLB, etc.? There are easier ways to control the output language (by language tag) rather than using corrupted demonstrations. Also, post-trained LLMs adopted in the experiments such as Qwen3 do not perform ICL tasks as good as base models, which could compromise the representativity of the experimental results.

---

> ### Author Rebuttal · Authors · 2026-03-31
>
> Dear reviewer Sqb6, thank you for your review.
>
> > W1: The corrupted ICL settings are like "undefined behaviors"
> > Q1: [...] some output cases of the LLM under the two corrupted settings?
>
> We want to identify the attention heads that are important for each core aspect of MT as defined by the task itself (producing an equivalent sentence to an input in a target language). Activation patching relies on a corrupted prompt (obtained by minimally modifying a clean prompt) which is designed in a way to prevent the LLM from accurately performing the task. While these corrupted prompts could be interpreted as inducing "undefined" behaviors, our objective is not to analyze the exact outputs they produce. Rather, we only require that they yield incorrect outputs with respect to the task. For instance, a corruption can constrain the output to a certain class (e.g., sentences in correct language but with an incorrect meaning). What matters is that, by intervening on activations derived from a corrupted prompt, we are able to recover outputs consistent with those produced by clean MT prompts.
>
> We use two types of corruption, each deliberately designed to isolate one of the two aspects of MT:
> - A1: The generation of an equivalent sentence, independently of the language, which we term “sentence equivalence”.
> - A2: Producing a sentence in the target language, independently of its meaning (target language "identification").
>
> For each aspect (e.g., A1), the corrupted prompt encourages the LLM to produce an output that satisfies A1 while not satisfying A2. By contrasting the outputs with what we obtain with a clean prompt (whose output satisfies both A1 and A2), we can identify which components are critical for enabling A1. We argue that each corrupted prompt fulfills its role. To prove this and answer your question, we provide examples of outputs in 5-shot under both corruptions with `Gemma-12b-pt` in Eng→Zho. We also show aggregated target language accuracies and BLEU scores.
>
> **Query**: To finish, Turkish dance group Fire of Anatolia performed the show "Troy".
>
> **No Corruption**
> - Eng-Zho: 最后，土耳其舞蹈团“安纳托利亚之火”表演了“特洛伊”节目。
>
> **Language Corruption**
> - Eng-Zho: وفي الختام، قدمت فرقة الرقص التركية "نار الأناضول" عرض "طروادة".
>
> **Sentence equivalence Corruption**
> - Eng-Zho: 1990 年，美国宇航局发射了伽马射线观测卫星 (CGRO)，它发现了伽马射线暴。
>
> **Aggregated Results across all studied directions in the paper**
>
> *Under Language Corruption*
> - Avg. BLEU: 5.1
> - Language Acc.: 2.2%
> - Avg. BLEU with reference in the identified language: 30.5
>
> *Under Sentence equivalence Corruption*
>
> - Avg. BLEU: 2.1
> - Language Acc.: 99.3%
> - Avg. BLEU with reference in the identified language: 2.1
>
> We observe that the language corruption only hides the language identity while preserving the "equivalence behavior" and that the sentence equivalence corruption hides the "equivalence behavior" while keeping the target language.
>
> > W2: In Figure 3, contradictive trends of BLEU and MetricX-24 scores...
>
> As mentioned in the main paper (Section 4, Evaluation metrics), lower MetricX-24 scores mean better translation, whereas a higher BLEU score is better. The two metrics are therefore consistent with each other in Figure 3. Specifically, when comparing (a) and (c), lower (i.e., worse) BLEU scores appear in lighter shades of blue and correspond directly to higher (i.e., worse) MetricX-24 scores, which are shown in lighter shades of orange. We will clarify this in the caption to aid readability.
>
> > W3: The motivation of the paper is not clear enough. [...] There are easier ways to control the output language [...]
>
> The aim of the paper is to analyse LLMs in the context of MT. LLMs are able to do MT without being trained in a supervised way to do so, and this via in-context learning (ICL; as we study here). Encoder-decoder models are not the study of this article and the analysis is not designed for them. The aim is not to just control the output language in MT (i.e. with language tags), but to understand the mechanisms of MT by LLMs. Alternatively, we choose not to include language tags in our ICL prompt template to avoid, as we stated in our paper, effects from task specific instructions. Moreover, having the LLM infer the task solely based on demonstrations helps transfer our setup to other tasks.
>
> **Regarding Qwen3: as stated in Section 4, we exclusively use base models (see Appendix A.1).**
>
> > Q2: [...] other model components such as FFN neurons can also affect the MT behavior?
>
> We do not investigate FFN neurons in this work, although we acknowledge this could be an interesting direction for future research. Exploring them falls beyond the scope of this paper, given the breadth of experiments we conduct. We choose to focus on attention heads as they provide a natural and well-studied unit for analysis in mechanistic interpretability (MI). As noted in the paper, prior work has shown that attention heads play a central role in ICL setups (e.g., Olsson et al., 2022, ICL and Induction Heads).

---

> > ### Author Rebuttal · Reviewer_Sqb6 · 2026-04-02
> >
> > The authors' reply solves most of my concerns, and I will adjust my ratings accordingly.

---

> > > ### Author Response · Authors · 2026-04-08
> > >
> > > Dear Reviewer Sqb6, thank you for acknowledging our rebuttal and reassessing our study.
> > >
> > > Regarding the remaining points in W3, we would like to provide further clarification.
> > >
> > > The focus of this study on decoder-only models is a deliberate choice motivated by multiple factors: (i) decoder-only models match or outperform supervised encoder-decoder models, particularly when translating between high-resource languages (Hendy et al. 2023, "How Good are GPT Models at MT?"; Zhu et al., NAACL 2024, "Multilingual MT with LLMs"; Omnilingual MT Team et al. 2026, "Omnilingual MT: Machine Translation for 1600 languages"), and (ii) Mechanistic Interpretability findings on MT in decoder-only models can broaden our general understanding of this type of model which is used across a wide variety of tasks.
> > >
> > >
> > > Unlike decoder-only LLMs, MT systems such as M2M100 or NLLB-200 are trained in a supervised fashion on parallel data, with special tokens to guide the target language of generation (e.g. `eng_Latn`, `fra_Latn` etc. for NLLB-200). Decoder-only LMs, by contrast, perform sentence-level MT solely through In-Context Learning: they infer the task from input-output demonstrations and execute it accurately, all without being explicitly trained for translation. This raises an important open question: how do these models organize internally to perform MT? In particular, how do they handle its two aspects: producing a sentence semantically "equivalent" to a source, but expressed in a specific target language? Their lack of explicit (supervised) MT training makes them a particularly interesting object of study.
> > >
> > > As explained in our previous rebuttal, each aspect is isolated through a dedicated corruption, allowing us to improve our understanding of sentence-level MT in decoder-only models. We acknowledge that studying dedicated systems such as supervised encoder-decoders remains valuable. However, we consider this to be beyond the scope of the present paper, though we believe the insights derived from our work would likely help in such settings.
> > >
> > >
> > > We hope these clarifications address your remaining concerns.

---

### Official Review · Reviewer_CYfU · 2026-03-15

**Soundness:** 3
**Presentation:** 3
**Significance:** 3
**Originality:** 2
**Overall Recommendation:** 5
**Confidence:** 3

**Summary:**

This work applies activation patching for machine translation (MT) setting of language model prompting in order to investigate the meaning across languages. Basic idea is to prompt language models with clean and corrupted prompts and investigate the difference of the confidence in translation using the corrupted prompt after replacing the activations at intermediate layers with the clean ones. Two kinds of corruptions were investigated by changing demonstrations in the prompt: one by replacing the target language but preserving the meaning of the source sentence, and the other by preserving the target language but using random sentence without preserving the meaning. Experiments show the findings: only a small subset of activations are involved in MT setting, and they are rather consistent regardless of the translation directions. Corrupting these heads lead to degradation in performance, clearly demonstrating the importance of the activations.

**Compliance With Llm Reviewing Policy:**

Affirmed.

**Final Justification:**

Given the rebuttal, it resolves my concern regarding the design of the prompts.

**Key Questions For Authors:**

- I'd like to see if using a single corruption method would be sufficient for the investigation or not. If not, I'd like to see the reason.

**Limitations:**

yes.

**Strengths And Weaknesses:**

Strengths
- This work is investing an interesting topic of language models focusing on MT setting, what activations will have a large contribution. The methodology is sound enough to identify the important activations for a sentence-wise meaning.
- The proposed approach employs the first most important token position to identify the activations to investigate the difference of meanings in translation. I feel the approach is also well-motivated and empirically analyzed in section 6.

Weaknesses
- This work use two kinds of corrupted prompts, but it is not clear why this work employ both to find the activations. I'd like to see why only a single one might not work, at least empirically, or any reasons for using both.
- It is off-topic, but I'd like to find discussion on low-resource languages, whether the findings could be extended to such languages or not.

---

> ### Author Rebuttal · Authors · 2026-03-31
>
> Dear reviewer CYfU, thank you for your review.
>
> > W1: This work uses two kinds of corrupted prompts, but it is not clear why this work employs both to find the activations.
>
> > Q1: I'd like to see if using a single corruption method would be sufficient for the investigation or not.
>
> The main goal of the article is to find attention heads that are important for MT. We could have modelled the task as a whole, but we decided to study the task from the two aspects that intuitively characterise the MT task (sentence equivalence and target language identification). We need to use two prompts, each corresponding to an attempt to isolate each aspect separately. This decomposition helps identify which attention heads are responsible for different aspects of the task, as well as the extent of their overlap. It also reveals roles and behaviors that are shared across translation directions, which in turn informs the construction of task vectors that can be compared and even transferred across settings (Figure 5, Appendix C.2, Figure 70).
>
> One of our main findings is that (i) the important heads tend to be the same across language directions (i.e. there are consistent patterns) and (ii) the two isolated aspects (meaning preservation and target language identification) do lead to distinct sets of important heads (i.e. the two subtasks are represented in different ways by the model).
>
> We did run an additional experiment in response to your question, which involves creating a single corrupted prompt that incorporates examples from both types of corruption (ii.e., the in-context examples consist of source sentences paired with randomly selected sentences from multiple languages, resulting in incorrect translations in mismatched target languages.). For instance, for en-ja, here is an example of a merged corrupt prompt:
>
> ```
> Q: The death toll is at least 15, a figure which is expected to rise.
> A: Elas pareciam câmaras. Ele foi a primeira pessoa a observar células mortas.
>
> Q: Gosling and Stone received nominations for Best Actor and Actress respectively.
> A: Стандарт 802.11n работает на обоих частотах – 2.4 ГГц и 5.0 ГГц.
> …
> Q: They do this by emitting a tiny particle of light called a "photon".
> A:
> ```
>
> The results are consistent with our previous findings. As shown in the activation patching figure, the important heads found for the merged corrupt prompt are the exact union of those found with each corruption separately ([see image](https://ibb.co/0pPZqtjr)). We also provide results for the steering experiment ([see image](https://ibb.co/rfsTLgdL), with the zero-shot prompt `Q: {source sentence}\nA:`) that showed similar trends as ~1% of the identified heads are sufficient to recover a translation quality comparable to that obtained with explicit task instructions.
>
> However, this "merged setup" does not inform us on the necessity of subset of heads as shown in the figure 3 where we can see that both subset of language heads and translation heads are necessary. It also prevents us from conducting cross-lingual experiments that rely on distinguishing language-related and translation-related heads to analyze task-vector similarity and transferability (Section 5: Figure 5).
>
> > W2: It is off-topic, but I'd like to find discussion on low-resource languages, whether the findings could be extended to such languages or not
>
> Thank you for asking this question, we believe it is an important one and will add the following discussion in the revised version. In fact, our study does include Swahili and Wolof as mid- and low-resource languages, respectively, with per-language activation patching (Figures 17 and 18), steering (Figure 43), and ablation (Figure 56) results for the `Gemma-12b-pt` model reported in the appendix. Swahili exhibits activation patching patterns consistent with the other languages in our study, suggesting that our findings generalize reasonably well to mid-resource languages.
>
> Wolof is different however for both corruption types: head activations are weaker than for other languages, although they do have a similar set of top heads. This suggests the model has started building representations for Wolof in the same way as other languages but was unable to achieve clear ones due to extreme resource scarcity ([0.0003%](https://commoncrawl.github.io/cc-crawl-statistics/plots/languages) of Wolof in the last CC crawl). It is possible that the functional specialization we identify in translation heads may therefore emerge only when the model has sufficient proficiency in a given language.
>
> Additionally, our analysis of translation and language head outputs (Appendix C.2, Figure 70) confirms that the internal representations for Wolof are the least aligned with those of other languages. Investigating how these language directions develop as a function of language exposure and model scale remains a promising direction for future work.

---

> > ### Author Rebuttal · Reviewer_CYfU · 2026-04-03
> >
> > Thank you very much for your responses and additional experiments that sound interesting to me.
> >
> > I've adjusted my scores, accordingly.

---

### Official Review · Reviewer_7NHT · 2026-03-31

**Soundness:** 3
**Presentation:** 4
**Significance:** 2
**Originality:** 3
**Overall Recommendation:** 4
**Confidence:** 4

**Summary:**

This paper attempts to use mechanistic interpretability methods, in particular the activation patching of Vig et al. (2020) to identify attention heads to understand how LLMs encode various translation functions, in particular here two: determining the target language to use and preserving the sentence's meaning. Various results are presented in particular showing that a small number of heads can be identified such that steering them can produce good results without needing to provide few shot examples to an LLM.

**Compliance With Llm Reviewing Policy:**

Affirmed.

**Key Questions For Authors:**

1. On p.8 you leave hanging that "However, their most important head ([30,18]) was nowhere to be found in our classification. Did you make any attempt to identify what it was? Surely you could fairly easily do some patching of it to see what wiggling it is controlling? This might be a useful start in exploring what else is being controlled in the MT problem by various heads.

**Strengths And Weaknesses:**

# Strengths

The paper was interesting and technically sound.

The paper was presented clearly in a well-structured paper, and the work is done very thoroughly, as the fact that the paper + supplementary material has 70 figures attests!

The paper has some clear interesting results.

There is originality in the paper: Not really in the methods, but this paper extends mechanistic interpretability methods to sentence-to-sentence machine translation, where it has not previously been thoroughly studied.

# Weaknesses

The significance is only meaning. Nothing stunning of broad generality is found. It is not too surprising that there is attention to semantic content of the source and desired language of the output in an LLM trained to translate!

Something that troubled me a little is that the two tasks focused on seem very uneven in size: The choice of the target language is a single n-way classification task for n around 200, whereas semantic translation of the source is a huge task with billions of choices. I wondered about other choices that weren't being modeled: The fluency of the generated language, and finer-grained choices in the style of the translation, such as a formal to informal spectrum.

---

> ### Author Rebuttal · Authors · 2026-03-31
>
> Dear reviewer 7NHT, thank you for your review.
>
> > W1: The significance is only meaning. Nothing stunning of broad generality is found. It is not too surprising [...] trained to translate!
>
> A central aspect of our work is that the models we study are not explicitly trained for MT; instead, we use base LLMs that perform MT via ICL. Our goal is to understand how their internal components organize to solve this task. As you note, we are among the first to study sentence-level MT at this level of depth with MI.
>
> Given the definition of MT as producing a sentence equivalent to the source in a target language, we decompose the task into two components (sentence equivalence and target language identification) and analyze the corresponding attention heads separately. This decomposition, which we view as a contribution in itself, enables the study of cross-lingual behaviors that have not been previously examined in this setting. While some aspects may appear intuitive, we empirically validate them through systematic experiments. We find that the sets of attention heads associated with each aspect are distinct, yet consistent across translation directions. In particular, equivalence-related representations exhibit high cosine similarity across languages (Figure 70) and demonstrate transferability via steering experiments (Figure 5), supporting the existence of a language-agnostic notion of sentence equivalence. Similarly, language-related vectors align with the expected target language behavior (Section 5). Finally, we reconcile our findings to prior work on word-level MT.
>
> > W2: [...] two tasks focused on seem very uneven in size: [...]. I wondered about [...]: The fluency of the generated language, and finer-grained choices in the style of the translation, such as a formal [..].
>
> We agree that there is an inherent imbalance between the two aspects of MT. Target language identification is comparatively easier, as it involves selecting from a limited set of possible languages (e.g., those seen during pretraining), which is further constrained by the prompt context. In contrast, sentence equivalence is more abstract. Generation is token-by-token with a vocabulary of size |V|, but conditioning naturally rules out many implausible tokens at each step. More fine-grained phenomena, such as formality or style, add complexity to sentence-level MT analysis from an MI perspective. These aspects are inherently language- and sentence-dependent and are encompassed within the broader notion of sentence equivalence.
>
> Our study does include Swahili and Wolof as mid- and low-resource languages, respectively, with per-language activation patching (Figs 17 and 18), steering (Fig 43), and ablation (Fig 56) results for `Gemma-12b-pt` reported in the appendix. Swahili exhibits activation patching patterns consistent with the other languages in our study, suggesting that our findings generalize reasonably well to mid-resource languages.  Wolof is different however for both corruption types: head activations are weaker than for other languages, although they do have a similar set of top heads. This suggests the model has started building representations for Wolof in the same way as other languages but was unable to achieve clear ones due to extreme resource scarcity. Additionally, our analysis of translation and language head outputs (App. C.2, Figure 70) confirms that the internal representations for Wolof are the least aligned with those of other languages.
>
> > Q1: Surely you could fairly easily do some patching of it to see what wiggling it is controlling? [...].
>
> We conducted additional experiments with LLaMA-2-7B on English→Chinese to compare our results with those of Zhang et al. (NeurIPS 2025) in word-level MT. Your question also helped us identify a typo in the paper: we incorrectly referred to their top head as [30, 18] instead of [31, 8]. Thank you, we will correct it in the next revision.
>
> For head [31, 8], we find $\Delta \approx 0$: ablating it leaves BLEU nearly unchanged (18.3 → 18.2). In contrast, ablating [30, 18] reduces performance to 16.7. Steering (in 0-shot `Q:x\nA:`) [31, 8] with different values of α has no positive effect (outputs remain in English), while steering [30, 18] induces Chinese token generation, though not enough to recover performance.
>
> Our method differs from theirs in that they identify heads by comparing the next-token logits under a clean prompt without intervention to those obtained after intervening with activations from a corrupted setting. Additionally, their analysis focuses on the average relative change in logits (i.e., percentage error). Their corruption is `{src_lang}: {tgt_word} - {tgt_lang}: "{tgt_word}"` → `{src_lang}: {tgt_word} -  Nothing: "{tgt_word}"` and the experiments we run which such a corruption resulted in a set of heads not including language heads identified with our corruption.
>
> Given the time at which we received the review, we could not conduct all experiments we wanted.

---

### Decision · Program_Chairs · 2026-04-30

**Decision:**

Accept (regular)

**Comment:**

This paper makes use of activation patching to identify attention heads needed to understand how LLMs encode translation focusing on two properties: determining the target language to use and preserving the sentence's meaning. They show that a small number of heads can be identified such that steering them can produce good results without needing to provide few shot examples to an LLM.


Reviewers found the work to be interesting with extensive experiments covering several languages including mid- to- low-resource ones like Swahili and Wolof (Reviewer 7NHT and CYfU). There are few issues on the originality of the method used but I believe this does not diminish the work. Many other issues have been addressed in the rebuttal especially similarity in pattern to earlier works.